# A Unified Framework for Diffusion Model Unlearning with f-Divergence

## Abstract

Machine unlearning aims to remove specific knowledge from a trained model. While diffusion models (DMs) have shown remarkable generative capabilities, existing unlearning methods for text-to-image (T2I) models often rely on minimizing the mean squared error (MSE) between the output distribution of a target and an anchor concept. We show that this MSE-based approach is a special case of a unified $f$-divergence-based framework, in which any $f$-divergence can be utilized. We analyze the benefits of using different $f$-divergences, that mainly impact the convergence properties of the algorithm and the quality of unlearning. The proposed unified framework offers a flexible paradigm that allows to select the optimal divergence for a specific application, balancing different trade-offs between aggressive unlearning and concept preservation.

## 1 Introduction

Deep learning algorithms for T2I generation based on DMs have demonstrated remarkable success and widespread adoption in recent years. Some notable examples are Stable Diffusion (Rombach et al., 2022), DALL·E 2 (Ramesh et al., 2022), and Imagen (Saharia et al., 2022), which find a large variety of applications in real-world scenarios. However, a significant challenge associated with these algorithms stems from the fact that they are trained over large scale datasets (such as LAION-5B (Schuhmann et al., 2022)) that are scraped from the Internet, and contain not safe for work (NSFW) content, including explicit material (Schramowski et al., 2023), and copyrighted works, such as artistic styles and even personal information (Jiang et al., 2023; Carlini et al., 2023). Since DMs are able to learn and memorize all these contents (Somepalli et al., 2023b), it is crucial to develop algorithms that erase specific concepts from the final trained models. To force a model to forget a concept, the trivial solution would be to re-train the model with a curated dataset where the specific concept has been removed. However, this is impractical, as it is resource-intensive and time-consuming, and in some cases the training dataset could have been deleted. Therefore, it is fundamental to develop techniques that would allow a given trained diffusion model to unlearn a specific concept. Many algorithms proposed for DM unlearning are grounded on a shared idea: shifting the model's prediction corresponding to the concept to be unlearned (referred to as "target") towards prediction of a substitute concept (referred to as "anchor"). The latter can be selected in many different ways, depending on the user's goal: null concept, hyper-class, semantically close/distant to the concept to forget. Usually, this is achieved by minimizing an MSE-based loss (Gandikota et al., 2023; Kumari et al., 2023; Huang et al., 2024), derived from formulating the problem as the minimization of the Kullback-Leibler (KL) divergence between two Gaussian distributions, conditioned on the two concepts, target and anchor, characterizing the image generation process.

In this paper, instead of minimizing the standard KL divergence, we propose minimizing the $f$-divergence between the two distributions conditioned on the target and anchor concept. For specific $f$-divergences, we derive the loss function for DM unlearning from the closed-form expression of the $f$-divergence between two Gaussian distributions. These loss functions, similarly to the standard MSE, benefit from a simple implementation and training complexity, while differing from the MSE in terms of convergence properties. When the closed-form expression does not exist, we use the variational representation of the $f$-divergence formulated as a min-max problem, which allows us to tackle the DM unlearning task with *any* $f$-divergence. We refer to the proposed method as $f$-divergence-based DM unlearning ($f$-DMU).

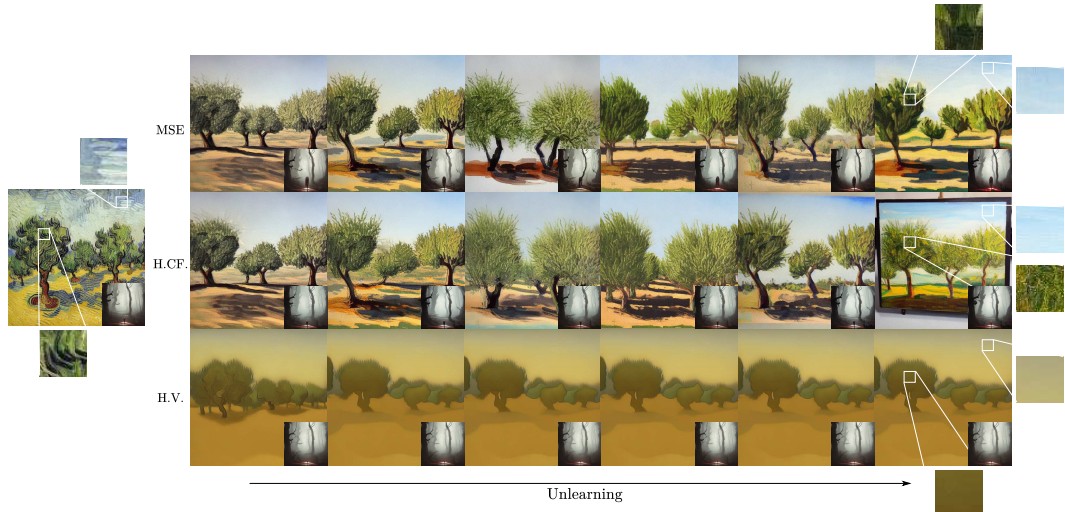

Figure 1: Unlearning a concept using different $f$-divergences yield different unlearning dynamics (top: mean squared error, middle: Hellinger closed-form, bottom: Hellinger variational). The inset in the bottom right corner of each image reports a not erased concept. The white boxes are zoomed-in portions to show the removal of the Van Gogh brushstroke style.

To the best of our knowledge, $f$-DMU is the first approach based on general $f$-divergences for the DM unlearning task. We theoretically study the gradients of the $f$-DMU closed-form objective functions and empirically verify the theoretical findings. Then, we theoretically prove the local convergence property of the proposed min-max formulation, and relate the choice of different $f$-divergences with the algorithm convergence speed. Finally, we perform an extensive numerical analysis to compare results with different $f$-divergences, and we demonstrate the superiority of the proposed unified framework with respect to the standard MSE approach. In particular, we empirically demonstrate that the deployment of $f$-divergences outperforms the MSE loss consistently across different unlearning scenarios, such as style ablation and object ablation.

## 2 PRELIMINARIES

### 2.1 RELATED WORK

In this section, we present a condensed version of the related work in which we focus only on methods in the context of DMs, while we defer to Appendix A for an extended overview.

**Post-processing techniques** Post-processing techniques target the elimination of unsafe generated images through the usage of filtering or inference guiding. Stable Diffusion (SD) (Rombach et al., 2022) adopts a NSFW filter that removes all generated images with embeddings close to those of 17 pre-chosen nudity concepts (Rando et al., 2022). Safe Latent Diffusion (SLD) (Schramowski et al., 2023) is applied during inference and acts by repelling the generation from unsafe contents. Another inference guidance-based approach is SAFREE (Yoon et al., 2025), which can be applied for both image and video generation. The main drawback of post-processing algorithms is that it is possible to remove them from the inference pipeline to allow the model to generate what should have been forgotten.

**Fine-tuning techniques** Fine-tuning approaches modify the weights of a trained model. Many fine-tuning approaches share the common idea of aligning the target and anchor concepts by minimizing the KL divergence. Erased Stable Diffusion (ESD) (Gandikota et al., 2023) (and other methods derived from it (Huang et al., 2024)) fine-tunes the model to align the probability distributions of the model's output fed with a target concept and a neutral concept. Concept Ablation (CAbl) (Kumari et al., 2023) minimizes the MSE between the model's prediction corresponding to the target and anchor concept. With the goal of finding the best anchor concept, Bui et al. propose adversarial

learning frameworks with a loss function comprising two MSE losses, and show that the best anchor concept should be semantically close to the target (Bui et al., 2024; 2025). This knowledge is applied to design an MSE loss that shifts the target concept to semantically close concepts (Thakral et al., 2025). While the previous algorithms are limited in using loss functions based on the KL divergence between the corresponding probability distributions, another line of work proposes to update the model weights in closed-form, leading to faster computations (Gandikota et al., 2024; Gong et al., 2024; Lu et al., 2024). These methods also target the minimization of the MSE, but the loss is formulated directly on the attention weights. However, these approaches are constrained in modifying cross-attention weights, while, in principle, distribution-based methods can update any network parameter and work for different models, such as flow-based generative models (Zhang et al., 2025). Domain Correction (DoCo) (Wu et al., 2025) extends CAbl by using a GAN-based (Goodfellow et al., 2014) loss function. We show in Appendix A that DoCo and CAbl's loss functions can be obtained as special cases of the proposed $f$-DMU framework.

## 2.2 DIFFUSION MODELS

DMs (Sohl-Dickstein et al., 2015; Ho et al., 2020) are state-of-the-art generative models that consist of two components that can be modeled by Markov chain processes. In the forward process, Gaussian noise (referred to as $\epsilon$) is gradually added to an input image $\mathbf{x}_0$ over multiple steps $t \in [0, \ldots, T]$, to obtain $\mathbf{x}_T \sim \mathcal{N}(0, I)$. At each time step, the noisy image can be obtained as $\mathbf{x}_t = \sqrt{\alpha_t}\mathbf{x}_{t-1} + \sqrt{1 - \alpha_t}\epsilon$, where $\alpha_t$ regulates the noise effect. In the reverse process, $\mathbf{x}_T$ is transformed - following a transformation that is the inverse of the forward process - to obtain a denoised image using a denoising network $\Phi(\mathbf{x}_t, \mathbf{c}, t)$, where, for T2I models, the concept $\mathbf{c}$ is a text prompt. The denoising process is characterized as $p_\Phi(\mathbf{x}_0, \ldots, \mathbf{x}_T|\mathbf{c}) = p(\mathbf{x}_T) \prod_{t=1}^{T} p_\Phi(\mathbf{x}_{t-1}|\mathbf{x}_t, \mathbf{c})$, where $p_\Phi(\mathbf{x}_{t-1}|\mathbf{x}_t, \mathbf{c})$ describes the probability of $\mathbf{x}_{t-1}$ given the noisy image $\mathbf{x}_t$ and the concept $\mathbf{c}$.

## 2.3 $f$-DIVERGENCE

Let $p(\mathbf{x})$ and $q(\mathbf{x})$ be two probability density functions on domain $\mathcal{X}$. The $f$-divergence between $p(\mathbf{x})$ and $q(\mathbf{x})$ is defined as (Ali & Silvey, 1966; Csiszár, 1967)

$$D_f(p||q) = \int_\mathcal{X} q(\mathbf{x}) f\left(\frac{p(\mathbf{x})}{q(\mathbf{x})}\right) d\mathbf{x}, \tag{1}$$

where $p \ll q$ (i.e., $p$ is absolutely continuous with respect to $q$) and the *generator function* $f : \mathbb{R}_+ \longrightarrow \mathbb{R}$ is a convex, lower-semicontinuous function such that $f(1) = 0$. The KL divergence is a special case of $f$-divergence, where $f(u) = u \log u$. The variational representation of $f$-divergence (Nguyen et al., 2010) reads as

$$D_f(p||q) = \sup_{T:\mathcal{X}\to\mathbb{R}} \{\mathbb{E}_p[T(\mathbf{x})] - \mathbb{E}_q[f^*(T(\mathbf{x}))]\}, \tag{2}$$

where $T$ is a parametric function (e.g., a neural network) and $f^*$ represents the *Fenchel conjugate* of $f$, defined as $f^*(t) = \sup_{u\in dom_f}\{ut - f(u)\}$, with $dom_f$ being the domain of $f$. The supremum in equation 2 is attained for $T^\diamond(\mathbf{x}) = f'(p(\mathbf{x})/q(\mathbf{x}))$, where $f'$ is the first derivative of $f$.

## 3 CONCEPT ERASING WITH $f$-DIVERGENCE

In this paper, we present an $f$-divergence-based framework able to unlearn a target concept $\mathbf{c}^*$ from a DM. The concept $\mathbf{c}^*$ is erased by shifting the model generation corresponding to $\mathbf{c}^*$ to the generation attained using an anchor concept $\mathbf{c}$. Kumari et al. (2023) achieved this by minimizing the KL divergence between the reverse processes of the original and unlearned DMs, i.e., $D_{KL}(p_\Phi(\mathbf{x}_{(0\ldots T)}|\mathbf{c})||p_{\hat{\Phi}}(\mathbf{x}_{(0\ldots T)}|\mathbf{c}^*))$. Leveraging the Markov property of diffusion processes (see Appendix B.1 for the derivation details), Kumari et al. showed that such an objective can be rewritten as the KL divergence between the DM outputs at specific time instants. We generalize this idea using the $f$-divergence, formulating the unlearning problem as

$$\min_{\hat{\Phi}} \mathbb{E}_{p_\Phi(\mathbf{x}_t|\mathbf{c})}\left[D_f\left(p_\Phi(\mathbf{x}_{t-1}|\mathbf{x}_t, \mathbf{c})||p_{\hat{\Phi}}(\mathbf{x}_{t-1}|\mathbf{x}_t, \mathbf{c}^*)\right)\right]. \tag{3}$$

The KL-based loss employed in prior works can be recovered from equation 3 using the KL divergence. Although theoretically the optimal distribution that minimizes equation 3 is the same for any $f$-divergence, different $f$-divergences have different convergence properties and mode-seeking behaviors (Li & Farnia, 2023). Notably, the divergence-based formulation in equation 3 allows any choice of the anchor concept $\mathbf{c}$. For instance, $\mathbf{c}$ can be chosen as a neutral concept (Gandikota et al., 2023), superset concept (Kumari et al., 2023), a concept with high semantic similarity to $\mathbf{c}^*$ (Bui et al., 2025), or a concept with low semantic similarity to $\mathbf{c}^*$ (George et al., 2025). In the literature, there is not an agreement on the best anchor concept to use. For instance, while the superclass or semantically close options are usually considered the best options, since they are more similar to the target prompt, they appear to be more sensitive to subsequent fine-tuning of the model (George et al., 2025), leading to possible remembering of forgotten concepts. On the contrary, semantically distant concepts are more robust to a future fine-tuning, but lead to unexpected generation and worse erasure effectiveness. In addition, divergence-based objectives also apply to flow-based generative models (Lipman et al., 2022), contrary to architecture-specific approaches (Zhang et al., 2025).

Previous work solved equation 3 when the chosen $f$-divergence is the KL by using the fact that the KL divergence between Gaussian distributions leads to the MSE between their means. In this paper, we extend this solution to any $f$-divergence. In particular, it is possible to express $D_f(P||Q)$, and thus equation 3, in closed-form when $P$ and $Q$ are Gaussian, for a subset of divergences. In the following, we provide three examples: Jeffreys divergence, squared Hellinger distance, and Pearson $\chi^2$ divergence. We extend this approach to an entire subset of $f$-divergences, referred to as $\alpha$-divergences (Amari, 1985; Sourla et al., 2024) in Appendix B.1.1. We defer to Appendix B.1.1 for the calculations.

**Jeffreys divergence**: Jeffreys divergence between two probability distributions $P$ and $Q$ is defined as $D_J(P||Q) = D_{KL}(P||Q) + D_{KL}(Q||P)$. The objective function is reported in Appendix B.1.1.

**Squared Hellinger distance**: Squared Hellinger distance between two probability distributions $P$ and $Q$ can be expressed in terms of the Bhattacharyya coefficient ($BC(P,Q)$) as $H^2(P,Q) = 1 - BC(P,Q)$. The objective function based on squared Hellinger distance reads as

$$\mathcal{J}_H(\hat{\Phi}) = \mathbb{E}_{\mathbf{x},\mathbf{c}^*,\mathbf{c},t}\left[-\omega_t \exp\left\{-||\Phi(\mathbf{x}_t,\mathbf{c},t) - \hat{\Phi}(\mathbf{x}_t,\mathbf{c}^*,t)||_2^2\right\}\right], \tag{4}$$

where $\Phi(\cdot)$ and $\hat{\Phi}(\cdot)$ are the outputs of the original DM and of the DM during unlearning, respectively.

$\chi^2$ **divergence**: The closed-form expression for $\chi^2$ divergence between two Gaussian random variables $P$ and $Q$ exists under a mild assumption on the variance, which will always hold in our case. The objective function based on the Pearson $\chi^2$ divergence becomes

$$\mathcal{J}_{\chi^2}(\hat{\Phi}) = \mathbb{E}_{\mathbf{x},\mathbf{c}^*,\mathbf{c},t}\left[\omega_t \exp\left\{||\Phi(\mathbf{x}_t,\mathbf{c},t) - \hat{\Phi}(\mathbf{x}_t,\mathbf{c}^*,t)||_2^2\right\}\right]. \tag{5}$$

In this work, we follow the standard practice of fixing the variance of DMs to a constant value. Thus, our loss functions do not include a variance term, even though closed-form expressions for $f$-divergences between Gaussians with different variances are available (Appendix B.1.1). Although any $f$-divergence between two multivariate normal distributions with the same covariance matrix $\Sigma$ is an increasing function of their Mahalanobis distance (Ali & Silvey, 1966; Nielsen & Okamura, 2024) $\Delta_\Sigma(\mu_P, \mu_Q) = \sqrt{(\mu_Q - \mu_P)^T \Sigma^{-1}(\mu_Q - \mu_P)}$, not all $f$-divergences can be expressed in closed-form. One example is the total variation distance (Nielsen & Okamura, 2024).

Driven by the intent of providing a unified $f$-divergence-based framework, we present a comprehensive formulation that allows us to solve the problem in equation 3 by using *any* $f$-divergence by adopting its variational representation (more details in Appendix B.1.3)

$$\min_{\hat{\Phi}} \max_{T} \mathbb{E}_{p_\Phi(\mathbf{x}_t|\mathbf{c})}\left[\mathbb{E}_{p_\Phi(\mathbf{x}_{t-1}|\mathbf{x}_t,\mathbf{c})}[T(\Phi)] - \mathbb{E}_{p_{\hat{\Phi}}(\mathbf{x}_{t-1}|\mathbf{x}_t,\mathbf{c}^*)}\left[f^*(T(\hat{\Phi}))\right]\right], \tag{6}$$

where $T(\cdot)$ can be parametrized as a neural network fed with the output samples of the original DM $\Phi$ and the unlearned model $\hat{\Phi}$. This optimization problem can be treated as a min-max game between the generator $\hat{\Phi}$ and the discriminator $T$, where $T$ estimates the divergence between $p_\Phi$ and $p_{\hat{\Phi}}$, while the generator aims at minimizing the same divergence. Solving equation 6 leads to the minimization of $D_f(p_\Phi(\mathbf{x}_{t-1}|\mathbf{x}_t,\mathbf{c})||p_{\hat{\Phi}}(\mathbf{x}_{t-1}|\mathbf{x}_t,\mathbf{c}^*))$ for any $\mathbf{x}_t$.

The loss functions derived from closed-form expressions of $f$-divergences share a key benefit with the MSE: they only require solving a minimization problem. They are characterized by different gradients (see Section 4.1), which leads to improved convergence properties. In contrast, the general loss function expressed using the variational formulation requires solving a min-max problem. This approach, however, can be applied to any $f$-divergence, leading to a broader class of loss functions, for which we study the local convergence properties in Section 4.2.

We observe that, following Lu et al. (2024), instead of solving equation 3, it is possible to minimize the divergence between the probability distributions over a subset of the original generation trajectory (from $\mathbf{x}_t$ to $\mathbf{x}_0$, with $t < T$), and the $f$-divergence generalization still holds. This reasoning can be taken to the extreme by training the model with only $\mathbf{x}_0$, as proposed in Zhang et al. (2025).

# 4 THEORETICAL ANALYSIS

## 4.1 GRADIENT ANALYSIS

By analyzing the gradients of the loss functions obtained in closed-form (referred to as $\mathcal{J}_f^c$), it is possible to observe a significant difference between the gradient of the MSE, obtained for KL divergence, and those of the loss functions obtained from the squared Hellinger distance and $\chi^2$ divergence. For simplicity in the notation, let $\mathrm{MSE}(\Phi, \hat{\Phi})$ refer to $\mathrm{MSE}(\Phi(\mathbf{x}_i, \mathbf{c}, i), \hat{\Phi}(\mathbf{x}_i, \mathbf{c}^*, i))$, and let $\phi$ be the vector of parameters of $\hat{\Phi}$. Then, the gradients of the loss functions can be written as

$$\frac{\partial \mathcal{J}_f^c(\Phi, \hat{\Phi})}{\partial \phi} = \begin{cases} \nabla_\phi \mathrm{MSE}(\Phi, \hat{\Phi}) & \text{for KL} \\ e^{-\mathrm{MSE}(\Phi, \hat{\Phi})} \nabla_\phi \mathrm{MSE}(\Phi, \hat{\Phi}) & \text{for H}^2 \\ e^{\mathrm{MSE}(\Phi, \hat{\Phi})} \nabla_\phi \mathrm{MSE}(\Phi, \hat{\Phi}) & \text{for } \chi^2 \end{cases} . \tag{7}$$

Both the gradients of $H^2$ and $\chi^2$ are proportional to the gradient of KL. However, they are characterized by two opposite behaviors. When $\mathrm{MSE} \to \infty$, the gradients of $H^2$ and $\chi^2$ tend to 0 and $\infty$, respectively. When $\mathrm{MSE} \to 0$, the gradients of $H^2$ and $\chi^2$ both tend to the gradient of the MSE. In summary, we can conclude that $\left| \frac{\partial \mathcal{J}_{\mathrm{H}^2}(\Phi, \hat{\Phi})}{\partial \phi} \right| \leq \left| \frac{\partial \mathcal{J}_{\mathrm{KL}}(\Phi, \hat{\Phi})}{\partial \phi} \right| \leq \left| \frac{\partial \mathcal{J}_{\chi^2}(\Phi, \hat{\Phi})}{\partial \phi} \right|$, where the inequalities become equalities when the MSE is zero. More details can be found in Appendix B.1.2, where we also analyze the $\alpha$-divergence gradients (Fig. 12).

## 4.2 LOCAL CONVERGENCE ANALYSIS

In this section, we address the local convergence of the variational form of $f$-DMU to show that, under mild assumptions, the proposed algorithm is locally exponentially stable around equilibrium points. Similarly to how previous work studied the convergence of GANs (Nagarajan & Kolter, 2017; Mescheder et al., 2018) and energy-based models (Yu et al., 2020a), we use nonlinear dynamical systems theory (Hassan K, 1996). We consider the optimization of the model parameters of the min-max problem in equation 6 as a dynamical system. The system can be linearized (around the optimal convergence point) to study the local convergence properties. By evaluating the Jacobian matrix, we can conclude that if the Jacobian at an equilibrium point is a Hurwitz matrix, the system converges to the equilibrium (i.e., the equilibrium is locally exponentially stable). Let $\hat{\Phi}$ and $T$ be parametrized by the vectors of parameters $\phi$ and $\omega$, respectively. We rewrite the objective function in equation 6 as $\min_\phi \max_\omega \mathcal{J}_f(\phi, \omega)$. Following Nagarajan & Kolter (2017), we focus on the analysis of continuous time ordinary differential equations, which implies similar results for discrete time updates when the learning rate is sufficiently small. Following a gradient update rule, the dynamical system describing the models' update is given by

$$\begin{aligned} \dot{\phi} &= -\nabla_\phi \mathcal{J}_f(\phi, \omega) \\ \dot{\omega} &= \nabla_\omega \mathcal{J}_f(\phi, \omega) \end{aligned} . \tag{8}$$

### 4.2.1 MAIN CONVERGENCE RESULTS

Theorem 4.1 provides the Jacobian of the dynamical system describing the training of $\hat{\Phi}$ and $T$ at an equilibrium point. Theorem 4.2 provides the main theoretical result of this paper, stating the stability of the dynamical system in equation 8.

**Theorem 4.1.** *The Jacobian for the dynamical system defined in equation 8, at an equilibrium point* $(\phi^*, \omega^*)$ *is*

$$J = \begin{pmatrix} \mathbf{0} & -\boldsymbol{K}_{TP} \\ \boldsymbol{K}_{TP}^T & \boldsymbol{K}_{TT} \end{pmatrix}, \tag{9}$$

*where*

$$\boldsymbol{K}_{TP} \triangleq \mathbb{E}_{p_\Phi(\mathbf{x}_t|\mathbf{c})}\Big[\mathbb{E}_{p_\Phi(\mathbf{x}_{t-1}|\mathbf{x}_t,\mathbf{c})}\Big[-\nabla_\phi \log(p_\phi(\mathbf{x}_{t-1}|\mathbf{x}_t, \mathbf{c}^*))\Big(\nabla_\omega^T T_\omega\Big)\Big]\Big]\Big|_{(\phi^*,\omega^*)}, \tag{10}$$

$$\boldsymbol{K}_{TT} \triangleq \mathbb{E}_{p_\Phi(\mathbf{x}_t|\mathbf{c})}\Big[\mathbb{E}_{p_\Phi(\mathbf{x}_{t-1}|\mathbf{x}_t,\mathbf{c})}\Big[-(f^*)''(T_\omega)\nabla_\omega T_\omega \nabla_\omega^T T_\omega\Big]\Big]\Big|_{\omega^*}. \tag{11}$$

**Theorem 4.2.** *The dynamical system defined in equation 8 is locally exponentially stable with respect to an equilibrium point* $(\phi^*, \omega^*)$ *under Assumptions B.6, B.7, B.8. Let* $\lambda_m(\cdot)$ *and* $\lambda_M(\cdot)$ *be the smallest and largest eigenvalues of a given matrix, respectively. The rate of convergence of the system is governed by the eigenvalues of the Jacobian* $\mathbf{J}$ *which have a negative real part upper bounded as*

- *When* $Im(\lambda) = 0$, $\quad Re(\lambda) \leq -\frac{\lambda_m(-\boldsymbol{K}_{TT})\lambda_m(\boldsymbol{K}_{TP}\boldsymbol{K}_{TP}^T)}{\lambda_m(-\boldsymbol{K}_{TT})\lambda_M(-\boldsymbol{K}_{TT})+\lambda_m(\boldsymbol{K}_{TP}\boldsymbol{K}_{TP}^T)}.$

- *When* $Im(\lambda) \neq 0$, $\quad Re(\lambda) \leq -\frac{\lambda_m(-\boldsymbol{K}_{TT})}{2}.$

Besides the result on the local convergence of the dynamical system in 8, from Theorem 4.2 we can infer properties on the convergence speed of different $f$-divergences. This can be done by leveraging the knowledge about the upper bounds on the real part of the eigenvalues and about the fact that the bottleneck of the convergence speed is the largest eigenvalue. In Appendix B.2.4, we show that the $f$-divergences characterized by a larger $(f^*)''(T_\omega)|_{(\phi^*,\omega^*)}$ (i.e., a smaller $f''(1)$) are favored by a faster convergence property. This theoretical result is fundamental, as it provides guidelines on the choice of $f$-divergence. In particular, we show that $f''_{\mathrm{H}^2}(1) < f''_{\mathrm{KL}}(1) < f''_{\chi^2}(1)$, implying that $\mathrm{H}^2$ is characterized by a faster convergence near the equilibrium point.

Finally, it is possible to compare different $f$-divergences based on their mode-seeking (Li & Farnia, 2023) and saturation (Goodfellow et al., 2014; Nowozin et al., 2016a) properties. In the unlearning context, mode-seeking divergences tend to overfit by converging to a subset of modes of the distribution, thus reducing the diversity of generated images corresponding to the unlearned concepts, while mode-covering distributions may assign higher probability to the empty areas between the modes. Both JS and $\mathrm{H}^2$ are characterized by medium mode-seeking properties, while KL and Pearson $\chi^2$ divergences are mode-covering (the Pearson $\chi^2$ is more mode-covering than KL). The saturation problem also appears for $f$-divergence-based generative frameworks. It refers to the presence of weak gradients in the initial stages of training, when the density ratio is either very large or very small, presenting optimization difficulties. Although JS and $\mathrm{H}^2$ divergences are subject to strong saturation, in the considered unlearning scenario, such as in Xu et al. (2025), the saturation problem is alleviated because we fine-tune a generative model starting from an already trained DM. A summary of the differences between $f$-DMU losses can be found in Appendix B.3 (Tab. 5).

## 5 RESULTS

We present a comprehensive empirical evaluation of $f$-DMU. Our experiments are designed to: 1) validate our theoretical analysis of gradient amplitudes, 2) demonstrate the effectiveness of using alternative $f$-divergences compared to the standard MSE-based approach, and 3) assess the framework's scalability with multi-concepts erasure. In the following, we will refer to the squared Hellinger distance as $\mathrm{H}^2$ and to the Pearson $\chi^2$ as $\chi^2$.

### 5.1 IMPLEMENTATION DETAILS

The experiments presented in this paper are based on the publicly available `Stable Diffusion v1.4`, `Stable Diffusion v2.1`, and `Stable Diffusion XL` models. We measure how successfully the concept has been unlearned using CLIP Score (Hessel et al., 2021) (CS) and CLIP Accuracy (Radford et al., 2021) (CA). The lower the CS and CA for the target concept, the better the erasure has been. We also measure how the model maintains the knowledge of non-target concepts

by measuring again the CS and CA for unrelated concepts. The higher, the better, which means fewer collateral damages. Kernel Inception Distance (Bińkowski et al., 2018) (KID) is used to address the generative distribution change of the models. For non-target concepts, a smaller KID value implies good maintenance of the overall quality and coherence of the model. For target concepts, a smaller KID refers to a clean and realistic replacement, while a bigger KID might refer to a more destructive erasure, producing largely incoherent outputs.

## 5.2 GRADIENT ANALYSIS: EMPIRICAL VALIDATION OF THEORY

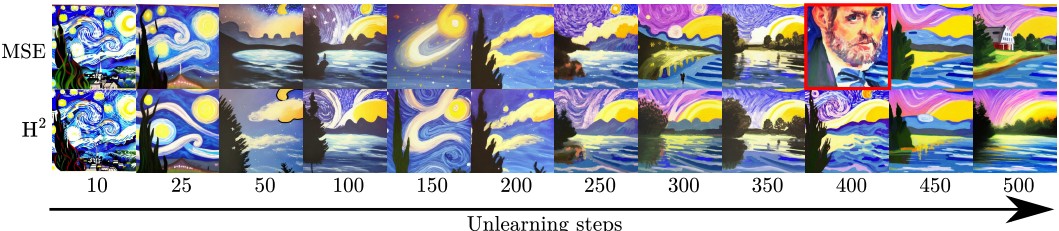

Figure 2: Instability during fine-tuning caused by large gradients. Top: MSE, bottom: $H^2$.

In Sec. 4.1, we have theoretically compared the gradient amplitudes between the different loss functions of $f$-DMU derived from the closed-form expression of divergences. We numerically demonstrate this behavior with an example in Fig. 3, where we visualize the (moving average of) average gradient amplitude for MSE (blue), $H^2$ (orange), and $\chi^2$ (green) during the fine-tuning of the unlearning procedure. While $H^2$ leads to bounded gradients significantly smaller than the other two losses, $\chi^2$ is characterized by meaningfully larger gradients. This behavior characterizes the losses properties and explains why, for instance, $H^2$ less probably leads to strange artifacts during fine-tuning. One example is reported in Fig. 2, where MSE leads the model to suddenly generate the drawing of a man's face at epoch $400$, while the more limited gradients that characterize $H^2$ lead to a behavior which is more robust to degenerate image gener-

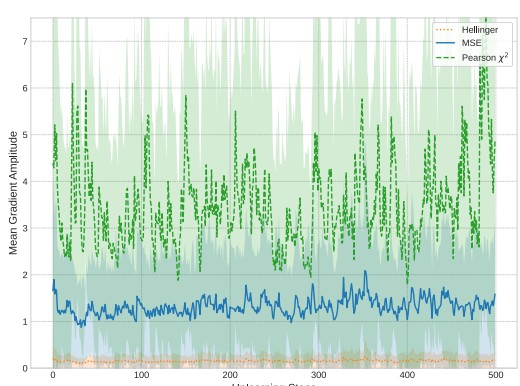

Figure 3: Average gradient amplitude of closed-form-based losses. The theoretical analysis of Sec. 4.1 holds empirically.

ation. Another consequence of $H^2$ bounded gradients is the better prior preservation with respect to MSE and Pearson $\chi^2$ (e.g., Figs. 1 and 4).

## 5.3 COMPARISON OF DIFFERENT $f$-DIVERGENCES

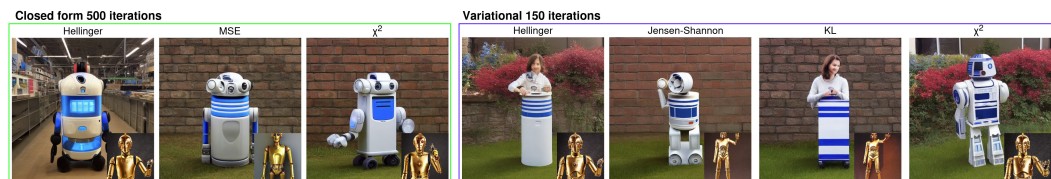

Figure 4: Comparison of unlearning the concept "R2D2" using different $f$-divergences with closed-form and variational loss. Knowledge preservation: the inset in the bottom right of each image shows an image of "C-3PO", which was not an erased concept. The unlearning is applied to SD 1.4.

**Closed-form solutions** The results of the closed-form losses reveal many differences in output quality. For object removal (see an example in Fig. 4), $H^2$ provides a replacement for the target

Table 1: Prior preservation capability of the H$^2$ method using a superclass anchor. SD 1.4 is fine-tuned for 2000 steps. We report CLIP Score (CS), CLIP Accuracy (CA), and Kernel Inception Distance (KID). For unerased concepts, higher CS/CA and lower KID indicate better performance.

| | Grumpy Cat | | | Snoopy | | | Wall-E | | | R2D2 | | | Van Gogh | | | Salvador Dali | | |
|---|---|---|---|---|---|---|---|---|---|---|---|---|---|---|---|---|---|---|
| *Erasing Grumpy Cat* | CS↓ | CA↓ | KID | CS↑ | CA↑ | KID↓ | CS↑ | CA↑ | KID↓ | CS↑ | CA↑ | KID↓ | CS↑ | CA↑ | KID↓ | CS↑ | CA↑ | KID↓ |
| Original Model | 0.74 | 1.00 | - | 0.73 | 1.00 | - | 0.73 | 1.00 | - | 0.75 | 1.00 | - | 0.80 | 1.00 | - | 0.68 | 0.80 | - |
| MSE | **0.56** | **0.53** | 0.194 | 0.62 | 0.70 | 0.070 | 0.65 | 0.73 | 0.082 | 0.70 | 0.90 | 0.122 | 0.72 | 0.97 | 0.049 | 0.67 | 0.77 | 0.031 |
| H$^2$ | **0.56** | 0.70 | 0.147 | **0.68** | **0.97** | **0.014** | **0.75** | **0.93** | **0.015** | **0.77** | **1.00** | **0.061** | **0.77** | **1.00** | **-0.009** | **0.69** | **0.87** | **-0.022** |
| *Erasing Salvador Dali* | CS↑ | CA↑ | KID↓ | CS↑ | CA↑ | KID↓ | CS↑ | CA↑ | KID↓ | CS↑ | CA↑ | KID↓ | CS↑ | CA↑ | KID↓ | CS↓ | CA↓ | KID |
| Original Model | 0.74 | 1.00 | - | 0.73 | 1.00 | - | 0.73 | 1.00 | - | 0.75 | 1.00 | - | 0.80 | 1.00 | - | 0.68 | 0.80 | - |
| MSE | 0.62 | 0.73 | 0.250 | 0.61 | 0.67 | 0.151 | 0.65 | 0.93 | 0.135 | 0.73 | **1.00** | 0.298 | 0.61 | 0.50 | 0.459 | **0.57** | **0.07** | 0.167 |
| H$^2$ | **0.73** | **1.00** | **0.005** | **0.73** | **1.00** | **0.004** | **0.78** | **1.00** | **0.048** | 0.76 | **1.00** | **0.040** | **0.62** | **0.53** | **0.361** | 0.58 | 0.10 | 0.102 |

concept that is more visually plausible than MSE, while better preserving non-erased concepts. Conversely, $\chi^2$ generally results in a stronger unlearning (for instance, fully removing any R2D2-related feature). These qualitative observations are confirmed by the quantitative results in Tab. 1 and the additional Tabs. 6, 9, and 10 reported in Appendix C. For the removal of artistic styles, analog observations can be made, as the H$^2$ loss is characterized by a slightly less strong erasure compared to MSE and $\chi^2$, but a significantly better prior preservation (Tab. 1). While MSE works well for the removal of concepts, it degrades the image quality more prominently than H$^2$. Furthermore, MSE methods generally have worse KID scores, indicating a greater influence on generative quality overall.

We test the different closed-form losses on SDXL Podell et al. (2023) with two goals: 1) demonstrate the applicability of our method to the SDXL architecture; 2) observe the gradient amplitude effect (theoretically analyzed in Sec. 4.1) on a significantly larger model. The results are reported in Fig. 5, where the impact of a higher gradient amplitude (increasing from H$^2$ to MSE and from MSE to $\chi^2$) visibly affects the generated images.

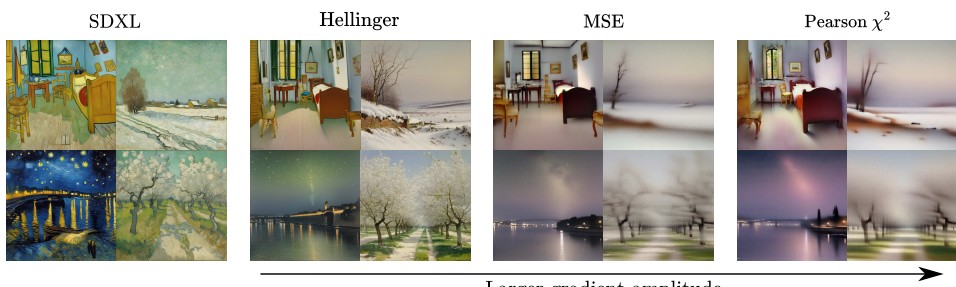

Larger gradient amplitude

Figure 5: Erasing Van Gogh on SDXL. Comparison between three closed-form $f$-DMU losses.

**Variational solutions** Qualitative results of the variational methods after 150 iterations are displayed in the right panel of Fig. 4. All methods manage to remove the concept from the model so that no generation of "R2D2" can take place. However, they more likely produce artifacts compared to closed-form methods. As demonstrated by the 150-step results recorded in Tab 8 in Appendix C, the variational approach is more "aggressive" than the closed-form as it causes lower CA in fewer iterations. However, this rapid semantic disruption comes at the cost of generative quality, resulting in a higher KID. This is clearly visible also in Fig. 7. The motivation is that when applying unlearning methods with typical small batch sizes, the divergence estimate is rough, yielding an imprecise minimization target. This results in a large noisy distribution shift that rapidly erodes the concept but leads to disruptive changes in the surrounding distribution. On the other hand, the closed-form losses guarantee that we are minimizing a specific divergence for any batch size, and having a small batch size only causes the model to see fewer examples of the concept to unlearn.

This highlights a key trade-off: non-variational (closed-form) methods are best for realistic concept replacement. Instead, variational methods are better suited for scenarios where the goal is rapid, aggressive semantic removal, and the realism of the output is a lower priority.

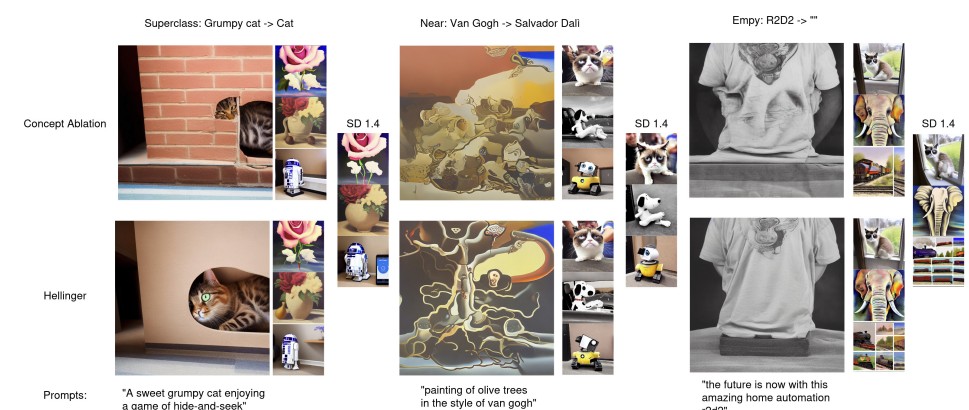

Figure 6: Different anchor concept substitutions. Base model: SD 1.4.

Table 2: Unlearning *Van Gogh* on SD 2.1. Comparison with state-of-the-art methods.

| Method | Erased (Van Gogh) CS ($\downarrow$) | CA ($\downarrow$) | J. Mann CS ($\uparrow$) | CA ($\uparrow$) | KID ($\downarrow$) | J. Vermeer CS ($\uparrow$) | CA ($\uparrow$) | KID ($\downarrow$) | S. Dali CS ($\uparrow$) | CA ($\uparrow$) | KID ($\downarrow$) | G. Rutkowski CS ($\uparrow$) | CA ($\uparrow$) | KID ($\downarrow$) | C. Monet CS ($\uparrow$) | CA ($\uparrow$) | KID ($\downarrow$) |
|---|---|---|---|---|---|---|---|---|---|---|---|---|---|---|---|---|---|
| CAbl Kumari et al. (2023) | 0.635 | 0.2 | 0.756 | 1.0 | -0.030 | 0.703 | 0.8 | -0.0330 | 0.639 | 0.6 | -0.014 | 0.553 | 0.5 | -0.042 | 0.688 | 1.0 | 0.008 |
| DoCo Wu et al. (2025) | 0.737 | 0.9 | 0.763 | 1.0 | -0.014 | 0.752 | 1.0 | -0.023 | 0.679 | 0.9 | -0.019 | 0.551 | 0.4 | -0.013 | 0.710 | 1.0 | 0.095 |
| UCE Gandikota et al. (2024) | 0.718 | 0.9 | 0.762 | 1.0 | 0.111 | 0.683 | 0.8 | 0.028 | 0.663 | 1.0 | 0.006 | 0.530 | 0.4 | 0.032 | 0.730 | 1.0 | 0.032 |
| Hellinger (Closed-Form) | 0.624 | 0.2 | 0.765 | 1.0 | -0.029 | 0.706 | 0.9 | -0.029 | 0.656 | 0.6 | 0.012 | 0.554 | 0.4 | -0.061 | 0.679 | 1.0 | 0.005 |
| $\chi^2$ (Closed-Form) | 0.628 | 0.1 | 0.776 | 1.0 | -0.034 | 0.688 | 0.9 | -0.041 | 0.639 | 0.5 | -0.011 | 0.552 | 0.5 | -0.045 | 0.707 | 0.9 | 0.008 |
| KL (Variational) | 0.755 | 1.0 | 0.749 | 1.0 | -0.039 | 0.730 | 0.9 | -0.030 | 0.646 | 0.7 | -0.030 | 0.550 | 0.5 | -0.040 | 0.690 | 1.0 | 0.087 |
| Hellinger (Variational) | 0.645 | 0.5 | 0.737 | 1.0 | -0.012 | 0.794 | 1.0 | 0.023 | 0.671 | 1.0 | -0.002 | 0.567 | 0.5 | -0.043 | 0.739 | 0.9 | 0.173 |
| Jensen-Shannon (Variational) | 0.738 | 0.8 | 0.752 | 1.0 | 0.010 | 0.708 | 0.9 | -0.032 | 0.665 | 0.8 | -0.017 | 0.560 | 0.4 | -0.021 | 0.686 | 0.9 | 0.109 |

**Comparison with state-of-the-art methods** We directly compare our $H^2$-based approach against the widely-used CAbl. Using both methods to replace a specific concept with a specific anchor concept, our results show that $H^2$ offers a superior trade-off between effective concept removal and the visual quality of the resulting generation. This is demonstrated qualitatively in Fig. 6 and is supported by the quantitative results in Tab. 6 in Appendix C. The improved visual quality aligns with our theoretical analysis of the gradient dynamics. The bounded gradients of $H^2$ (as validated in Fig. 3) prevent the large, destructive updates that can harm image coherence. A quantitative comparison between different $f$-DMU losses and state-of-the-art approaches is reported in Tab. 2, where we use the same fine-tuning and evaluation prompts for each method[1]. Tab. 2 highlights the excellent erasure/preservation trade-off of the $H^2$ closed-form loss compared to other closed-form losses, and the good convergence properties of the $H^2$ variational loss.

**Ring-A-Bell robustness** We test different $f$-DMU losses on the Ring-A-Bell (RAB) Tsai et al. (2023) framework, which generates ad-hoc prompts with the target of generating erased concepts. Fig. 7 studies the "Van Gogh" erasure with different $f$-divergences, and reports the CA computed on the images generated feeding the unlearned model with RAB prompts. For each $f$-divergence, we plot two images generated from standard prompts explicitly mentioning "Van Gogh" (to test the erasure performance), two images generated using the RAB prompts (to analyze the robustness), and four images generated from other artists prompts (to check the preservation of other concepts). While variational methods appear to be more sensitive to RAB prompts, leading to a high CA, $H^2$ closed-form loss leads to higher robustness to RAB prompts compared to the standard MSE.

## 5.4 SEQUENTIAL MULTI CONCEPT ERASURE

The observations previously reported for the $f$-DMU framework scale effectively to erasing multiple concepts, as demonstrated in Fig. 8 and Tab. 3, where we sequentially erase 10 artistic styles (50 steps for each artist). We compare unlearning with closed-form $H^2$ and standard MSE loss (used in CAbl). The $H^2$ loss shows superior prior preservation and better erasure performance. Further information and additional results can be found in Appendix C.2.

---

[1]UCE does not require full prompts as it requires the artists surnames, so we set "Van Gogh" as a concept to erase, and the surnames of all the other artists as concepts to preserve

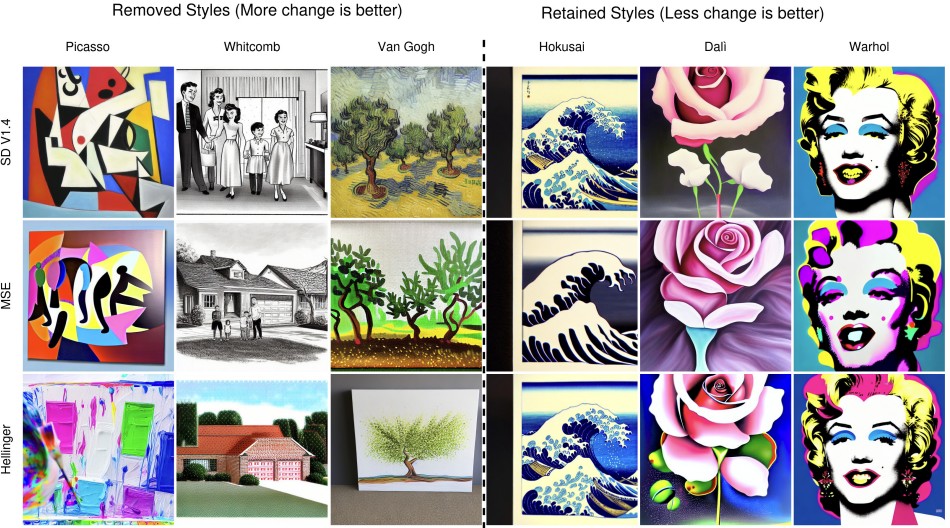

Figure 7: Robustness of erasure using different $f$-divergences. Each set of images includes generations from: original prompts (delimited by black lines), RAB prompts (delimited by red lines), other artists prompts (delimited by yellow lines). Lower CA implies a higher robustness to RAB prompts.

Figure 8: Sequential erasure of 10 artistic styles. First row: Original SD 1.4. Second row: Unlearning with standard MSE loss. Third row: Unlearning with our Hellinger-based closed-form loss.

Table 3: Evaluation of Sequential Unlearning on SD 1.4. Comparison of MSE and H$^2$ closed-form losses. Values are averaged over multiple runs. **Bold** indicates the best performance.

| Metric | Evaluation on Erased Artists (Concept Erasure) | | | | | | | | | Evaluation on Retained Artists (Prior Preservation) | | | | | | | | |
|---|---|---|---|---|---|---|---|---|---|---|---|---|---|---|---|---|---|---|
| | Monet | | | Picasso | | | Van Gogh | | | Dalí | | | Hokusai | | | Warhol | | |
| | Base | MSE | H$^2$ | Base | MSE | H$^2$ | Base | MSE | H$^2$ | Base | MSE | H$^2$ | Base | MSE | H$^2$ | Base | MSE | H$^2$ |
| **KID** *[Goal: ≈ 0 for Retained]* | — | 0.12 | 0.05 | — | 0.10 | 0.06 | — | 0.05 | 0.06 | — | 0.07 | **0.02** | — | 0.03 | **0.01** | — | **-0.01** | -0.02 |
| **CS** *[Goal: ↓ for Erased, ↑ for Retained]* | 0.74 | 0.60 | **0.59** | 0.72 | 0.63 | **0.60** | 0.80 | 0.61 | **0.61** | 0.72 | 0.62 | **0.62** | 0.77 | 0.72 | **0.74** | 0.72 | 0.67 | **0.70** |
| **CA** *[Goal: ↓ for Erased, ↑ for Retained]* | 1.00 | **0.45** | 0.54 | 0.90 | 0.40 | **0.35** | 1.00 | 0.55 | **0.40** | 0.80 | 0.50 | **0.60** | 1.00 | 0.80 | **0.95** | 0.90 | 0.85 | **1.00** |

# 6 CONCLUSIONS

In this paper, we propose a unified $f$-divergence-based framework for DM unlearning, which comprises two groups of loss functions: closed-form-based losses and variational form-based losses. We theoretically analyze the proposed loss functions and numerically evaluate them in different scenarios, demonstrating their relevance for aggressive unlearning or stronger concept preservation, depending on the $f$-divergence and on the closed-form or variational-based derivation. To support future research efforts, we summarize our theoretical findings in Table 5 of Appendix B.3, providing a guide for selecting the most appropriate $f$-divergence for the user goal.

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

# A    RELATED WORK

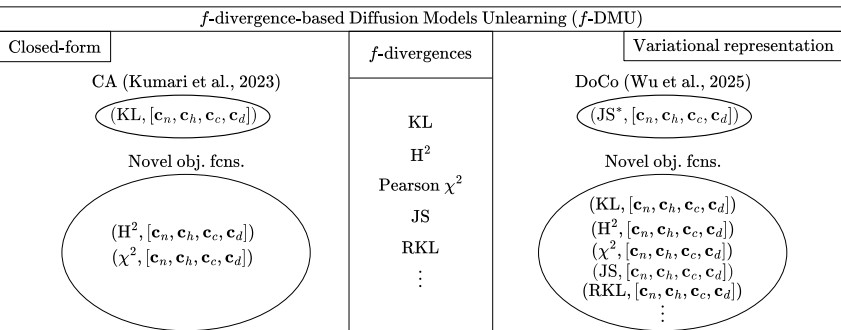

Figure 9: $f$-divergence-based Diffusion Model Unlearning Framework. Every pair $(f, \mathbf{c})$ indicates a different objective function based on a specific $f$-divergence and using a specific type of concept $\mathbf{c}$. A list of concepts $[\mathbf{c}_n, \mathbf{c}_h, \mathbf{c}_c, \mathbf{c}_d]$ indicates a specific $f$-divergence objective where it is possible to use, respectively, a null, hyper-class, close, and distant concept. JS$^*$ indicates that it is the variational formulation of the JS divergence with a change of variable (see Appendix B.1.3).

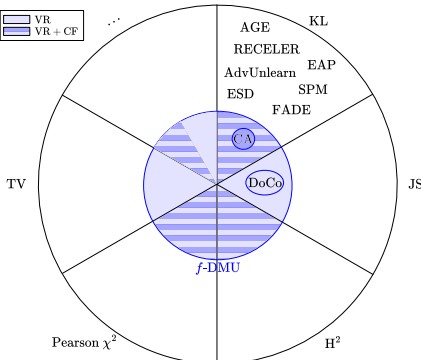

Figure 10: Divergence-based methods for diffusion models unlearning. Each sector represents an $f$-divergence, except from the sector described by the triple dots, indicating multiple other divergences. In each sector we can locate the unlearning methods that are based on minimizing such an $f$-divergence. The inner circle (in blue) represents $f$-DMU, which comprises any $f$-divergence. The light blue shade represents the existence of an objective function based on the variational representation of the $f$-divergence. Notably, it exists for any $f$-divergence. For some specific $f$-divergences, the closed-form expression of the loss exists, and this is highlighted by a stripe pattern of darker blue. For the $f$-divergences that are not explicitly represented in the circle (thus identified by the triple dots), $f$-DMU could either exist only in variational representation, or both as variational representation and closed-form (e.g., RKL, Jeffreys).

Machine unlearning studies the fundamental problem of removing specific knowledge learned from a learning algorithm (Cao & Yang, 2015; Bourtoule et al., 2021). In the deep learning era, this problem acquires significant meaning, due to the huge deep learning models that are trained on datasets scraped from the Internet.

In principle, machine unlearning can be performed following two different approaches: *exact* and *approximate* unlearning. Exact unlearning is considered the gold standard and consists in re-training the learning algorithm from scratch after having removed from the dataset the subset of data to be forgotten. Exact machine unlearning is often infeasible for two reasons: *i*) re-training a large model requires significant computational resources and time, *ii*) the training data could be deleted after their usage for training due to memory reasons, thus not being available for re-training. For these two reasons, researchers are focusing on approximate unlearning methods, which are faster, computationally lighter, and, for generative models, do not require the storage of the training dataset.

Initially, most of the machine unlearning literature was focused on discriminative tasks. For instance, many techniques were developed for classification problems, with the goal of unlearning a specific class or a subset of the dataset (Golatkar et al., 2020; Tarun et al., 2023; Chundawat et al., 2023a;b; Kurmanji et al., 2023; Fan et al., 2023; Foster et al., 2024; Choi & Na, 2025; Spartalis et al., 2025; Bagheri et al., 2025). Notably, $f$-SCRUB (Bagheri et al., 2025) is, to the best of our knowledge, the only existing $f$-divergence-based machine unlearning framework for classification. $f$-SCRUB generalizes SCRUB (Kurmanji et al., 2023) using the $f$-divergence. While for classification there is a large amount of literature related to machine unlearning, most classification techniques either cannot be applied for DMs or have shown poor performance on data generation tasks, thus requiring researchers to develop ad hoc unlearning methods for various generative models, including variational autoencoders and large language models (LLMs) (Heng & Soh, 2024; Wang et al., 2025; Liu et al., 2025). In particular, Wang et al. (2025) propose to maximize the $f$-divergence between template answers and forget data for LLM unlearning.

In this paper, we focus on the task of erasing concepts from DMs, which is a crucial task due to the widespread usage of DMs for image generation (Nichol et al., 2021; Rombach et al., 2022; Saharia et al., 2022), and due to their memorization capabilities (Somepalli et al., 2023a;b). In the following, we analyze the related work for DM unlearning, also referred to as "concept erasing" or "concept ablating". We categorize the existing techniques into two main groups: post-processing and fine-tuning techniques. Additionally, pre-processing techniques (Nichol et al., 2021) are also used. However, they require dataset curation followed by model training, while in this paper we assume to already have a trained model available.

Post-processing techniques target the elimination of unsafe generated images through the usage of filtering or inference guiding. SD (Rombach et al., 2022) adopts a NSFW filter that filters out all the generated images whose embeddings are close to the embeddings of 17 pre-chosen nudity concepts (Rando et al., 2022). Schramowski et al. (2023) present a method that is applied during inference and that pushes away the generation from unsafe contents. Another example of inference guidance-based approach is SAFREE (Yoon et al., 2025), which can be applied for both image and video generation. The main drawback of post-processing algorithms is that, when the user has access to the model, the post-processing operation can be removed to allow the model to generate what should have been ablated. Fine-tuning methods, instead, modify the weights of the model, thus being more robust than post-processing approaches when the user is granted access to the unlearned model. In this paper, we focus on fine-tuning methods.

Between fine-tuning methods, the distribution-based approaches, formulated as the minimization of a distance measure between probability distributions, can update any network parameter and work for different models, such as flow-based generative models (Zhang et al., 2025). This formulation is used as a ground idea in a wide variety of approaches. Erased Stable Diffusion (ESD) (Gandikota et al., 2023) fine-tunes the model by aligning the probability distributions of the model's output fed with a target and a null concept. To achieve that, the authors include in the loss function a classifier-free guidance-based term. Concept Ablation (CA) (Kumari et al., 2023) minimizes the MSE between the model's output when the model is fed with a target concept and an anchor hyper-class concept. CA's loss function corresponds to $f$-DMU when using the closed-form expression of the KL divergence[2]. This can also be observed from Figure 9, where we schematically depict $f$-DMU and highlight its relationship with existing approaches. Concept-SemiPermeable Membrane (SPM) (Lyu et al., 2024) proposes the usage of adapters that can be shared across different models and that rely on a novel Latent Anchoring (LA) fine-tuning strategy. The proposed loss comprises two terms: *i*) the first term coincides with the ESD loss and matches target and anchor concepts, *ii*) the second term (anchoring loss) minimizes the MSE to preserve the consistency of distant (in the CLIP space) concepts. Reliable concept erasing via Lightweight Erasers (RECELER) (Huang et al., 2024) introduces a new component into the neural network, the eraser, that acts on the cross-attention layers of the U-Net, and that is trained as in Gandikota et al. (2023), additionally including a concept-localized regularization term, to ensure the effective model performance on local concepts, while leveraging adversarial prompt learning to ensure robustness. Fine-grained attenuation for diffusion erasure (FADE) (Thakral et al., 2025) extracts the concepts that are semantically close to the concept to erase and is trained by minimizing an MSE loss that shifts the target concept to semantically close concepts and preserves the model generation on such a set.

---

[2]Here we are referring to the main loss term of both approaches, excluding considerations on the prior preservation terms.

Unified Concept Editing (UCE) (Gandikota et al., 2024) is a closed-form parameter editing method that builds upon Orgad et al. (2023) and Meng et al. (2022). UCE updates the cross-attention parameters in closed-form, ensuring a fast computational time. While the loss function is still formulated as the MSE, differently from the KL-based approaches previously presented, UCE minimizes the MSE directly at the cross-attention parameters level. Similarly, RECE (Gong et al., 2024) improves UCE by taking into consideration that the weights modification of UCE is not robust to adversarial prompts. Mass concept erasure (MACE) (Lu et al., 2024) extends UCE (Gandikota et al., 2024) by modifying the cross-attention layers and utilizing Low-Rank Adaptation (LoRA) to remove a large number of concepts. The major advantage of these methods performing the closed-form update of the weights is the computational complexity. However, the main drawback is that these methods can only be applied to attention weights and specific architectures.

Recently, some methods relying on a double optimization problem have been proposed. Most of them still rely on the KL-minimization idea previously analyzed. AdvUnlearn (Zhang et al., 2024) is an adversarial unlearning method which modifies the text encoder. Although the unlearned text encoder can be used as a plug-and-play robust module for various diffusion models, it needs a retain set. EAP (Bui et al., 2024) is trained with a loss comprising two terms that are being optimized over the model weights and the adversarial concept: the first term minimizes the distance from the concept to erase to the neutral concept, while the second term minimizes the distance from the adversarial concept in the original network to the adversarial concept in the new model. Adaptive Guided Erasure (AGE) (Bui et al., 2025), proposes a bi-level optimization framework that dynamically selects the optimal anchor concepts. Shirkavand et al. (2025) propose a min-max optimization problem that, in addition to minimizing the MSE between target and anchor concepts, includes a strategy for finding the optimal pruning method. The techniques previously discussed rely on the standard KL minimization approach. More concurrent with our work, Domain Correction DoCo (Wu et al., 2025) starts from CA (Kumari et al., 2023) and, in a membership inference attack fashion, replaces the MSE with the JS divergence. DoCo's loss function corresponds to $f$-DMU using the variational representation of the JS divergence when $\mathbf{c}^*$ is chosen as the hyper-class concept[3].

Figure 10 represents a schematic view of divergence-based methods for diffusion models unlearning. It shows how the KL divergence is largely adopted for diffusion models unlearning, while other divergence measures have been completely ignored. Our framework fills this gap, highlighting the potential of using different divergences. We hope that our work will encourage researchers to study techniques for unlearning that are not based on the KL divergence. In fact, we purposely leave empty white spaces in Figure 10 corresponding to methods that are different from $f$-DMU but leverage divergences that are different from the KL.

$f$**-Divergence** $f$-divergence-based methods have been effectively employed for the design of objective functions in a large number of applications, such as classification (Novello & Tonello, 2024), classification with label noise (Wei & Liu, 2021; Novello & Tonello, 2025), generation (Nowozin et al., 2016a), semi-supervised learning (Aminian et al., 2024), mutual information estimation (Letizia et al., 2024), and distillation (Roulet et al., 2025).
For diffusion models, the $f$-divergence has been used for distillation and alignment tasks, but not for solving unlearning problems. Tang (2024) uses an $f$-divergence-based entropy regularization term for fine-tuning diffusion models. Sun et al. (2025) propose an $f$-divergence-based framework for text-to-image models alignment. Xu et al. (2025) present an $f$-divergence-based approach for variational score distillation, to accelerate the diffusion models sampling process.
For machine unlearning, apart from Bagheri et al. (2025) for classification and Wang et al. (2025) for LLMs, general $f$-divergence frameworks have not been proposed. Specific $f$-divergences have only been applied as evaluation metrics. One example is the specific case of the JS divergence (Chundawat et al., 2023a; Choi et al., 2024; Bonato et al., 2024).

In conclusion, to the best of our knowledge, we propose the first diffusion model unlearning framework based on $f$-divergences.

---

[3]Here we are referring to the main loss term of both approaches, excluding considerations on the prior preservation terms.

# B PROOFS

## B.1 $f$-DIVERGENCE-BASED DIFFUSION MODELS UNLEARNING

To motivate equation 3, it is sufficient to notice that we are tackling the goal of minimizing a divergence measure between the model output distribution conditioned on the target concept $\mathbf{c}^*$ and the anchor concept $\mathbf{c}$, which can be rewritten as

$$\min_{\hat{\Phi}} \mathrm{d}(p_\Phi(\mathbf{x}_{t-1}|\mathbf{x}_t, \mathbf{c}), p_{\hat{\Phi}}(\mathbf{x}_{t-1}|\mathbf{x}_t, \mathbf{c}^*)), \tag{12}$$

which is satisfied $\forall t$ when the output distribution of the unlearned model $\hat{\Phi}$, conditioned on the concept to be unlearned $\mathbf{c}^*$ coincides with the distribution of the original model conditioned on the anchor concept $\mathbf{c}$.

In the seminal work of Kumari et al. (2023), equation 12 was obtained (using the KL divergence as divergence metric) by imposing as target goal the equivalence between the entire DM trajectories:

$$D_{KL}(p_\Phi(\mathbf{x}_{(0...T)}|\mathbf{c})||p_{\hat{\Phi}}(\mathbf{x}_{(0...T)}|\mathbf{c}^*)). \tag{13}$$

We report the steps to go from equation 13 to equation 12 for completeness, although most of them can also be found in Kumari et al. (2023). Then, we conclude by generalizing such a result using the $f$-divergence.

Following Kumari et al. (2023), equation 13 can be rewritten as

$$D_{KL}(p_\Phi(\mathbf{x}_{(0...T)}|\mathbf{c})||p_{\hat{\Phi}}(\mathbf{x}_{(0...T)}|\mathbf{c}^*))$$
$$= \mathbb{E}_{p(\mathbf{x}_{(0...T)}|\mathbf{c})} \left[ \log \frac{\prod_{t=1}^T p_\Phi(\mathbf{x}_{t-1}|\mathbf{x}_t, \mathbf{c})p_\Phi(\mathbf{x}_T)}{\prod_{t=1}^T p_{\hat{\Phi}}(\mathbf{x}_{t-1}|\mathbf{x}_t, \mathbf{c}^*)p_{\hat{\Phi}}(\mathbf{x}_T)} \right]$$
$$= \sum_{\hat{t}=1}^T \mathbb{E}_{p(\mathbf{x}_{(0...T)}|\mathbf{c})} \left[ \log \frac{p_\Phi(\mathbf{x}_{\hat{t}-1}|\mathbf{x}_{\hat{t}}, \mathbf{c})}{p_{\hat{\Phi}}(\mathbf{x}_{\hat{t}-1}|\mathbf{x}_{\hat{t}}, \mathbf{c}^*)} \right], \tag{14}$$

where the first step is obtained using the definition of KL divergence and the Markovianity of diffusion processes. In fact,

$$p(\mathbf{x}_{(0...T)}|\mathbf{c}) = p(\mathbf{x}_T|\mathbf{c}) \cdot p(\mathbf{x}_{T-1}|\mathbf{x}_T, \mathbf{c}) \cdot p(\mathbf{x}_{T-2}|\mathbf{x}_{T-1}, \mathbf{x}_T, \mathbf{c}) \cdot p(\mathbf{x}_{T-3}|\mathbf{x}_{T-2}, \mathbf{x}_{T-1}, \mathbf{x}_T, \mathbf{c}) \cdots \tag{15}$$
$$= p(\mathbf{x}_T|\mathbf{c}) \cdot p(\mathbf{x}_{T-1}|\mathbf{x}_T, \mathbf{c}) \cdot p(\mathbf{x}_{T-2}|\mathbf{x}_{T-1}, \mathbf{c}) \cdot p(\mathbf{x}_{T-3}|\mathbf{x}_{T-2}, \mathbf{c}) \cdots \tag{16}$$
$$= p(\mathbf{x}_T) \cdot p(\mathbf{x}_{T-1}|\mathbf{x}_T, \mathbf{c}) \cdot p(\mathbf{x}_{T-2}|\mathbf{x}_{T-1}, \mathbf{c}) \cdot p(\mathbf{x}_{T-3}|\mathbf{x}_{T-2}, \mathbf{c}) \cdots \tag{17}$$
$$= \prod_{t=1}^T p(\mathbf{x}_{t-1}|\mathbf{x}_t, \mathbf{c})p(\mathbf{x}_T), \tag{18}$$

where the second equality is a consequence of the Markovianity of DMs, and the third equality derives from the fact that $\mathbf{x}_T \sim \mathcal{N}(0, 1)$, which is independent from $\mathbf{c}$. From equation 14, it is possible to

study the generic term corresponding to a particular time step $\hat{t}$

$$\mathbb{E}_{p_\Phi(\mathbf{x}_0...\mathbf{x}_T)}\left[\log\frac{p_\Phi(\mathbf{x}_{\hat{t}-1}|\mathbf{x}_{\hat{t}},\mathbf{c})}{p_{\hat{\Phi}}(\mathbf{x}_{\hat{t}-1}|\mathbf{x}_{\hat{t}},\mathbf{c}^*)}\right]$$

$$=\int_{\mathbf{x}_{(0...T)}}\prod_{t=1}^{T}p_\Phi(\mathbf{x}_{t-1}|\mathbf{x}_t,\mathbf{c})p(\mathbf{x}_T)\log\frac{p_\Phi(\mathbf{x}_{\hat{t}-1}|\mathbf{x}_{\hat{t}},\mathbf{c})}{p_{\hat{\Phi}}(\mathbf{x}_{\hat{t}-1}|\mathbf{x}_{\hat{t}},\mathbf{c}^*)}d\mathbf{x}_{(0...T)}$$

$$=\int_{\mathbf{x}_{(\hat{t}...T)}}p_\Phi(\mathbf{x}_{(\hat{t}...T)}|\mathbf{c})\left[\int_{\mathbf{x}_{(0...\hat{t}-1)}}\prod_{t=1}^{\hat{t}}p_\Phi(\mathbf{x}_{t-1}|\mathbf{x}_t,\mathbf{c})\log\frac{p_\Phi(\mathbf{x}_{\hat{t}-1}|\mathbf{x}_{\hat{t}},\mathbf{c})}{p_{\hat{\Phi}}(\mathbf{x}_{\hat{t}-1}|\mathbf{x}_{\hat{t}},\mathbf{c}^*)}d\mathbf{x}_{(\hat{t}-1...0)}\right]d\mathbf{x}_{(\hat{t}...T)}$$

$$=\int_{\mathbf{x}_{(\hat{t}...T)}}p_\Phi(\mathbf{x}_{(\hat{t}...T)}|\mathbf{c})\left[\int_{\mathbf{x}_{(0...\hat{t}-1)}}\left(\prod_{t=1}^{\hat{t}-1}p_\Phi(\mathbf{x}_{t-1}|\mathbf{x}_t,\mathbf{c})\right)p_\Phi(\mathbf{x}_{\hat{t}-1}|\mathbf{x}_{\hat{t}},\mathbf{c})\right.$$

$$\left.\cdot\log\frac{p_\Phi(\mathbf{x}_{\hat{t}-1}|\mathbf{x}_{\hat{t}},\mathbf{c})}{p_{\hat{\Phi}}(\mathbf{x}_{\hat{t}-1}|\mathbf{x}_{\hat{t}},\mathbf{c}^*)}d\mathbf{x}_{(\hat{t}-1...0)}\right]d\mathbf{x}_{(\hat{t}...T)}$$

$$=\int_{\mathbf{x}_{\hat{t}}}p_\Phi(\mathbf{x}_{\hat{t}}|\mathbf{c})\left[\int_{\mathbf{x}_{\hat{t}-1}}p_\Phi(\mathbf{x}_{\hat{t}-1}|\mathbf{x}_{\hat{t}},\mathbf{c})\log\frac{p_\Phi(\mathbf{x}_{\hat{t}-1}|\mathbf{x}_{\hat{t}},\mathbf{c})}{p_{\hat{\Phi}}(\mathbf{x}_{\hat{t}-1}|\mathbf{x}_{\hat{t}},\mathbf{c}^*)}\right.$$

$$\left.\cdot\underbrace{\left[\int_{\mathbf{x}_{(0...\hat{t}-2)}}\prod_{t=1}^{\hat{t}-1}p_\Phi(\mathbf{x}_{t-1}|\mathbf{x}_t,\mathbf{c})d\mathbf{x}_{(\hat{t}-2...0)}\right]}_{=1}d\mathbf{x}_{\hat{t}-1}\right]d\mathbf{x}_{\hat{t}}$$

$$=\int_{\mathbf{x}_{\hat{t}}}p_\Phi(\mathbf{x}_{\hat{t}}|\mathbf{c})\left[\int_{\mathbf{x}_{\hat{t}-1}}p_\Phi(\mathbf{x}_{\hat{t}-1}|\mathbf{x}_{\hat{t}},\mathbf{c})\log\frac{p_\Phi(\mathbf{x}_{\hat{t}-1}|\mathbf{x}_{\hat{t}},\mathbf{c})}{p_{\hat{\Phi}}(\mathbf{x}_{\hat{t}-1}|\mathbf{x}_{\hat{t}},\mathbf{c}^*)}d\mathbf{x}_{\hat{t}-1}\right]d\mathbf{x}_{\hat{t}}$$

$$=\mathbb{E}_{p_\Phi(\mathbf{x}_t|\mathbf{c})}\left[D_{KL}\left(p_\Phi(\mathbf{x}_{t-1}|\mathbf{x}_t,\mathbf{c})||p_{\hat{\Phi}}(\mathbf{x}_{t-1}|\mathbf{x}_t,\mathbf{c}^*)\right)\right],\tag{19}$$

achieved using the fact that the integral over $p_\Phi(\mathbf{x}_{(\hat{t}...T)}|\mathbf{c})$ corresponds to the integral over $p_\Phi(\mathbf{x}_{\hat{t}}|\mathbf{c})$ because it contains all the information about the image versions from time step $T$ (pure noise) to the current step $\hat{t}$. Since $p_\Phi$ is a Gaussian distribution if the diffusion step sizes are small enough, in Kumari et al. (2023) the authors propose to use the objective function

$$\mathbb{E}_{p_\Phi(\mathbf{x}_t|\mathbf{c})}\left[\eta\left|\left|\Phi(\mathbf{x}_t,\mathbf{c},t)-\hat{\Phi}(\mathbf{x}_t,\mathbf{c}^*,t)\right|\right|_2^2\right],\tag{20}$$

which corresponds to the MSE between the two DMs distributions.

The minimization of the objective function in equation 19 w.r.t. $\hat{\Phi}$ can be generalized as the minimization of an $f$-divergence between the same probability distributions. Instead of solving equation 20, we solve

$$\mathbb{E}_{p_\Phi(\mathbf{x}_t|\mathbf{c})}\left[D_f\left(p_\Phi(\mathbf{x}_{t-1}|\mathbf{x}_t,\mathbf{c})||p_{\hat{\Phi}}(\mathbf{x}_{t-1}|\mathbf{x}_t,\mathbf{c}^*)\right)\right].\tag{21}$$

To minimize equation 21, we propose two different approaches:

- To compute the closed-form expression of a divergence between Gaussian distributions: in Sec. B.1.1, we provide the closed-form expressions of some $f$-divergences between Gaussian distributions. Each of them leads to a specific loss function.
- To use the variational representation of the $f$-divergence: in Sec. B.1.3 we report the variational representations of the $f$-divergences used in the experiments.

### B.1.1 CLOSED-FORM EXPRESSIONS OF THE OBJECTIVE FUNCTIONS FOR SPECIFIC $f$-DIVERGENCES

In the following, we report the main closed-form expressions of $f$-divergences between Gaussian distributions. For each of them, we will provide the information of whether it is a bounded or

unbounded $f$-divergence, as this will be related to the boundedness and unboundedness of the gradients. To check the boundedness of $f$-divergences, it is possible to use Theorem B.1. It is first necessary to define

$$f^{\star}(t) \triangleq t f\left(\frac{1}{t}\right), \tag{22}$$

for all $t > 0$. Furthermore, by definition

$$f^{\star}(0) = \lim_{u \to \infty} \frac{f(u)}{u}. \tag{23}$$

**Theorem B.1** (Range of values). *(see Vajda (1972)) Let P and Q be two probability distributions. Then, the range of an $f$-divergence is given by*

$$0 \le D_f(P||Q) \le f(0) + f^{\star}(0). \tag{24}$$

In the following, we use $p(x)$ and $q(x)$ as two multivariate normal distributions:

$$p(x) = \frac{1}{\sqrt{(2\pi)^d \det(\Sigma_P)}} \exp\left\{-\frac{1}{2}(x - \mu_P)^T \Sigma_P^{-1}(x - \mu_P)\right\} \tag{25}$$

$$q(x) = \frac{1}{\sqrt{(2\pi)^d \det(\Sigma_Q)}} \exp\left\{-\frac{1}{2}(x - \mu_Q)^T \Sigma_Q^{-1}(x - \mu_Q)\right\}. \tag{26}$$

**Kullback-Leibler divergence**: It is easy to show that the KL divergence between two multivariate Normal distributions reads as

$$D_{KL}(P||Q) = \frac{1}{2} \log\left(\frac{\det\Sigma_Q}{\det\Sigma_P}\right) - \frac{d}{2} + \frac{1}{2}\left[\text{Tr}\left(\Sigma_Q^{-1}\Sigma_P\right) + (\mu_P - \mu_Q)^T \Sigma_Q^{-1}(\mu_P - \mu_Q)\right]. \tag{27}$$

In the scalar case, equation 27 reduces to

$$D_{KL}(P||Q) = \frac{1}{2} \log \frac{\sigma_Q^2}{\sigma_P^2} - \frac{1}{2} + \frac{1}{2}\left(\frac{(\mu_P - \mu_Q)^2}{\sigma_Q^2} + \frac{\sigma_P^2}{\sigma_Q^2}\right). \tag{28}$$

It is well-known that the KL divergence is unbounded.

The training objective becomes the one proposed by Kumari et al. (2023), i.e., the MSE between the two diffusion models outputs.

**Jeffreys divergence**: From the definition of Jeffreys divergence, we obtain

$$\begin{aligned} D_J(P||Q) =& \frac{1}{2} \log\left(\frac{\det(\Sigma_Q)}{\det(\Sigma_P)}\right) + \frac{1}{2} \log\left(\frac{\det(\Sigma_P)}{\det(\Sigma_Q)}\right) - d \\ &+ \frac{1}{2}\Big[\text{Tr}\left(\Sigma_P^{-1}\Sigma_Q\right) + \text{Tr}\left(\Sigma_Q^{-1}\Sigma_P\right) \\ &+ (\mu_P - \mu_Q)^T \Sigma_Q^{-1}(\mu_P - \mu_Q) + (\mu_Q - \mu_P)^T \Sigma_P^{-1}(\mu_Q - \mu_P)\Big], \end{aligned} \tag{29}$$

which simplifies to

$$\begin{aligned} D_J(P||Q) =& -d + \frac{1}{2}\Big[\text{Tr}\left(\Sigma_Q^{-1}\Sigma_P\right) + \text{Tr}\left(\Sigma_P^{-1}\Sigma_Q\right) \\ &+ (\mu_P - \mu_Q)^T \left(\Sigma_Q^{-1} + \Sigma_P^{-1}\right)(\mu_P - \mu_Q)\Big]. \end{aligned} \tag{30}$$

In the scalar case, equation 30 becomes

$$D_J(P||Q) = -1 + \frac{1}{2}\left[\frac{\sigma_P^4 + \sigma_Q^4}{\sigma_Q^2 \sigma_P^2} + \frac{(\mu_P - \mu_Q)^2}{\sigma_P^2 \sigma_Q^2}(\sigma_P^2 + \sigma_Q^2)\right]. \tag{31}$$

Since the Jeffreys divergence is the sum of two KL divergences, it is unbounded.

Under the assumption $\sigma_P = \sigma_Q = \sigma$, the training objective becomes

$$\mathcal{J}_J(\mathbf{x}, \mathbf{c}, \mathbf{c}^*) = \mathbb{E}_{\epsilon, \mathbf{x}, \mathbf{c}^*, \mathbf{c}, t}\left[\frac{\omega_t}{\sigma^2}||\Phi(\mathbf{x}_t, \mathbf{c}, t) - \hat{\Phi}(\mathbf{x}_t, \mathbf{c}^*, t)||_2^2\right]. \tag{32}$$

**Squared Hellinger distance**: The squared Hellinger distance can be expressed as a function of the Bhattacharyya coefficient ($BC(P, Q)$) as $H^2(P, Q) = 1 - BC(P, Q)$. From Pardo (2018), the squared Hellinger distance between two multivariate Normal distributions reads as

$$H^2(P, Q) = 1 - \frac{\det(\Sigma_P)^{\frac{1}{4}}\det(\Sigma_Q)^{\frac{1}{4}}}{\det(\frac{\Sigma_P + \Sigma_Q}{2})^{\frac{1}{2}}}\exp\left\{-\frac{1}{8}(\mu_P - \mu_Q)^T\left(\frac{\Sigma_P + \Sigma_Q}{2}\right)^{-1}(\mu_P - \mu_Q)\right\}. \tag{33}$$

In the scalar case, equation 33 becomes

$$H^2(P, Q) = 1 - \frac{\sqrt{\sigma_P \sigma_Q}}{\sqrt{\frac{\sigma_P^2 + \sigma_Q^2}{2}}}\exp\left\{-\frac{1}{4}\frac{(\mu_P - \mu_Q)^2}{\sigma_P^2 + \sigma_Q^2}\right\}. \tag{34}$$

The H$^2$ is bounded. More precisely,

$$0 \leq H^2(P, Q) \leq 1, \tag{35}$$

which can be proven using the range of values theorem (Vajda, 1972). A quick intuition of the correctness of equation 35 can be derived by noticing that when $\mu_P = \mu_Q$ and $\sigma_P = \sigma_Q$, $H^2(P, Q) = 0$, while $H^2(P, Q) \to 1$ when $(\mu_P - \mu_Q)^2 \to \infty$.

Assuming $\sigma_P = \sigma_Q = \sigma$, the training objective becomes

$$\mathcal{J}_H(\mathbf{x}, \mathbf{c}, \mathbf{c}^*) = \mathbb{E}_{\epsilon, \mathbf{x}, \mathbf{c}^*, \mathbf{c}, t}\left[-\omega_t\exp\left\{-\frac{1}{8\sigma^2}||\Phi(\mathbf{x}_t, \mathbf{c}, t) - \hat{\Phi}(\mathbf{x}_t, \mathbf{c}^*, t)||_2^2\right\}\right]. \tag{36}$$

$\chi^2$ **divergence**: The $\chi^2$ divergence is defined as

$$\chi^2(P||Q) = \int_{\mathcal{X}}\frac{(p(x) - q(x))^2}{q(x)}dx = \int\frac{p(x)^2}{q(x)}dx - 1. \tag{37}$$

Substituting the expressions of the pdfs of Gaussian random variables, it becomes

$$\chi^2(P||Q) = \int\frac{\sigma_Q}{\sqrt{2\pi\sigma_P^2}}e^{-\frac{1}{2}\left(\frac{2(x - \mu_P)^2}{\sigma_P^2} - \frac{(x - \mu_Q)^2}{\sigma_Q^2}\right)}dx - 1 \tag{38}$$

$$= \frac{\sigma_Q^2}{\sigma_P\sqrt{2\sigma_Q^2 - \sigma_P^2}}\exp\left\{\frac{(\mu_P - \mu_Q)^2}{2\sigma_Q^2 - \sigma_P^2}\right\} - 1. \tag{39}$$

The expression in equation 39 holds only for $2\sigma_Q^2 > \sigma_P^2$.

The $\chi^2$ divergence is unbounded because $f^\star(0) = \infty$.

Under the assumption that $\sigma_P = \sigma_Q = \sigma$,

$$\mathcal{J}_{\chi^2}(\mathbf{x}, \mathbf{c}, \mathbf{c}^*) = \mathbb{E}_{\epsilon, \mathbf{x}, \mathbf{c}^*, \mathbf{c}, t}\left[\omega_t\exp\left\{\frac{1}{\sigma^2}||\Phi(\mathbf{x}_t, \mathbf{c}, t) - \hat{\Phi}(\mathbf{x}_t, \mathbf{c}^*, t)||_2^2\right\}\right]. \tag{40}$$

$\alpha$**-divergence**   We provide a general loss function derived from the closed-form expression of a subclass of $f$-divergences: the $\alpha$-divergence (Amari, 1985). The $\alpha$-divergence between two probability distributions $p(\mathbf{x})$ and $q(\mathbf{x})$ is defined as (Read & Cressie, 2012)

$$D_\alpha(p||q) = \frac{1}{\alpha(\alpha - 1)}\left(\int_{-\infty}^{\infty}p(\mathbf{x})^\alpha q(\mathbf{x})^{1-\alpha}d\mathbf{x} - 1\right), \tag{41}$$

where $\alpha \in \mathbb{R} \setminus \{0, 1\}$. The $\alpha$-divergence is a specific subclass of $f$-divergences obtained by imposing the generator function as

$$f(u) = \begin{cases} \frac{1}{\alpha(\alpha-1)}(u^\alpha - 1 - \alpha(u-1)) & \text{for } \alpha \notin \{0, 1\} \\ u \log u & \text{for } \alpha = 1 \\ -\log u & \text{for } \alpha = 0 \end{cases} . \tag{42}$$

As it is clear from equation 42, when $\alpha = 1$, we get the KL divergence, when $\alpha = 0$, we obtain the RKL divergence. Furthermore, when $\alpha = 1/2$, we attain the squared Hellinger distance, while when $\alpha = 2$, we get the Pearson $\chi^2$ divergence.

In general, by varying $\alpha$, we obtain different divergences with different properties. In fact, similarly to the discussion in Sec. 4, it is possible to identify which $\alpha$-divergences have mode covering or mode seeking properties.

The whole class of $\alpha$-divergences allows an analytical characterization when the probability density functions are Gaussian distributions $p \sim \mathcal{N}(\mu_p, \sigma_p^2)$ and $q \sim \mathcal{N}(\mu_q, \sigma_q^2)$ (Sourla et al., 2024):

$$D_\alpha(p||q) = \frac{1}{\alpha(1-\alpha)}(1 - H_\alpha(p, q)) \tag{43}$$

$$H_\alpha(p||q) = \frac{\sigma_p^{1-\alpha}\sigma_q^\alpha}{\sqrt{(1-\alpha)\sigma_p^2 + \alpha\sigma_q^2}} e^{-\frac{\alpha(1-\alpha)(\mu_p - \mu_q)^2}{2((1-\alpha)\sigma_p^2 + \alpha\sigma_q^2)}}. \tag{44}$$

In general, the $\alpha$-divergence between Gaussian distributions is real only when $\alpha \in [0, 1]$. For $\alpha > 1$, equation 44 is a real-valued function when $\sigma_p^2 < \frac{\alpha}{\alpha-1}\sigma_q^2$, while for $\alpha < 0$ equation 44 is a real-valued function when $\sigma_p^2 > \frac{\alpha}{\alpha-1}\sigma_q^2$.

In our scenario, $\sigma_p = \sigma_q = \sigma$, thus the above conditions are always verified, and the $\alpha$-divergence becomes

$$D_\alpha(p||q) = \frac{1}{\alpha(1-\alpha)}\left(1 - e^{-\frac{\alpha(1-\alpha)(\mu_p - \mu_q)^2}{2\sigma^2}}\right). \tag{45}$$

From equation 45, we can obtain the loss function corresponding to a general $\alpha$-divergence as

$$\mathcal{J}_\alpha(\mathbf{x}, \mathbf{c}, \mathbf{c}^*) = \mathbb{E}_{\epsilon, \mathbf{x}, \mathbf{c}^*, \mathbf{c}, t}\left[-\omega_t \exp\left\{-\frac{\alpha(1-\alpha)}{2\sigma^2}||\Phi(\mathbf{x}_t, \mathbf{c}, t) - \hat{\Phi}(\mathbf{x}_t, \mathbf{c}^*, t)||_2^2\right\}\right]. \tag{46}$$

### B.1.2 GRADIENT ANALYSIS

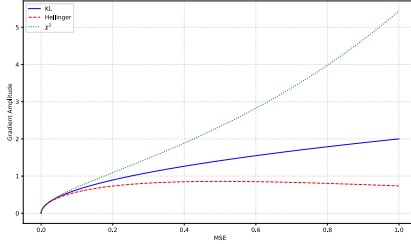

Figure 11: Gradients comparison between MSE, squared Hellinger distance, and Pearson divergence, as a function of the MSE. All the functions are plotted up to a multiplicative factor.

In this section, we analyze the gradients of the different closed-form loss functions presented in Section B.1.1.

**Squared Hellinger distance and Pearson divergence**   Let $\hat{\Phi}$ be a function of the parameters vector $\phi$, then

$$\sum_{i=1}^{n} \frac{\partial \mathcal{J}_f(\Phi, \hat{\Phi})}{\partial \phi} = \begin{cases} \sum_{i=1}^{n} \nabla_\phi \mathrm{MSE}(\Phi(\mathbf{x}_i, \mathbf{c}, i), \hat{\Phi}(\mathbf{x}_i, \mathbf{c}^*, i)) & \text{for KL} \\ \sum_{i=1}^{n} e^{-\mathrm{MSE}(\Phi(\mathbf{x}_i, \mathbf{c}, i), \hat{\Phi}(\mathbf{x}_i, \mathbf{c}^*, i))} \nabla_\phi MSE(\Phi(\mathbf{x}_i, \mathbf{c}, i), \hat{\Phi}(\mathbf{x}_i, \mathbf{c}^*, i)) & \text{for H}^2 \\ \sum_{i=1}^{n} e^{\mathrm{MSE}(\Phi(\mathbf{x}_i, \mathbf{c}, i), \hat{\Phi}(\mathbf{x}_i, \mathbf{c}^*, i))} \nabla_\phi MSE(\Phi(\mathbf{x}_i, \mathbf{c}, i), \hat{\Phi}(\mathbf{x}_i, \mathbf{c}^*, i)) & \text{for } \chi^2 \end{cases} \tag{47}$$

Since all the losses depend on the gradient of the MSE, it is easy to compare them:

- The gradients of $\mathrm{H}^2$ weight the MSE with a negative exponential. Thus, when $\mathrm{MSE} \to 0$, the gradients of $\mathrm{H}^2$ coincide with the gradients of MSE. Meanwhile, when $\mathrm{MSE} \to \infty$, the gradients of $\mathrm{H}^2$ tend to 0.

- For $\chi^2$ divergence, the gradients are weighted versions of the MSE, where the weight is a positive exponential. When $\mathrm{MSE} \to 0$, the gradients coincide with the MSE gradients, while when $\mathrm{MSE} \to \infty$, the gradients grow to $\infty$.

Therefore,

$$\left| \frac{\partial \mathcal{J}_{\mathrm{H}^2}(\Phi, \hat{\Phi})}{\partial \phi} \right| \leq \left| \frac{\partial \mathcal{J}_{\mathrm{KL}}(\Phi, \hat{\Phi})}{\partial \phi} \right| \leq \left| \frac{\partial \mathcal{J}_{\chi^2}(\Phi, \hat{\Phi})}{\partial \phi} \right|. \tag{48}$$

To provide a visual representation, we explicit the dependence of the MSE gradient on the difference between $\Phi$ and $\hat{\Phi}$:

$$\nabla_\phi \mathcal{J}_{\mathrm{KL}} = -2(\Phi - \hat{\Phi})\nabla_\phi \hat{\Phi} \tag{49}$$

$$\nabla_\phi \mathcal{J}_{\mathrm{H}^2} = -2e^{-(\Phi - \hat{\Phi})^2}(\Phi - \hat{\Phi})\nabla_\phi \hat{\Phi} \tag{50}$$

$$\nabla_\phi \mathcal{J}_{\chi^2} = -2e^{(\Phi - \hat{\Phi})^2}(\Phi - \hat{\Phi})\nabla_\phi \hat{\Phi} \tag{51}$$

and we depict in Figure 11 their behaviors, up to the multiplicative factor $\nabla_\phi \hat{\Phi}$.

The two opposite behaviors of $\mathrm{H}^2$ and $\chi^2$ divergences lead, in practice, to two very different behaviors during training. It appears that the $\chi^2$ divergence focuses more on those cases where the generated images with target and anchor concept are significantly different. This behavior is similar to the one obtained using the KL divergence, with the difference that the gradients of the $\chi^2$ divergence grow significantly faster. Meanwhile, the loss derived from the closed-form of the $\mathrm{H}^2$ divergence shows a peculiar behavior as it is less effected from outliers: it weights more the samples that have an intermediate MSE value.

Notably, both the KL and $\chi^2$ divergences are unbounded and have unbounded gradients. Differently, the $\mathrm{H}^2$ distance leads to a bounded loss function and bounded gradients.

**$\alpha$-divergence**   The gradient of the loss in equation 46 is

$$\frac{\partial \mathcal{J}_\alpha(\Phi, \hat{\Phi})}{\partial \phi} = \frac{\alpha(1-\alpha)}{2\sigma^2} e^{-\frac{\alpha(1-\alpha)}{2\sigma^2}\mathrm{MSE}(\Phi(\mathbf{x}_i, \mathbf{c}, i), \hat{\Phi}(\mathbf{x}_i, \mathbf{c}^*, i))} \nabla_\phi \mathrm{MSE}(\Phi(\mathbf{x}_i, \mathbf{c}, i), \hat{\Phi}(\mathbf{x}_i, \mathbf{c}^*, i)). \tag{52}$$

By rewriting the gradient of the MSE as previously done for KL, $\mathrm{H}^2$, and $\chi^2$, we report in Fig. 12 the gradient amplitude for different values of $\alpha$ and MSE: Fig. 12a shows the behavior when $\alpha \in (0, 1)$, while Fig. 12b represents $\alpha > 1$. The two plots show a distinctive behavior for each value of $\alpha$. While the squared Hellinger distance ($\alpha = 0.5$) has bounded gradients, it achieves the highest gradient amplitude compared to all the $\alpha$-divergences with $\alpha \in (0, 1)$, but also shows the steeper descent for increasing MSE values. This implies that, within the class of $\alpha$-divergences with $\alpha \in (0, 1)$, the squared Hellinger distance is more affected by cases in which the MSE is medium-low, while it is less affected by samples where the MSE is high. For $\alpha > 1$, although the Pearson $\chi^2$ divergence (corresponding to $\alpha = 2$) is characterized by an exponentially increasing gradient magnitude, we can observe that the more we increase $\alpha$, the more the corresponding divergence is characterized by steeper gradients.

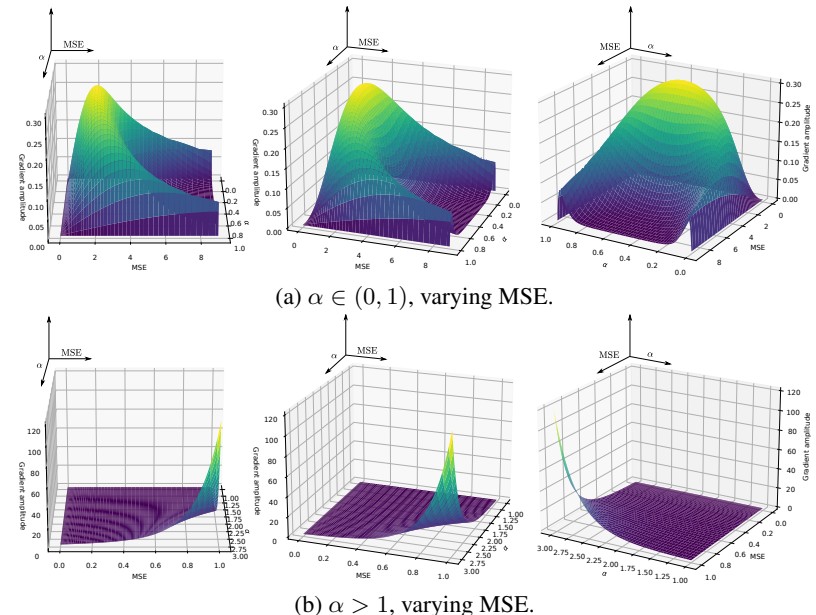

(a) $\alpha \in (0,1)$, varying MSE.

(b) $\alpha > 1$, varying MSE.

Figure 12: Gradient amplitude varying $\alpha$ and MSE. The plots in (a) also shows the gradient amplitude of the squared Hellinger distance, for $\alpha = 0.5^4$. Meanwhile, the plots in (b) also comprise the gradient amplitude of the Pearson $\chi^2$ divergence, for $\alpha = 2$. For (b), the MSE range is more limited as the gradient amplitude grows exponentially fast.

### B.1.3 VARIATIONAL REPRESENTATION

In this section, we report the variational-based version of $f$-DMU.

For simplicity in the notation, we will write $T(\Phi)$ to indicate $T(\Phi(\mathbf{x}_t, \mathbf{c}))$. The variational representation of the $f$-divergence would rewrite equation 3 as

$$\min_{\hat{\Phi}} \mathbb{E}_{p_\Phi(\mathbf{x}_t|\mathbf{c})} \left[ \sup_T \left\{ \mathbb{E}_{p_\Phi(\mathbf{x}_{t-1}|\mathbf{x}_t,\mathbf{c})} \left[ T(\Phi) \right] - \mathbb{E}_{p_{\hat{\Phi}}(\mathbf{x}_{t-1}|\mathbf{x}_t,\mathbf{c}^*)} \left[ f^*(T(\hat{\Phi})) \right] \right\} \right]. \tag{53}$$

Instead, we propose to solve the following problem:

$$\min_{\hat{\Phi}} D_f \left( p_\Phi(\mathbf{x}_{t-1}|\mathbf{x}_t,\mathbf{c}) || p_{\hat{\Phi}}(\mathbf{x}_{t-1}|\mathbf{x}_t,\mathbf{c}^*) \right) \quad \forall \mathbf{x}_t, \tag{54}$$

which corresponds to solving the following objective function

$$\min_{\hat{\Phi}} \max_T \mathbb{E}_{p_\Phi(\mathbf{x}_t|\mathbf{c})} \left[ \mathbb{E}_{p_\Phi(\mathbf{x}_{t-1}|\mathbf{x}_t,\mathbf{c})} \left[ T(\Phi) \right] - \mathbb{E}_{p_{\hat{\Phi}}(\mathbf{x}_{t-1}|\mathbf{x}_t,\mathbf{c}^*)} \left[ f^*(T(\hat{\Phi})) \right] \right]. \tag{55}$$

The result above can be shown by rewriting equation 54 as

$$\min_{\hat{\Phi}} D_f \left( p_\Phi(\mathbf{x}_{t-1}|\mathbf{x}_t,\mathbf{c}) p_\Phi(\mathbf{x}_t|\mathbf{c}) || p_{\hat{\Phi}}(\mathbf{x}_{t-1}|\mathbf{x}_{\hat{t}},\mathbf{c}^*) p_\Phi(\mathbf{x}_t|\mathbf{c}) \right) \quad \forall \mathbf{x}_t, \tag{56}$$

which is possible because $p_\Phi(\mathbf{x}_t|\mathbf{c})$ is Gaussian, thus ensuring to avoid degenerate cases in which $p_\Phi(\mathbf{x}_t|\mathbf{c})$ zeroes some areas of $p_\Phi(\mathbf{x}_{t-1}|\mathbf{x}_t,\mathbf{c})$ and $p_{\hat{\Phi}}(\mathbf{x}_{t-1}|\mathbf{x}_t,\mathbf{c}^*)$ that would be different, that would allow the divergence to be zero even for $p_\Phi(\mathbf{x}_{t-1}|\mathbf{x}_t,\mathbf{c}) \neq p_{\hat{\Phi}}(\mathbf{x}_{t-1}|\mathbf{x}_t,\mathbf{c}^*)$. Now, the variational representation of equation 56 comprises the term

$$\int_{\mathbf{x}_{t-1},\mathbf{x}_t} p_\Phi(\mathbf{x}_t|\mathbf{c}) p_\Phi(\mathbf{x}_{t-1}|\mathbf{x}_t,\mathbf{c}) T(\mathbf{x}_{t-1}) d\mathbf{x}_{t-1} d\mathbf{x}_t =$$

$$= \int_{\mathbf{x}_t} p_\Phi(\mathbf{x}_t|\mathbf{c}) \left[ \int_{\mathbf{x}_{t-1}} p_\Phi(\mathbf{x}_{t-1}|\mathbf{x}_t,\mathbf{c}) T(\mathbf{x}_{t-1}) d\mathbf{x}_{t-1} \right] d\mathbf{x}_t$$

$$= \mathbb{E}_{p_\Phi(\mathbf{x}_t|\mathbf{c})} \left[ \mathbb{E}_{p_\Phi(\mathbf{x}_{t-1}|\mathbf{x}_t,\mathbf{c})} \left[ T(\mathbf{x}_{t-1}) \right] \right]. \tag{57}$$

Table 4: $f$-divergences table.

| NAME | $f(u)$ | $f^*(t)$ | $g_f$ |
|------|--------|----------|-------|
| KULLBACK-LEIBLER | $u\log(u)$ | $e^{t-1}$ | $v$ |
| REVERSE KULLBACK-LEIBLER | $-\log(u)$ | $-1-\log(-t)$ | $-e^{-v}$ |
| SQUARED HELLINGER | $(\sqrt{u}-1)^2$ | $\frac{t}{1-t}$ | $1-e^{-v}$ |
| JENSEN-SHANNON | $u\log(u)-(u+1)\log\left(\frac{u+1}{2}\right)$ | $-\log(2-e^t)$ | $\log(2)-\log(1+e^{-v})$ |
| GAN | $u\log(u)-(u+1)\log(u+1)$ | $-\log(1-e^t)$ | $-\log(1+e^{-v})$ |
| $\chi^2$ | $(u-1)^2$ | $\frac{1}{4}t^2+t$ | $v$ |
| JEFFREYS | $(u-1)\log u$ | $W(e^{1-t})+\frac{1}{W(e^{1-t})}+t-2$ | $v$ |
| TOTAL VARIATION | $\frac{1}{2}|u-1|$ | $t$ | $\frac{1}{2}\tanh(v)$ |

We can obtain a similar result for $f^*(T(\cdot))$. Thus, for the linearity property of the expectation equation 55 follows.

Following the work in Nowozin et al. (2016b), we assume $T(x) = g_f(V_\omega(x))$, to respect the domain of the Fenchel conjugate function, i.e., $V : \mathcal{X} \to \mathbb{R}$ and $g_f : \mathbb{R} \to dom_{f^*}$. Thus, equation 55 becomes

$$\min_{\hat{\Phi}} \max_{V} \mathbb{E}_{p_\Phi(\mathbf{x}_t|\mathbf{c})}\left[\mathbb{E}_{p_\Phi(\mathbf{x}_{t-1}|\mathbf{x}_t,\mathbf{c})}\left[g_f(V(\Phi))\right] - \mathbb{E}_{p_{\hat{\Phi}}(\mathbf{x}_{t-1}|\mathbf{x}_t,\mathbf{c}^*)}\left[f^*(g_f(V(\Phi)))\right]\right]. \tag{58}$$

The generator functions and corresponding Fenchel conjugates of the $f$-divergences used in this paper are reported in Tab. 4. In addition, we report the corresponding output activation functions $g_f$.

**KL divergence** The loss function corresponding to the variational representation of the KL divergence reads as

$$\min_{\hat{\Phi}} \max_{V} \mathbb{E}_{p_\Phi(\mathbf{x}_t|\mathbf{c})}\left[\mathbb{E}_{p_\Phi(\mathbf{x}_{t-1}|\mathbf{x}_t,\mathbf{c})}\left[V(\Phi)\right] - \mathbb{E}_{p_{\hat{\Phi}}(\mathbf{x}_{t-1}|\mathbf{x}_t,\mathbf{c}^*)}\left[e^{V(\Phi)-1}\right]\right]. \tag{59}$$

**H$^2$ distance** The loss function corresponding to the variational representation of the squared Hellinger distance reads as

$$\min_{\hat{\Phi}} \max_{V} \mathbb{E}_{p_\Phi(\mathbf{x}_t|\mathbf{c})}\left[\mathbb{E}_{p_\Phi(\mathbf{x}_{t-1}|\mathbf{x}_t,\mathbf{c})}\left[1-e^{-V(\Phi)}\right] - \mathbb{E}_{p_{\hat{\Phi}}(\mathbf{x}_{t-1}|\mathbf{x}_t,\mathbf{c}^*)}\left[\frac{1-e^{-V(\Phi)}}{e^{-V(\Phi)}}\right]\right]. \tag{60}$$

**JS divergence** The loss function corresponding to the variational representation of the JS divergence reads as

$$\min_{\hat{\Phi}} \max_{V} \mathbb{E}_{p_\Phi(\mathbf{x}_t|\mathbf{c})}\left[\mathbb{E}_{p_\Phi(\mathbf{x}_{t-1}|\mathbf{x}_t,\mathbf{c})}\left[-\log\left(1+e^{-V(\Phi)}\right)\right] + \mathbb{E}_{p_{\hat{\Phi}}(\mathbf{x}_{t-1}|\mathbf{x}_t,\mathbf{c}^*)}\left[\log\left(\frac{e^{-V(\Phi)}}{1+e^{-V(\Phi)}}\right)\right]\right]. \tag{61}$$

Using the change of variable $T = -\log(1+e^{-v}) = \log(D)$, we obtain

$$\min_{\hat{\Phi}} \max_{V} \mathbb{E}_{p_\Phi(\mathbf{x}_t|\mathbf{c})}\left[\mathbb{E}_{p_\Phi(\mathbf{x}_{t-1}|\mathbf{x}_t,\mathbf{c})}\left[-\log\left(1+e^{-V(\Phi)}\right)\right] + \mathbb{E}_{p_{\hat{\Phi}}(\mathbf{x}_{t-1}|\mathbf{x}_t,\mathbf{c}^*)}\left[\log\left(\frac{e^{-V(\Phi)}}{1+e^{-V(\Phi)}}\right)\right]\right]. \tag{62}$$

Thus, we observe that DoCo's objective function corresponds to the variational representation of the JS divergence after a specific change of variable.

### B.2 LOCAL CONVERGENCE STUDY

In this section, we present in Sec. B.2.1 the necessary preliminaries to develop the local convergence study, and in Sec. B.2.3 the proofs of the local convergence of the algorithm proposed.

### B.2.1 PRELIMINARIES

Let us consider a system consisting of variables $\theta \in \Theta \subseteq \mathbb{R}^n$. Let its time derivative be defined by the vector field $v(\phi)$:

$$\dot{\phi} = v(\phi), \tag{63}$$

where $v : \mathbf{\Phi} \to \mathbb{R}^n$ is a locally Lipschitz mapping from a domain $\mathbf{\Phi}$ into $\mathbb{R}^n$.

Assume $\phi^*$ is an equilibrium point of the system in equation 63 (i.e., $v(\phi^*) = 0$). Let $\phi_t$ be the state of the system at time $t$.

**Definition B.2.** (see Hassan K (1996)) The equilibrium point $\phi^*$ for the system defined in equation 63 is

- stable if for each $\epsilon > 0$, there is $\delta = \delta(\epsilon) > 0$ such that

$$||\phi_0 - \phi^*|| < \delta \implies \forall t \geq 0, ||\phi_t - \phi^*|| < \epsilon \tag{64}$$

- unstable if not stable

- asymptotically stable if it is stable and $\delta > 0$ can be chosen such that

$$||\phi_0 - \phi^*|| < \delta \implies \lim_{t \to \infty} \phi_t = \phi^* \tag{65}$$

- exponentially stable if it is asymptotically stable and $\delta, k, \lambda > 0$ can be chosen such that:

$$||\phi_0 - \phi^*|| < \delta \implies ||\phi_t|| \leq k||\phi_0|| \exp\{(-\lambda t)\}. \tag{66}$$

The definition of stability can imply two different situations: 1) the systems converges to the equilibrium point, 2) the system is restricted to a ball of radius $\epsilon$ from the equilibrium. The asymptotic stability, instead, guarantees that the system reaches the equilibrium, when the initial condition belongs to a $\delta$ neighborhood of the equilibrium.

Theorem B.3 associates the stability of a non-linear system in the equilibrium point with its Jacobian, allowing to study the non-linear system by evaluating the eigenvalues of $\mathbf{J}$.

**Theorem B.3.** *(see Hassan K (1996)) Let $\phi^*$ be an equilibrium point for the non-linear system*

$$\dot{\phi} = v(\phi) \tag{67}$$

*where $v : \Phi \to \mathbb{R}^n$ is continuously differentiable and $\Phi$ is a neighborhood of $\phi^*$. Let $\mathbf{J}$ be the Jacobian of the system in equation 8 at the equilibrium point:*

$$\mathbf{J} = \left. \frac{\partial v(\phi)}{\partial \phi} \right|_{\phi=\phi^*}. \tag{68}$$

*Then, we have:*

- *The equilibrium point $\phi^*$ is asymptotically stable and exponentially stable if $\mathbf{J}$ is a Hurwitz matrix, i.e., $Re(\lambda) < 0$ for all eigenvalues $\lambda$ of $\mathbf{J}$.*

- *The equilibrium point $\phi^*$ is unstable if $Re(\lambda) > 0$ for one or more of the eigenvalues of $\mathbf{J}$.*

Theorem B.4 upper bounds the eigenvalues of a Jacobian as in equation 69. Theorem B.4 is used in Nagarajan & Kolter (2017) to prove the local convergence of GANs, and we use it in this paper to prove the local convergence of the proposed algorithm.

**Theorem B.4.** *(see Lemma G.2 in Nagarajan & Kolter (2017)) Let $\lambda_{min}$ and $\lambda_{max}$ denote the smallest and largest eigenvalues of a matrix, respectively. Suppose $\mathbf{J} \in \mathbb{R}^{(m+n)\times(m+n)}$ is of the following form:*

$$\mathbf{J} = \begin{bmatrix} \mathbf{0} & \mathbf{P} \\ -\mathbf{P}^T & -\mathbf{Q} \end{bmatrix} \tag{69}$$

*where $\mathbf{Q} \in \mathbb{R}^{n\times n}$ is a symmetric real positive definite matrix and $\mathbf{P}^T \in \mathbb{R}^{n\times m}$ is a full column rank matrix. Then, for every eigenvalue $\lambda$ of $\mathbf{J}$, $\mathrm{Re}(\lambda) < 0$. More precisely, we have:*

- *When $\mathrm{Im}(\lambda) = 0$,*

$$\mathrm{Re}(\lambda) \leq -\frac{\lambda_{min}(\mathbf{Q})\lambda_{min}(\mathbf{PP}^T)}{\lambda_{min}(\mathbf{Q})\lambda_{max}(\mathbf{Q}) + \lambda_{min}(\mathbf{PP}^T)}, \tag{70}$$

- *When* $\text{Im}(\lambda) \neq 0$,

$$\text{Re}(\lambda) \leq -\frac{\lambda_{min}(\mathbf{Q})}{2}. \tag{71}$$

Lemma B.5 reports a known relationship between the generator function of an $f$-divergence and its Fenchel conjugate. This relationship is useful in our Jacobian study.

**Lemma B.5.** *Let* $f : \mathbb{R}_+ \longrightarrow \mathbb{R}$ *be the generator function of any* $f$-*divergence. Let* $f^*$ *be the Fenchel conjugate of* $f$. *Then,*

$$(f^*)'(t) = (f')^{-1}(t), \tag{72}$$

*where* $f'$ *indicates the first derivative of the generator function.*

*Proof.* Since $f(\cdot)$ is a convex function, $\hat{u}$ achieving the supremum is

$$\hat{u} = (f')^{-1}(t). \tag{73}$$

Then, substituting equation 73 in the definition of the Fenchel conjugate, we obtain

$$f^*(t) = (f')^{-1}(t)t - f((f')^{-1}(t)). \tag{74}$$

The first derivative reads as

$$(f^*)'(t) = ((f')^{-1})'(t)t + (f')^{-1}(t) - \underbrace{f'((f')^{-1}(t))}_{=t}((f')^{-1})'(t). \tag{75}$$

The first and third terms cancel out, leading to equation 72. $\qquad\square$

### B.2.2 NOTATION AND SETUP

Let us define

$$\min_{\phi} \max_{\omega} \mathcal{J}(\phi, \omega) = \min_{\phi} \max_{\omega} \mathbb{E}_{p_\Phi(\mathbf{x}_{\hat{t}}|\mathbf{c})} \Big[ \mathbb{E}_{p_\Phi(\mathbf{x}_{\hat{t}-1}|\mathbf{x}_{\hat{t}},\mathbf{c})} [T_\omega(\Phi)] - \mathbb{E}_{p_\phi(\mathbf{x}_{\hat{t}-1}|\mathbf{x}_{\hat{t}},\mathbf{c}^*)} [f^*(T_\omega(\Phi))] \Big]. \tag{76}$$

Let us define the pair $(\phi^*, \omega^*)$ as the equilibrium point achieved when solving equation 76.

**Assumption B.6.** $f^*(\cdot)$ is a strictly convex function.

Assumption B.6 excludes from the set of $f$-divergences few specific cases that we do not use in this paper. For instance, the Total Variation distance does not satisfy Assumption B.6, as $f^*(t) = t$. Meanwhile, the KL divergence, Reverse KL (RKL) divergence, $\chi^2$ divergence, squared Hellinger distance, Jeffreys divergence, and Jensen-Shannon divergence satisfy Assumption B.6.

**Assumption B.7.** $\exists \phi^*, \omega^*$ such that $\forall \mathbf{x} \in \text{supp}(p), p_{\phi^*}(\mathbf{x}) = p_\Phi(\mathbf{x})$ and $T_{\omega^*} = f'(p_\Phi/p_\phi)$.

Assumption B.7 is achieved when we reach the optimal convergence, and implies that, at the equilibrium,

$$T_{\omega^*} = f'(p_\Phi/p_{\phi^*}) = f'(1), \tag{77}$$

thus

$$(f^*)'(T_{\omega^*}) = 1, \tag{78}$$

because $(f^*)'(t) = (f')^{-1}(t)$ (from Lemma B.5).

**Assumption B.8.** $\mathbb{E}_{p_\Phi(\mathbf{x}_{\hat{t}}|\mathbf{c})} \Big[ \mathbb{E}_{p_\Phi(\mathbf{x}_{\hat{t}-1}|\mathbf{x}_{\hat{t}},\mathbf{c})} \Big[ -\nabla_\phi \log\big(p_\phi(\mathbf{x}_{\hat{t}-1}|\mathbf{x}_{\hat{t}},\mathbf{c}^*)\big)\big(\nabla_\omega^T T_\omega\big) \Big] \Big] \Big|_{(\phi^*,\omega^*)}$ and

$\mathbb{E}_{p_\Phi(\mathbf{x}_{\hat{t}}|\mathbf{c})} \Big[ \mathbb{E}_{p_\Phi(\mathbf{x}_{\hat{t}-1}|\mathbf{x}_{\hat{t}},\mathbf{c})} \big(\nabla_\omega T_\omega \nabla_\omega^T T_\omega\big) \Big] \Big|_{\omega^*}$ are full row rank.

Assumption B.8 is similar to the assumptions used in Nagarajan & Kolter (2017); Yu et al. (2020a). In particular,

$$\mathbb{E}_{p_\Phi(\mathbf{x}_{\hat{t}}|\mathbf{c})} \Big[ \mathbb{E}_{p_\Phi(\mathbf{x}_{\hat{t}-1}|\mathbf{x}_{\hat{t}},\mathbf{c})} \Big[ -\nabla_\phi \log\big(p_\phi(\mathbf{x}_{\hat{t}-1}|\mathbf{x}_{\hat{t}},\mathbf{c}^*)\big)\big(\nabla_\omega^T T_\omega\big) \Big] \Big] \Big|_{(\phi^*,\omega^*)}$$

$$= \mathbb{E}_{p_\Phi(\mathbf{x}_{\hat{t}}|\mathbf{c})} \Big[ -\int_{\mathbf{x}_{\hat{t}-1}} \nabla_\phi p_\phi(\mathbf{x}_{\hat{t}-1}|\mathbf{x}_{\hat{t}},\mathbf{c}^*)\big(\nabla_\omega^T T_\omega\big) d\mathbf{x}_{\hat{t}-1} \Big] \Big|_{(\phi^*,\omega^*)}, \tag{79}$$

where $\nabla_\phi p_\phi(\mathbf{x}_{\hat{t}-1}|\mathbf{x}_{\hat{t}},\mathbf{c}^*) \neq 0$, as $p_\phi$ is the probability density function of a Gaussian random variable.

### B.2.3 Proofs of Local Convergence

**Theorem 4.1.** *The Jacobian for the dynamical system defined in equation 8, at an equilibrium point* $(\phi^*, \omega^*)$ *is*

$$J = \begin{pmatrix} \boldsymbol{0} & -\boldsymbol{K}_{TP} \\ \boldsymbol{K}_{TP}^T & \boldsymbol{K}_{TT} \end{pmatrix}, \tag{80}$$

*where*

$$\boldsymbol{K}_{TP} \triangleq \mathbb{E}_{p_\Phi(\mathbf{x}_{\hat{t}}|\mathbf{c})} \left[ \mathbb{E}_{p_\Phi(\mathbf{x}_{\hat{t}-1}|\mathbf{x}_{\hat{t}},\mathbf{c})} \left[ -\nabla_\phi \log\big(p_\phi(\mathbf{x}_{\hat{t}-1}|\mathbf{x}_{\hat{t}},\mathbf{c}^*)\big)\big(\nabla_\omega^T T_\omega\big) \right] \right] \Bigg|_{(\phi^*,\omega^*)} \tag{81}$$

$$\boldsymbol{K}_{TT} \triangleq \mathbb{E}_{p_\Phi(\mathbf{x}_{\hat{t}}|\mathbf{c})} \left[ \mathbb{E}_{p_\Phi(\mathbf{x}_{\hat{t}-1}|\mathbf{x}_{\hat{t}},\mathbf{c})} \left[ -(f^*)''(T_\omega)\nabla_\omega T_\omega \nabla_\omega^T T_\omega \right] \right] \Bigg|_{\omega^*} \tag{82}$$

*Proof.* For the proof, we rewrite the objective as

$$\min_\phi \sup_\omega \mathbb{E}_{p_\Phi(\mathbf{x}_{\hat{t}}|\mathbf{c})} \Big[ \mathbb{E}_{p_\Phi(\mathbf{x}_{\hat{t}-1}|\mathbf{x}_{\hat{t}},\mathbf{c})} \left[ T_\omega(\mathbf{x}_{\hat{t}},\mathbf{c}) \right] - \mathbb{E}_{p_\phi(\mathbf{x}_{\hat{t}-1}|\mathbf{x}_{\hat{t}},\mathbf{c}^*)} \left[ f^*(T_\omega(\mathbf{x}_{\hat{t}},\mathbf{c})) - f^*(f'(1)) \right] \Big]$$
$$- f^*(f'(1)), \tag{83}$$

where $f^*(f'(1))$ is a constant. The gradient of $\mathcal{J}(\phi,\omega)$ w.r.t. $\phi$ is

$$\nabla_\phi \mathcal{J}(\phi,\omega) = \mathbb{E}_{p_\Phi(\mathbf{x}_{\hat{t}}|\mathbf{c})} \left[ -\int_{\mathbf{x}_{\hat{t}-1}} \nabla_\phi p_\phi(\mathbf{x}_{\hat{t}-1}|\mathbf{x}_{\hat{t}},\mathbf{c}^*) \left( f^*(T_\omega(\mathbf{x}_{\hat{t}},\mathbf{c})) - f^*(f'(1)) \right) d\mathbf{x}_{\hat{t}-1} \right]. \tag{84}$$

The gradient of $\mathcal{J}(\phi,\omega)$ w.r.t. $\omega$ is

$$\nabla_\omega \mathcal{J}(\phi,\omega) = \mathbb{E}_{p_\Phi(\mathbf{x}_{\hat{t}}|\mathbf{c})} \left[ \int_{\mathbf{x}_{\hat{t}-1}} p_\Phi(\mathbf{x}_{\hat{t}-1}|\mathbf{x}_{\hat{t}},\mathbf{c})\nabla_\omega T_\omega - p_\phi(\mathbf{x}_{\hat{t}-1}|\mathbf{x}_{\hat{t}},\mathbf{c}^*)(f^*)'(T_\omega)\nabla_\omega T_\omega d\mathbf{x}_{\hat{t}-1} \right]. \tag{85}$$

Then,

$$\nabla_\phi^2 \mathcal{J}(\phi,\omega) = \mathbb{E}_{p_\Phi(\mathbf{x}_{\hat{t}}|\mathbf{c})} \left[ -\int_{\mathbf{x}_{\hat{t}-1}} \nabla_\phi^2 p_\phi(\mathbf{x}_{\hat{t}-1}|\mathbf{x}_{\hat{t}},\mathbf{c}^*) \left( f^*(T_\omega(\mathbf{x}_{\hat{t}},\mathbf{c})) - f^*(f'(1)) \right) d\mathbf{x}_{\hat{t}-1} \right]. \tag{86}$$

$$\nabla_\omega \nabla_\phi \mathcal{J}(\phi,\omega) = \mathbb{E}_{p_\Phi(\mathbf{x}_{\hat{t}}|\mathbf{c})} \left[ -\int_{\mathbf{x}_{\hat{t}-1}} \nabla_\phi p_\phi(\mathbf{x}_{\hat{t}-1}|\mathbf{x}_{\hat{t}},\mathbf{c}^*) \left( (f^*)'(T_\omega(\mathbf{x}_{\hat{t}},\mathbf{c}))\nabla_\omega^T T_\omega \right) d\mathbf{x}_{\hat{t}-1} \right]. \tag{87}$$

$$\nabla_\omega^2 \mathcal{J}(\phi,\omega) = \mathbb{E}_{p_\Phi(\mathbf{x}_{\hat{t}}|\mathbf{c})} \Bigg[ \int_{\mathbf{x}_{\hat{t}-1}} p_\Phi(\mathbf{x}_{\hat{t}-1}|\mathbf{x}_{\hat{t}},\mathbf{c})\nabla_\omega^2 T_\omega - p_\phi(\mathbf{x}_{\hat{t}-1}|\mathbf{x}_{\hat{t}},\mathbf{c}^*)\cdot \tag{88}$$

$$\left( (f^*)''(T_\omega)\nabla_\omega T_\omega \nabla_\omega^T T_\omega + (f^*)'(T_\omega)\nabla_\omega^2 T_\omega \right) d\mathbf{x}_{\hat{t}-1} \Bigg] \tag{89}$$

With Assumption B.7, we obtain

$$\nabla_\phi^2 \mathcal{J}(\phi,\omega) \Bigg|_{(\phi^*,\omega^*)} = \mathbb{E}_{p_\Phi(\mathbf{x}_{\hat{t}}|\mathbf{c})} \left[ -\int_{\mathbf{x}_{\hat{t}-1}} \nabla_\phi^2 p_\phi(\mathbf{x}_{\hat{t}-1}|\mathbf{x}_{\hat{t}},\mathbf{c}^*)\Big( f^*(T_{\omega^*}(\mathbf{x}_{\hat{t}},\mathbf{c})) \right.$$
$$\left. - f^*(f'(1))\Big) d\mathbf{x}_{\hat{t}-1} \right] \tag{90}$$

$$= \mathbb{E}_{p_\Phi(\mathbf{x}_{\hat{t}}|\mathbf{c})} \left[ -\int_{\mathbf{x}_{\hat{t}-1}} \nabla_\phi^2 p_\phi(\mathbf{x}_{\hat{t}-1}|\mathbf{x}_{\hat{t}},\mathbf{c}^*)\Big( f^*(f'(1)) - f^*(f'(1))\Big) d\mathbf{x}_{\hat{t}-1} \right] \tag{91}$$

$$= \boldsymbol{0}. \tag{92}$$

Under Assumption B.7, we have

$$\nabla_\omega \nabla_\phi \mathcal{J}(\phi, \omega)\bigg|_{(\phi^*, \omega^*)} = \mathbb{E}_{p_\Phi(\mathbf{x}_{\hat{t}}|\mathbf{c})}\left[-\int_{\mathbf{x}_{\hat{t}-1}} \nabla_\phi p_\phi(\mathbf{x}_{\hat{t}-1}|\mathbf{x}_{\hat{t}}, \mathbf{c}^*)\Big((f^*)'(T_{\omega^*}(\mathbf{x}_{\hat{t}}, \mathbf{c}))\nabla_\omega^T T_\omega\Big)d\mathbf{x}_{\hat{t}-1}\right]$$

$$\text{(93)}$$

$$= \mathbb{E}_{p_\Phi(\mathbf{x}_{\hat{t}}|\mathbf{c})}\left[-\int_{\mathbf{x}_{\hat{t}-1}} \nabla_\phi p_\phi(\mathbf{x}_{\hat{t}-1}|\mathbf{x}_{\hat{t}}, \mathbf{c}^*)\Big(\nabla_\omega^T T_\omega\Big)d\mathbf{x}_{\hat{t}-1}\right] \quad \text{(94)}$$

$$= \mathbb{E}_{p_\Phi(\mathbf{x}_{\hat{t}}|\mathbf{c})}\left[-\int_{\mathbf{x}_{\hat{t}-1}} p_\Phi(\mathbf{x}_{\hat{t}-1}|\mathbf{x}_{\hat{t}}, \mathbf{c})\nabla_\phi \log\big(p_\phi(\mathbf{x}_{\hat{t}-1}|\mathbf{x}_{\hat{t}}, \mathbf{c}^*)\big)\right.$$

$$\left. \cdot \Big(\nabla_\omega^T T_\omega\Big)d\mathbf{x}_{\hat{t}-1}\right] \quad \text{(95)}$$

$$= \mathbb{E}_{p_\Phi(\mathbf{x}_{\hat{t}}|\mathbf{c})}\left[\mathbb{E}_{p_\Phi(\mathbf{x}_{\hat{t}-1}|\mathbf{x}_{\hat{t}}, \mathbf{c})}\left[-\nabla_\phi \log\big(p_\phi(\mathbf{x}_{\hat{t}-1}|\mathbf{x}_{\hat{t}}, \mathbf{c}^*)\big)\Big(\nabla_\omega^T T_\omega\Big)\right]\right]\bigg|_{(\phi^*, \omega^*)}$$

$$\text{(96)}$$

$$\triangleq K_{TP} \quad \text{(97)}$$

With similar steps, we obtain

$$\nabla_\phi \nabla_\omega \mathcal{J}(\phi, \omega)\bigg|_{(\phi^*, \omega^*)} = \mathbb{E}_{p_\Phi(\mathbf{x}_{\hat{t}}|\mathbf{c})}\left[\mathbb{E}_{p_\Phi(\mathbf{x}_{\hat{t}-1}|\mathbf{x}_{\hat{t}}, \mathbf{c})}\left[-\Big(\nabla_\omega T_\omega\Big)\nabla_\phi^T \log\big(p_\phi(\mathbf{x}_{\hat{t}-1}|\mathbf{x}_{\hat{t}}, \mathbf{c}^*)\big)\right]\right]\bigg|_{(\phi^*, \omega^*)}$$

$$\text{(98)}$$

$$= K_{TP}^T \quad \text{(99)}$$

With Assumption B.7, we get

$$\nabla_\omega^2 \mathcal{J}(\phi, \omega)\bigg|_{(\phi^*, \omega^*)} = \mathbb{E}_{p_\Phi(\mathbf{x}_{\hat{t}}|\mathbf{c})}\left[\int_{\mathbf{x}_{\hat{t}-1}} p_\Phi(\mathbf{x}_{\hat{t}-1}|\mathbf{x}_{\hat{t}}, \mathbf{c})\nabla_\omega^2 T_\omega - p_{\phi^*}(\mathbf{x}_{\hat{t}-1}|\mathbf{x}_{\hat{t}}, \mathbf{c}^*)\cdot\right. \quad \text{(100)}$$

$$\left.\Big((f^*)''(T_{\omega^*})\nabla_\omega T_\omega \nabla_\omega^T T_\omega + (f^*)'(T_{\omega^*})\nabla_\omega^2 T_\omega\Big)d\mathbf{x}_{\hat{t}-1}\right] \quad \text{(101)}$$

$$= \mathbb{E}_{p_\Phi(\mathbf{x}_{\hat{t}}|\mathbf{c})}\left[\int_{\mathbf{x}_{\hat{t}-1}} p_\Phi(\mathbf{x}_{\hat{t}-1}|\mathbf{x}_{\hat{t}}, \mathbf{c})\Big(\nabla_\omega^2 T_\omega -\right. \quad \text{(102)}$$

$$\left.(f^*)''(T_{\omega^*})\nabla_\omega T_\omega \nabla_\omega^T T_\omega - (f^*)'(T_{\omega^*})\nabla_\omega^2 T_\omega\Big)d\mathbf{x}_{\hat{t}-1}\right] \quad \text{(103)}$$

$$= \mathbb{E}_{p_\Phi(\mathbf{x}_{\hat{t}}|\mathbf{c})}\left[\int_{\mathbf{x}_{\hat{t}-1}} p_\Phi(\mathbf{x}_{\hat{t}-1}|\mathbf{x}_{\hat{t}}, \mathbf{c})\Big(\nabla_\omega^2 T_\omega -\right. \quad \text{(104)}$$

$$\left.(f^*)''(T_{\omega^*})\nabla_\omega T_\omega \nabla_\omega^T T_\omega - \nabla_\omega^2 T_\omega\Big)d\mathbf{x}_{\hat{t}-1}\right] \quad \text{(105)}$$

$$= \mathbb{E}_{p_\Phi(\mathbf{x}_{\hat{t}}|\mathbf{c})}\left[\int_{\mathbf{x}_{\hat{t}-1}} p_\Phi(\mathbf{x}_{\hat{t}-1}|\mathbf{x}_{\hat{t}}, \mathbf{c})\Big(-(f^*)''(T_{\omega^*})\nabla_\omega T_\omega \nabla_\omega^T T_\omega\Big)d\mathbf{x}_{\hat{t}-1}\right] \quad \text{(106)}$$

$$= \mathbb{E}_{p_\Phi(\mathbf{x}_{\hat{t}}|\mathbf{c})}\left[\mathbb{E}_{p_\Phi(\mathbf{x}_{\hat{t}-1}|\mathbf{x}_{\hat{t}}, \mathbf{c})}\left[-(f^*)''(T_{\omega^*})\nabla_\omega T_\omega \nabla_\omega^T T_\omega\right]\right]\bigg|_{(\phi^*, \omega^*)} \quad \text{(107)}$$

$$\triangleq K_{TT} \quad \text{(108)}$$

where $K_{TT} \prec 0$ under Assumptions B.6 and B.8. $\qquad \square$

Therefore, the Jacobian is

$$J = \begin{pmatrix} -\nabla_\phi^2 \mathcal{J}(\phi, \omega) & -\nabla_\omega \nabla_\phi \mathcal{J}(\phi, \omega) \\ \nabla_\phi \nabla_\omega \mathcal{J}(\phi, \omega) & \nabla_\omega^2 \mathcal{J}(\phi, \omega) \end{pmatrix} \tag{109}$$

$$= \begin{pmatrix} \mathbf{0} & -K_{TP} \\ K_{TP}^T & K_{TT} \end{pmatrix} \tag{110}$$

**Theorem 4.2.** *The dynamical system defined in equation 8 is locally exponentially stable with respect to an equilibrium point $(\phi^*, \omega^*)$ under Assumptions B.6, B.7, B.8. Let $\lambda_{min}(\cdot)$ and $\lambda_{max}(\cdot)$ be the smallest and largest eigenvalues of a given matrix, respectively. The rate of convergence of the system is governed by the eigenvalues of the Jacobian $\mathbf{J}$ which have a negative real part upper bounded as*

- *When $Im(\lambda) = 0$,*

$$Re(\lambda) \leq -\frac{\lambda_{min}(-\mathbf{K}_{TT})\lambda_{min}(\mathbf{K}_{TP}\mathbf{K}_{TP}^T)}{\lambda_{min}(-\mathbf{K}_{TT})\lambda_{max}(-\mathbf{K}_{TT}) + \lambda_{min}(\mathbf{K}_{TP}\mathbf{K}_{TP}^T)} \tag{111}$$

- *When $Im(\lambda) \neq 0$,*

$$Re(\lambda) \leq -\frac{\lambda_{min}(-\mathbf{K}_{TT})}{2} \tag{112}$$

*Proof.* From assumptions B.6, B.7, B.8, we know that $-\mathbf{K}_{TT}$ is positive definite and $\mathbf{K}_{TP}$ is full row rank. From Theorem B.4, we know that all the eigenvalues of the dynamical system in equation 8 have negative real part. From theorem B.3, we know that the system is stable because all the eigenvalues have negative real part. $\qquad \square$

### B.2.4 On the Effect of Different $f$-Divergences on Convergence

This section theoretically investigates the impact of the $f$-divergence choice on the local convergence of the dynamical system in equation 8. We establish that, within our framework, a proper choice of $f$-divergence theoretically accelerates convergence to an equilibrium point $(\phi^*, \omega^*)$.

While $f$-divergence formulations provide elegant and general solutions to a problem, identifying the optimal $f$-divergence remains a critical challenge of relevant interest. The class of well-known $f$-divergences is, in fact, large, thus presenting a significant computational burden for exhaustive empirical evaluation. There are two main approaches to assess the effectiveness of different $f$-divergences for a specific task: empirical and theoretical evaluation.

**Empirical evaluation** Empirical evaluation is the most frequent way to analyze the performance of different $f$-divergences in a specific task. This type of evaluation is possible for most algorithms and usually the best-performing $f$-divergence depends on the task. Usually, empirical evaluation is realized by implementing and testing one loss function for each $f$-divergence. Alternatively, few algorithms set the problem of finding the best $f$-divergence as an optimization problem that can be solved via gradient methods (Zhang et al., 2020). However, this latter approach usually leads to a higher computational complexity and more difficult training. We discuss the empirical comparison between $f$-divergences for the algorithms proposed in this paper in Sections 5 and C.

**Theoretical evaluation** While a theoretical evaluation is not always feasible, as it depends on both the specific problem and the $f$-divergence-based algorithm, when possible, such a study offers clear and elegant guidelines for selecting $f$ (e.g., Wei & Liu (2021)). Below, we provide a theoretical convergence comparison of $f$-divergences within our method.

Theorem 4.2 provides two important contributions. Firstly, it proves the local exponential stability of the dynamical system in equation 8. Secondly, it provides an expression for the upper bound on the eigenvalues of the same dynamical system. By studying these upper bounds, we evaluate the effect of different $f$-divergences on the system's rate of convergence. Immediately from the expression of $\mathbf{K}_{TT}$ is clear that the $f$-divergence plays a crucial role in the local convergence and that by choosing

different $f$-divergences (thus having different Fenchel conjugates), we impact the convergence speed. In the following, we show how to obtain some guidelines on the choice of the $f$-divergence to obtain faster convergence. For simplicity, we define $\mathbf{K}'_{TT} = -\mathbf{K}_{TT}$. Recall that $\lambda(\mathbf{K}'_{TT}) > 0$ and $\lambda(\mathbf{K}_{TP}\mathbf{K}_{TP}^T) > 0$, for any $\lambda$ eigenvalue. We refer to the eigenvalues of the Jacobian as $\lambda^J$, $\lambda^J < 0$.

To achieve faster convergence, we want the largest eigenvalue (smallest in absolute value) to be as small as possible (as large as possible in absolute value). When $\text{Im}(\lambda^J) \neq 0$, the bound in equation 112 implies that we have faster convergence when the smallest eigenvalue of $\mathbf{K}'_{TT}$ is larger. When $\text{Im}(\lambda^J) = 0$, we study the upper bound on the eigenvalues of the dynamical system:

- When $\lambda_m(\mathbf{K}_{TP}\mathbf{K}_{TP}^T) \gg \lambda_M(\mathbf{K}'_{TT})$, $\text{Re}(\lambda_M^J) \approx -\lambda_m(\mathbf{K}'_{TT})$. Thus, we achieve faster convergence when $\lambda_m(\mathbf{K}'_{TT})$ is larger.

- When $\lambda_m(\mathbf{K}_{TP}\mathbf{K}_{TP}^T) \approx \lambda_M(\mathbf{K}'_{TT})$, $\text{Re}(\lambda_M^J) \approx -\frac{\lambda_m(\mathbf{K}'_{TT})}{1+\lambda_m(\mathbf{K}'_{TT})}$. Thus, we attain faster convergence when $\lambda_m(\mathbf{K}'_{TT})$ is larger.

- When $\lambda_m(\mathbf{K}_{TP}\mathbf{K}_{TP}^T) \ll \lambda_M(\mathbf{K}'_{TT})$, $\text{Re}(\lambda_M^J) \approx -\frac{\lambda_m(\mathbf{K}_{TP}\mathbf{K}_{TP}^T)}{\lambda_M(\mathbf{K}'_{TT})} \approx 0$.

- $\lambda_M(\mathbf{K}'_{TT})$ is only present at the denominator. Therefore, we have faster convergence when $\lambda_M(\mathbf{K}'_{TT})$ is smaller.

Although it is not possible to clearly define some unique conditions to achieve faster convergence that hold true for all the possible cases, we can definitively affirm that a larger value of $\lambda_m(\mathbf{K}'_{TT})$ is probably beneficial when $\text{Im}(\lambda^J) = 0$ and certainly beneficial when $\text{Im}(\lambda^J) \neq 0$. We notice that $\mathbf{K}'_{TT}$ is directly proportional to $(f^*)''(T_\omega)|_{(\phi^*, \omega^*)}$. Thus, we have faster convergence when $(f^*)''(T_\omega)|_{(\phi^*, \omega^*)}$ is larger. Thus, we obtain

$$(f^*)''(T_\omega)|_{(\phi^*, \omega^*)} = (f^*)''(f'(1)) = \frac{1}{f''(1)}, \tag{113}$$

which can be demonstrated by using Lemma B.5. We evaluate equation 113 for different $f$-divergences:

- KL divergence: $\frac{1}{f''(1)}\Big|_{f=\text{KL}} = 1$.

- Reverse KL divergence: $\frac{1}{f''(1)}\Big|_{f=\text{RKL}} = 1$.

- $\chi^2$ divergence: $\frac{1}{f''(1)}\Big|_{f=\chi^2} = \frac{1}{2}$.

- Jensen-Shannon divergence: $\frac{1}{f''(1)}\Big|_{f=\text{JS}} = 2$.

- Squared Hellinger distance: $\frac{1}{f''(1)}\Big|_{f=H^2} = 2$.

Thus, we can conclude that JS and $H^2$ divergences have better convergence properties than the other divergences analyzed.

## B.3 SUMMARY OF $f$-DIVERGENCE CHOICE IN $f$-DMU

In this section, we summarize all the theoretical contributions in Table 5, which helps in choosing the correct method and $f$-divergence given the user desired characteristics. Additionally, in Section 5, we provide further comments on the different $f$-divergences, which are also related to more practical observations.

Table 5: Summary of $f$-divergences characteristics for the $f$-DMU framework

| Framework | Advantages | Disadvantages | $f$-Divergence | Characteristics |
|---|---|---|---|---|
| **Closed-form** | a) Computationally fast; b) Divergence minimization guarantee with any batch size | Does not allow the usage of any $f$-divergence | KL | Standard loss |
| | | | $H^2$ | Bounded gradients |
| | | | $\chi^2$ | Fast-growing gradients |
| | | | $\alpha$ | High flexibility |
| **Variational** | Flexible as it allows the usage of any $f$-divergence | a) Unlearning depends on divergence estimate; b) Need to train a discriminator | KL | Normal convergence |
| | | | $H^2$ | Fast convergence |
| | | | $\chi^2$ | Slow convergence |
| | | | JS | Fast convergence |
| | | | RKL | Normal convergence |

# C  ADDITIONAL EXPERIMENTAL RESULTS

## C.1  EXPERIMENTAL SETUP

To comprehensively evaluate the performance of our proposed unlearning methods, we designed a series of experiments spanning single-concept and multi-concept erasure scenarios. Our setup allows for a systematic analysis of different loss functions, unlearning strategies, and regularization techniques. The experiments are conducted on SD 1.4, SD 2.1, and SDXL, targeting a diverse set of concepts, including fictional characters (*Snoopy, Grumpy Cat, Wall-E, R2D2*) and distinct artistic styles (*Van Gogh, Salvador Dalì*). For the experiments, we used the finetuning and evaluating prompts from Gandikota et al. (2023). We used a DDIM sampler with 50 inference steps for the experiments in Tables 9 and 10, and 30 steps for all other experiments.

**Implementation Details**  All experiments were conducted on a workstation equipped with a single NVIDIA GeForce RTX 4090 GPU and an Intel Core i9 CPU. We utilized the PyTorch deep learning framework for all implementations. For SD 1.4 and SD 2.1, to manage memory constraints while maintaining a stable training process, we used a batch size of 4. This was combined with a gradient accumulation step of 2, resulting in an effective batch size of 8 for each weight update. For SDXL, due to memory constraints, we used a batch size of 1. For each closed-form method, the number of fine-tuning epochs is 500 unless specified differently. For each variational-based method, before performing unlearning (of $N$ steps), we perform $N$ steps of discriminator warm-up. Usually $N = 500$, unless specified differently. We noticed that, on different computer architectures and GPUs, the compute cost of the variational framework is about five times bigger than the closed-form framework (including the discriminator warm up). For all the methods, we use AdamW Loshchilov & Hutter (2017) optimizer with a learning rate $6 \cdot 10^{-6}$ and update the cross-attention parameters. For SDXL, we set the learning rate to $6 \cdot 10^{-5}$.

**Anchor Concepts**  For MSE and Hellinger loss functions, we investigate three distinct unlearning strategies defined by the choice of the anchor distribution, which is the distribution we guide the erased concept towards: Empty Anchor ('empty'), Near Anchor ('near'), and Superclass Anchor ('superclass'). With Empty anchor the target concept is pushed towards a null or generic distribution, represented by an empty text prompt. This strategy aims for the complete removal of the concept's specific features. With Near Anchor, the model is trained to associate the target concept's prompt with a semantically similar but distinct concept (e.g., erasing "van Gogh" by guiding it towards "Salvador Dali"). This evaluates the model's ability to remap, rather than simply erase, concepts. With Superclass Anchor, the target concept is guided towards a more general, categorical prompt (e.g., erasing "Van Gogh" by guiding it towards the generic prompt "a painting"). This method tests the ability to abstract a specific instance into its broader category. The results are showed in Fig. 6.

**Regularizations**  To ensure a stable unlearning process that does not degrade the model's overall performance, we incorporate several key techniques. We tested different regularization techniques: with the symbol ‡, we indicate the experiment using prior preservation loss (Gandikota et al., 2023) and Gradient Surgery (Yu et al., 2020b); with the symbol † we indicate that we sample only the final part of the DM generation trajectory (Lu et al., 2024) (referred to as importance sampling).

**Experimental Scenarios**  Our evaluation is structured around two main scenarios: single and multi-concept erasure. For single concept erasure, we perform experiments to erase each target concept individually. To understand the dynamics of unlearning over time, these experiments are conducted for varying numbers of training iterations: 500, 2000, and 5000 steps as shown in Tables 6, 1, and 7. A separate variational setup with 150 steps, detailed in Table 8, is also explored to assess performance under a limited computational budget.

## C.2  MULTI-CONCEPT ERASURE

Fig. 8 and Tab. 3 report the results of sequentially erasing 10 artistic styles: Claude Monet, Pablo Picasso, Vincent Van Gogh, Apollinary Vasnetsov, Eric Fischl, Greg Rutkowski, Jeremy Mann, Johannes Vermeer, John Whitcomb, and Nicolas Mignard.

Fig. 13 shows a qualitative evaluation of unlearning 3 unrelated concepts. Fig. 13 shows a trend: images produced by the $H^2$-based method are more coherent (image-wise and prompt-wise) and have fewer artifacts.

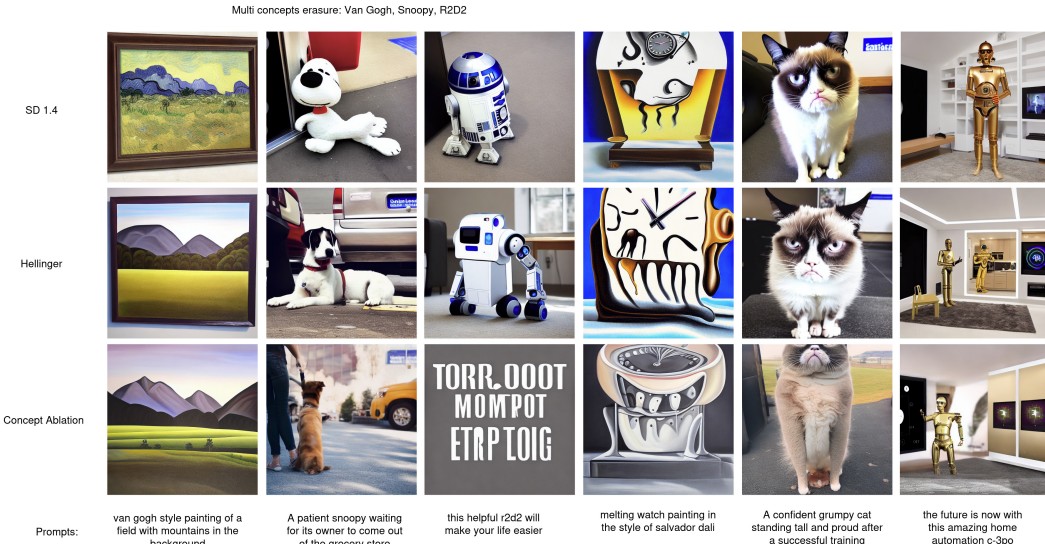

Figure 13: Qualitative comparison of superclass unlearning with sequential multiple erasure. The first three columns display concepts targeted for erasure. The last three columns show preserved concepts.

## C.3 ABLATION EFFICACY AND PRESERVATION

The main goal of concept unlearning is to efficiently remove a target concept while causing no collateral damage to unrelated concepts. Our experiments reveal performance traits of the various $f$-divergences.

**Understanding Kernel Inception Distance**    The Interpretation of KID needs to be clarified. A lower KID means greater similarity of two image distributions. We calculate it between the model before and after the erasure on the same prompts. For non-erased concepts, a lower KID is better: it means the model has effectively retained its ability to generate images of high quality and diversity for these concepts, with the original distribution being closely matched. For erased concepts, the interpretation of KID depends on the specific objective of unlearning: if the desired situation is for the model to output strange, monochrome, or unrecognizable images when prompted by the erased concept, then a higher KID (relative to the original concept's distribution) is preferred, indicating a strong deviation from any coherent features. On the other hand, if the objective is to shift the erased concept toward a realistic but generic or alternate representation, then achieving a lower KID would mean the erasure performed well, especially without turning the images into noise.

**Tracking by number of iterations**    The number of training iterations seems to have a strong impact on the effectiveness and retention of unlearning. More generally, the bigger the number of steps, ranging from 500 to 5000, the more thoroughly the target concept is unlearned, evidenced by consistently lower CS and CA values for the erased concept. For instance, "hellinger empty" for "Snoopy" returns, for 500 steps, CS=0.54, CA=0.47, KID=0.269 (Table 6), while for 5000 steps, CS=0.52, CA=0.1, KID=0.346 (in Table 7). While CS and CA values decrease, the KID value increases.

**Role of regularization**    The effects of regularization, namely Prior Preservation/Gradient Surgery ($\ddagger$) and Importance Sampling ($\dagger$), are highly context-dependent. In longer optimization runs, such as the 500 and 5000-step experiments (Table 6 and Table 7), applying these methods reveals a distinct trade-off. For instance, when erasing "Snoopy" for 500 steps, the non-regularized "hellinger empty"

Table 6: Quantitative results with 500 iterations, reporting CS, CA, and KID. For methods other than the original model, the best values are in bold and the second-best in italics. † denotes Importance Sampling and ‡ denotes Prior Preservation and Gradient Surgery. Base model: SD 1.4.

**Erasing Snoopy**

| Method | Snoopy CS↓ | CA↑ | KID | Grumpy Cat CS↑ | CA↓ | KID↓ | Wall-E CS↑ | CA↑ | KID↓ | R2D2 CS↑ | CA↑ | KID↓ | Van Gogh CS↑ | CA↑ | KID↓ | Salvador Dali CS↑ | CA↑ | KID↓ |
|---|---|---|---|---|---|---|---|---|---|---|---|---|---|---|---|---|---|---|
| Original Model | 0.73 | 1.00 | - | 0.74 | 1.00 | - | 0.73 | 1.00 | - | 0.75 | 1.00 | - | 0.80 | 1.00 | - | 0.68 | 0.80 | - |
| (Ours) hellinger empty | 0.54 | 0.47 | 0.269 | 0.74 | 0.90 | 0.001 | 0.61 | 0.77 | 0.064 | 0.77 | 1.00 | 0.063 | 0.77 | 1.00 | -0.012 | 0.68 | 0.80 | -0.014 |
| (Ours) hellinger empty ‡ | 0.59 | 0.83 | 0.059 | 0.71 | 1.00 | 0.025 | 0.70 | 0.93 | 0.020 | 0.76 | 1.00 | 0.035 | 0.76 | 1.00 | -0.022 | 0.68 | 0.83 | -0.024 |
| (Ours) hellinger empty ‡ † | 0.61 | 1.00 | 0.061 | 0.73 | 1.00 | -0.003 | 0.71 | 0.93 | 0.010 | 0.77 | 1.00 | 0.043 | 0.79 | 1.00 | -0.013 | 0.68 | 0.90 | -0.020 |
| (Ours) hellinger near | 0.51 | 0.70 | 0.298 | 0.78 | 1.00 | 0.141 | 0.69 | 0.93 | 0.027 | 0.77 | 1.00 | 0.045 | 0.78 | 1.00 | -0.007 | 0.69 | 0.90 | -0.029 |
| (Ours) hellinger near ‡ | 0.54 | 0.90 | 0.158 | 0.75 | 1.00 | 0.002 | 0.72 | 0.90 | -0.006 | 0.76 | 1.00 | 0.012 | 0.77 | 0.97 | -0.022 | 0.67 | 0.77 | -0.024 |
| (Ours) hellinger near ‡ † | 0.56 | 0.93 | 0.152 | 0.75 | 1.00 | 0.003 | 0.73 | 0.93 | 0.000 | 0.76 | 1.00 | 0.016 | 0.78 | 1.00 | -0.007 | 0.68 | 0.90 | -0.023 |
| (Ours) hellinger superclass | 0.55 | 0.93 | 0.127 | 0.70 | 1.00 | -0.002 | 0.71 | 0.97 | -0.000 | 0.77 | 1.00 | -0.000 | 0.78 | 1.00 | -0.028 | 0.69 | 0.93 | -0.031 |
| (Ours) hellinger superclass ‡ | 0.61 | 1.00 | 0.062 | 0.72 | 1.00 | -0.006 | 0.76 | 0.97 | 0.003 | 0.76 | 1.00 | -0.001 | 0.79 | 1.00 | -0.017 | 0.68 | 0.80 | -0.021 |
| (Ours) hellinger superclass ‡ † | 0.61 | 1.00 | 0.069 | 0.74 | 1.00 | -0.012 | 0.77 | 0.97 | -0.000 | 0.76 | 1.00 | -0.001 | 0.79 | 1.00 | -0.004 | 0.69 | 0.97 | -0.025 |
| MSE empty | 0.53 | 0.33 | 0.253 | 0.74 | 0.97 | 0.006 | 0.64 | 0.93 | 0.057 | 0.76 | 1.00 | 0.049 | 0.78 | 1.00 | -0.009 | 0.68 | 0.90 | -0.015 |
| MSE near | 0.50 | 0.83 | 0.288 | 0.78 | 1.00 | 0.167 | 0.71 | 0.97 | 0.023 | 0.76 | 1.00 | 0.039 | 0.78 | 1.00 | -0.015 | 0.69 | 0.83 | -0.025 |
| Concept Ablation | 0.56 | 1.00 | 0.127 | 0.68 | 1.00 | 0.016 | 0.71 | 0.90 | -0.005 | 0.77 | 1.00 | 0.006 | 0.78 | 1.00 | -0.023 | 0.68 | 0.90 | -0.029 |

**Erasing Grumpy Cat**

| Method | Snoopy CS↑ | CA↑ | KID↓ | Grumpy Cat CS↓ | CA↓ | KID | Wall-E CS↑ | CA↑ | KID↓ | R2D2 CS↑ | CA↑ | KID↓ | Van Gogh CS↑ | CA↑ | KID↓ | Salvador Dali CS↑ | CA↑ | KID↓ |
|---|---|---|---|---|---|---|---|---|---|---|---|---|---|---|---|---|---|---|
| Original Model | 0.73 | 1.00 | - | 0.74 | 1.00 | - | 0.73 | 1.00 | - | 0.75 | 1.00 | - | 0.80 | 1.00 | - | 0.68 | 0.80 | - |
| (Ours) hellinger empty | 0.65 | 1.00 | 0.010 | 0.47 | 0.03 | 0.250 | 0.61 | 0.83 | 0.055 | 0.75 | 1.00 | 0.044 | 0.77 | 1.00 | -0.009 | 0.69 | 0.83 | -0.012 |
| (Ours) hellinger empty ‡ | 0.71 | 1.00 | -0.014 | 0.65 | 0.93 | 0.056 | 0.75 | 1.00 | -0.001 | 0.76 | 1.00 | 0.003 | 0.77 | 0.97 | -0.013 | 0.68 | 0.83 | -0.029 |
| (Ours) hellinger empty ‡ † | 0.70 | 1.00 | -0.008 | 0.62 | 0.87 | 0.058 | 0.73 | 1.00 | 0.002 | 0.75 | 1.00 | 0.010 | 0.77 | 1.00 | -0.029 | 0.70 | 0.90 | -0.030 |
| (Ours) hellinger near | 0.75 | 1.00 | 0.030 | 0.52 | 0.23 | 0.223 | 0.74 | 0.90 | 0.028 | 0.76 | 1.00 | 0.036 | 0.78 | 1.00 | -0.016 | 0.69 | 0.80 | -0.027 |
| (Ours) hellinger near ‡ | 0.70 | 1.00 | -0.015 | 0.63 | 0.97 | 0.062 | 0.75 | 0.97 | -0.009 | 0.76 | 1.00 | 0.011 | 0.78 | 1.00 | -0.012 | 0.68 | 0.87 | -0.011 |
| (Ours) hellinger near ‡ † | 0.70 | 1.00 | -0.002 | 0.61 | 1.00 | 0.064 | 0.73 | 0.93 | -0.003 | 0.76 | 1.00 | 0.013 | 0.75 | 0.97 | 0.009 | 0.68 | 0.83 | -0.022 |
| (Ours) hellinger superclass | 0.70 | 1.00 | -0.004 | 0.57 | 0.83 | 0.140 | 0.74 | 0.93 | 0.002 | 0.77 | 1.00 | 0.030 | 0.78 | 1.00 | -0.026 | 0.69 | 0.83 | -0.024 |
| (Ours) hellinger superclass ‡ | 0.70 | 1.00 | -0.012 | 0.68 | 1.00 | 0.041 | 0.73 | 0.97 | -0.013 | 0.75 | 1.00 | 0.009 | 0.78 | 1.00 | -0.011 | 0.68 | 0.90 | -0.019 |
| (Ours) hellinger superclass ‡ † | 0.72 | 1.00 | -0.019 | 0.66 | 1.00 | 0.035 | 0.76 | 0.97 | -0.004 | 0.75 | 1.00 | 0.003 | 0.78 | 1.00 | -0.021 | 0.69 | 0.90 | -0.034 |
| MSE empty | 0.67 | 1.00 | 0.000 | 0.49 | 0.10 | 0.237 | 0.63 | 0.93 | 0.067 | 0.77 | 1.00 | 0.033 | 0.77 | 1.00 | -0.009 | 0.68 | 0.90 | -0.015 |
| MSE near | 0.78 | 1.00 | 0.171 | 0.54 | 0.27 | 0.249 | 0.74 | 0.97 | 0.042 | 0.76 | 1.00 | 0.028 | 0.77 | 1.00 | 0.015 | 0.68 | 0.80 | -0.026 |
| Concept Ablation | 0.69 | 1.00 | -0.013 | 0.58 | 0.97 | 0.147 | 0.72 | 0.93 | -0.003 | 0.77 | 1.00 | 0.038 | 0.78 | 1.00 | -0.026 | 0.69 | 0.90 | -0.025 |

**Erasing Wall-E**

| Method | Snoopy CS↑ | CA↑ | KID↓ | Grumpy Cat CS↑ | CA↑ | KID↓ | Wall-E CS↓ | CA↓ | KID | R2D2 CS↑ | CA↑ | KID↓ | Van Gogh CS↑ | CA↑ | KID↓ | Salvador Dali CS↑ | CA↑ | KID↓ |
|---|---|---|---|---|---|---|---|---|---|---|---|---|---|---|---|---|---|---|
| Original Model | 0.73 | 1.00 | - | 0.74 | 1.00 | - | 0.73 | 1.00 | - | 0.75 | 1.00 | - | 0.80 | 1.00 | - | 0.68 | 0.80 | - |
| (Ours) hellinger empty | 0.66 | 1.00 | 0.023 | 0.72 | 0.97 | -0.011 | 0.55 | 0.77 | 0.089 | 0.71 | 0.90 | 0.092 | 0.77 | 1.00 | -0.021 | 0.68 | 0.70 | -0.025 |
| (Ours) hellinger empty ‡ | 0.68 | 1.00 | -0.007 | 0.75 | 1.00 | -0.016 | 0.57 | 0.77 | 0.071 | 0.76 | 1.00 | 0.054 | 0.78 | 1.00 | 0.004 | 0.69 | 0.90 | -0.019 |
| (Ours) hellinger empty ‡ † | 0.69 | 1.00 | 0.004 | 0.73 | 1.00 | -0.014 | 0.57 | 0.90 | 0.057 | 0.75 | 0.97 | 0.038 | 0.78 | 1.00 | -0.002 | 0.69 | 0.90 | -0.029 |
| (Ours) hellinger near | 0.71 | 1.00 | -0.019 | 0.74 | 1.00 | -0.012 | 0.56 | 0.97 | 0.194 | 0.76 | 1.00 | 0.097 | 0.76 | 1.00 | -0.025 | 0.68 | 0.80 | -0.016 |
| (Ours) hellinger near ‡ | 0.73 | 1.00 | -0.013 | 0.75 | 1.00 | -0.008 | 0.59 | 0.97 | 0.153 | 0.76 | 1.00 | 0.039 | 0.78 | 1.00 | 0.006 | 0.69 | 0.83 | -0.022 |
| (Ours) hellinger near ‡ † | 0.71 | 1.00 | -0.010 | 0.74 | 1.00 | -0.012 | 0.59 | 1.00 | 0.146 | 0.77 | 1.00 | 0.089 | 0.78 | 1.00 | 0.010 | 0.69 | 0.87 | -0.017 |
| (Ours) hellinger superclass | 0.73 | 1.00 | -0.008 | 0.74 | 1.00 | -0.016 | 0.61 | 0.90 | 0.092 | 0.76 | 1.00 | 0.033 | 0.79 | 1.00 | -0.028 | 0.69 | 0.83 | -0.015 |
| (Ours) hellinger superclass ‡ | 0.72 | 1.00 | -0.006 | 0.74 | 1.00 | -0.008 | 0.68 | 0.90 | 0.072 | 0.76 | 1.00 | 0.039 | 0.79 | 0.97 | 0.004 | 0.69 | 0.90 | -0.021 |
| (Ours) hellinger superclass ‡ † | 0.72 | 1.00 | -0.018 | 0.74 | 1.00 | -0.009 | 0.68 | 0.93 | 0.064 | 0.77 | 1.00 | 0.091 | 0.80 | 1.00 | 0.004 | 0.70 | 0.93 | -0.024 |
| MSE empty | 0.66 | 1.00 | 0.013 | 0.74 | 1.00 | -0.009 | 0.54 | 0.57 | 0.098 | 0.67 | 0.87 | 0.111 | 0.78 | 1.00 | -0.021 | 0.68 | 0.77 | -0.023 |
| MSE near | 0.70 | 1.00 | -0.016 | 0.74 | 1.00 | -0.020 | 0.56 | 1.00 | 0.200 | 0.76 | 1.00 | 0.118 | 0.76 | 0.97 | 0.000 | 0.68 | 0.83 | -0.022 |
| Concept Ablation | 0.73 | 1.00 | -0.017 | 0.73 | 1.00 | -0.009 | 0.60 | 0.83 | 0.094 | 0.76 | 1.00 | 0.065 | 0.78 | 1.00 | -0.021 | 0.68 | 0.80 | -0.021 |

**Erasing R2D2**

| Method | Snoopy CS↑ | CA↑ | KID↓ | Grumpy Cat CS↑ | CA↑ | KID↓ | Wall-E CS↑ | CA↑ | KID↓ | R2D2 CS↓ | CA↓ | KID | Van Gogh CS↑ | CA↑ | KID↓ | Salvador Dali CS↑ | CA↑ | KID↓ |
|---|---|---|---|---|---|---|---|---|---|---|---|---|---|---|---|---|---|---|
| Original Model | 0.73 | 1.00 | - | 0.74 | 1.00 | - | 0.73 | 1.00 | - | 0.75 | 1.00 | - | 0.80 | 1.00 | - | 0.68 | 0.80 | - |
| (Ours) hellinger empty | 0.62 | 1.00 | 0.049 | 0.74 | 0.97 | -0.010 | 0.58 | 0.70 | 0.102 | 0.53 | 0.20 | 0.252 | 0.76 | 1.00 | -0.008 | 0.68 | 0.70 | -0.011 |
| (Ours) hellinger empty ‡ | 0.69 | 1.00 | -0.003 | 0.73 | 1.00 | -0.017 | 0.72 | 0.93 | 0.003 | 0.59 | 0.80 | 0.186 | 0.75 | 0.97 | -0.019 | 0.70 | 0.87 | -0.023 |
| (Ours) hellinger empty ‡ † | 0.69 | 1.00 | -0.004 | 0.71 | 1.00 | -0.006 | 0.74 | 0.93 | 0.011 | 0.58 | 0.77 | 0.154 | 0.77 | 0.97 | -0.009 | 0.70 | 0.83 | -0.032 |
| (Ours) hellinger near | 0.75 | 1.00 | 0.007 | 0.73 | 1.00 | -0.001 | 0.81 | 1.00 | 0.100 | 0.58 | 0.93 | 0.348 | 0.77 | 1.00 | -0.008 | 0.70 | 0.90 | -0.018 |
| (Ours) hellinger near ‡ | 0.73 | 1.00 | -0.008 | 0.74 | 1.00 | -0.012 | 0.78 | 0.97 | 0.008 | 0.64 | 1.00 | 0.235 | 0.79 | 1.00 | -0.015 | 0.70 | 0.90 | -0.023 |
| (Ours) hellinger near ‡ † | 0.74 | 1.00 | -0.013 | 0.73 | 1.00 | -0.016 | 0.79 | 1.00 | 0.024 | 0.64 | 1.00 | 0.219 | 0.80 | 1.00 | -0.006 | 0.69 | 0.83 | -0.029 |
| (Ours) hellinger superclass | 0.74 | 1.00 | -0.014 | 0.72 | 1.00 | -0.016 | 0.76 | 0.93 | 0.027 | 0.54 | 0.93 | 0.225 | 0.79 | 1.00 | -0.031 | 0.70 | 0.90 | -0.029 |
| (Ours) hellinger superclass ‡ | 0.74 | 1.00 | -0.018 | 0.74 | 1.00 | -0.007 | 0.75 | 0.93 | 0.005 | 0.64 | 1.00 | 0.203 | 0.79 | 1.00 | -0.010 | 0.70 | 0.90 | -0.024 |
| (Ours) hellinger superclass ‡ † | 0.72 | 1.00 | -0.018 | 0.75 | 1.00 | -0.017 | 0.77 | 1.00 | 0.010 | 0.67 | 1.00 | 0.187 | 0.80 | 1.00 | 0.013 | 0.71 | 0.90 | -0.021 |
| MSE empty | 0.63 | 1.00 | 0.012 | 0.75 | 0.97 | -0.011 | 0.58 | 0.87 | 0.092 | 0.53 | 0.30 | 0.260 | 0.75 | 1.00 | 0.016 | 0.68 | 0.80 | -0.013 |
| MSE near | 0.73 | 1.00 | 0.014 | 0.72 | 1.00 | 0.027 | 0.72 | 0.73 | 0.199 | 0.55 | 0.53 | 0.434 | 0.75 | 1.00 | 0.023 | 0.68 | 0.87 | 0.005 |
| Concept Ablation | 0.74 | 1.00 | -0.009 | 0.73 | 1.00 | 0.001 | 0.76 | 0.97 | 0.030 | 0.55 | 0.93 | 0.244 | 0.78 | 1.00 | -0.039 | 0.70 | 0.90 | -0.031 |

**Erasing Van Gogh**

| Method | Snoopy CS↑ | CA↑ | KID↓ | Grumpy Cat CS↑ | CA↑ | KID↓ | Wall-E CS↑ | CA↑ | KID↓ | R2D2 CS↑ | CA↑ | KID↓ | Van Gogh CS↓ | CA↓ | KID | Salvador Dali CS↑ | CA↑ | KID↓ |
|---|---|---|---|---|---|---|---|---|---|---|---|---|---|---|---|---|---|---|
| Original Model | 0.73 | 1.00 | - | 0.74 | 1.00 | - | 0.73 | 1.00 | - | 0.75 | 1.00 | - | 0.80 | 1.00 | - | 0.68 | 0.80 | - |
| (Ours) hellinger empty | 0.69 | 1.00 | -0.014 | 0.74 | 1.00 | -0.007 | 0.67 | 1.00 | 0.012 | 0.77 | 1.00 | 0.029 | 0.55 | 0.50 | 0.151 | 0.62 | 0.70 | 0.035 |
| (Ours) hellinger empty ‡ | 0.74 | 1.00 | -0.011 | 0.74 | 1.00 | -0.011 | 0.77 | 0.93 | 0.007 | 0.75 | 1.00 | -0.000 | 0.72 | 0.97 | 0.073 | 0.70 | 0.97 | -0.004 |
| (Ours) hellinger empty ‡ † | 0.73 | 1.00 | -0.020 | 0.75 | 1.00 | -0.016 | 0.76 | 0.97 | 0.007 | 0.75 | 1.00 | -0.002 | 0.72 | 0.90 | 0.098 | 0.70 | 0.90 | 0.001 |
| (Ours) hellinger near | 0.72 | 1.00 | -0.017 | 0.74 | 1.00 | -0.010 | 0.74 | 0.97 | -0.012 | 0.77 | 1.00 | 0.008 | 0.56 | 0.00 | 0.186 | 0.70 | 0.97 | 0.049 |
| (Ours) hellinger near ‡ | 0.75 | 1.00 | -0.017 | 0.74 | 1.00 | -0.016 | 0.76 | 0.97 | 0.003 | 0.75 | 1.00 | 0.010 | 0.73 | 0.90 | 0.068 | 0.70 | 0.90 | 0.006 |
| (Ours) hellinger near ‡ † | 0.74 | 1.00 | -0.018 | 0.75 | 1.00 | -0.013 | 0.75 | 0.90 | 0.013 | 0.75 | 1.00 | 0.004 | 0.70 | 0.87 | 0.075 | 0.72 | 1.00 | -0.004 |
| (Ours) hellinger superclass | 0.73 | 1.00 | -0.023 | 0.74 | 1.00 | -0.028 | 0.73 | 0.93 | -0.003 | 0.76 | 1.00 | -0.007 | 0.58 | 0.17 | 0.132 | 0.64 | 0.57 | 0.001 |
| (Ours) hellinger superclass ‡ | 0.74 | 1.00 | -0.015 | 0.74 | 1.00 | -0.016 | 0.77 | 1.00 | 0.006 | 0.74 | 1.00 | 0.006 | 0.71 | 0.93 | 0.081 | 0.69 | 0.90 | -0.019 |
| (Ours) hellinger superclass ‡ † | 0.74 | 1.00 | -0.022 | 0.73 | 1.00 | -0.019 | 0.75 | 0.93 | 0.014 | 0.75 | 1.00 | 0.007 | 0.67 | 0.73 | 0.093 | 0.69 | 0.90 | -0.013 |
| MSE empty | 0.70 | 1.00 | -0.013 | 0.74 | 1.00 | -0.005 | 0.69 | 1.00 | 0.009 | 0.77 | 1.00 | 0.028 | 0.55 | 0.50 | 0.170 | 0.62 | 0.77 | 0.033 |
| MSE near | 0.72 | 1.00 | -0.008 | 0.74 | 1.00 | -0.008 | 0.76 | 1.00 | -0.002 | 0.78 | 1.00 | -0.002 | 0.59 | 0.27 | 0.184 | 0.70 | 0.93 | 0.068 |
| Concept Ablation | 0.74 | 1.00 | -0.023 | 0.74 | 1.00 | -0.019 | 0.74 | 0.93 | -0.003 | 0.76 | 1.00 | -0.004 | 0.57 | 0.20 | 0.117 | 0.64 | 0.63 | 0.004 |

**Erasing Salvador Dali**

| Method | Snoopy CS↑ | CA↑ | KID↓ | Grumpy Cat CS↑ | CA↑ | KID↓ | Wall-E CS↑ | CA↑ | KID↓ | R2D2 CS↑ | CA↑ | KID↓ | Van Gogh CS↑ | CA↑ | KID↓ | Salvador Dali CS↓ | CA↓ | KID |
|---|---|---|---|---|---|---|---|---|---|---|---|---|---|---|---|---|---|---|
| Original Model | 0.73 | 1.00 | - | 0.74 | 1.00 | - | 0.73 | 1.00 | - | 0.75 | 1.00 | - | 0.80 | 1.00 | - | 0.68 | 0.80 | - |
| (Ours) hellinger empty | 0.70 | 1.00 | -0.015 | 0.74 | 1.00 | -0.012 | 0.69 | 0.90 | 0.011 | 0.78 | 1.00 | 0.024 | 0.71 | 0.97 | 0.055 | 0.59 | 0.73 | 0.091 |
| (Ours) hellinger empty ‡ | 0.74 | 1.00 | -0.014 | 0.74 | 1.00 | -0.013 | 0.76 | 0.93 | 0.007 | 0.75 | 1.00 | 0.005 | 0.76 | 0.97 | 0.051 | 0.67 | 0.77 | 0.008 |
| (Ours) hellinger empty ‡ † | 0.73 | 1.00 | -0.018 | 0.75 | 1.00 | -0.006 | 0.76 | 0.90 | 0.016 | 0.74 | 1.00 | 0.009 | 0.79 | 1.00 | 0.064 | 0.67 | 0.87 | 0.003 |
| (Ours) hellinger near | 0.74 | 1.00 | -0.017 | 0.74 | 1.00 | -0.014 | 0.75 | 0.93 | 0.012 | 0.76 | 1.00 | 0.024 | 0.80 | 1.00 | 0.069 | 0.62 | 0.17 | 0.199 |
| (Ours) hellinger near ‡ | 0.76 | 1.00 | -0.011 | 0.73 | 1.00 | -0.007 | 0.75 | 0.90 | 0.006 | 0.75 | 1.00 | 0.000 | 0.77 | 0.97 | 0.047 | 0.67 | 0.73 | 0.028 |
| (Ours) hellinger near ‡ † | 0.74 | 1.00 | -0.019 | 0.73 | 1.00 | -0.019 | 0.78 | 0.93 | 0.009 | 0.75 | 1.00 | -0.003 | 0.78 | 1.00 | 0.039 | 0.66 | 0.57 | 0.043 |
| (Ours) hellinger superclass | 0.75 | 1.00 | -0.007 | 0.75 | 1.00 | -0.008 | 0.75 | 0.90 | 0.009 | 0.74 | 1.00 | -0.009 | 0.75 | 1.00 | 0.050 | 0.60 | 0.30 | 0.054 |
| (Ours) hellinger superclass ‡ | 0.74 | 1.00 | -0.013 | 0.75 | 1.00 | -0.009 | 0.76 | 0.93 | 0.011 | 0.74 | 1.00 | -0.000 | 0.77 | 0.97 | 0.066 | 0.65 | 0.70 | 0.008 |
| (Ours) hellinger superclass ‡ † | 0.73 | 1.00 | -0.017 | 0.74 | 1.00 | -0.018 | 0.77 | 0.93 | 0.002 | 0.75 | 1.00 | -0.004 | 0.78 | 1.00 | 0.046 | 0.63 | 0.43 | 0.009 |
| MSE empty | 0.68 | 1.00 | -0.005 | 0.74 | 1.00 | -0.011 | 0.69 | 0.97 | 0.012 | 0.77 | 1.00 | 0.019 | 0.71 | 1.00 | 0.044 | 0.58 | 0.63 | 0.090 |
| MSE near | 0.73 | 1.00 | -0.001 | 0.74 | 1.00 | 0.001 | 0.75 | 0.90 | 0.014 | 0.75 | 1.00 | 0.027 | 0.77 | 1.00 | 0.062 | 0.58 | 0.10 | 0.239 |
| Concept Ablation | 0.75 | 1.00 | -0.007 | 0.75 | 1.00 | -0.015 | 0.77 | 0.93 | 0.018 | 0.74 | 1.00 | -0.001 | 0.75 | 1.00 | 0.100 | 0.58 | 0.00 | 0.056 |

method is most effective at erasure (CS 0.54, CA 0.47) but produces less realistic images (KID 0.269). Applying regularization weakens the erasure effect (e.g., "hellinger empty ‡ †" has CS 0.61, CA 1.00) but significantly improves image coherence, lowering the KID to 0.061.

Table 7: Quantitative results with 5000 iterations, reporting CS, CA, and KID. For methods other than the Original Model, the best values are in bold and the second-best in italics. [†] denotes Importance Sampling and [‡] denotes Prior Preservation and Gradient Surgery. Base model: SD 1.4.

| | Snoopy | | | Grumpy Cat | | | Wall-E | | | R2D2 | | | Van Gogh | | | Salvador Dali | | |
|---|---|---|---|---|---|---|---|---|---|---|---|---|---|---|---|---|---|---|
| **Erasing Snoopy** | | | | | | | | | | | | | | | | | | |
| Method | CS↓ | CA↓ | KID | CS↑ | CA↑ | KID↓ | CS↑ | CA↑ | KID↓ | CS↑ | CA↑ | KID↓ | CS↑ | CA↑ | KID↓ | CS↑ | CA↑ | KID↓ |
| Original Model | 0.76 | 1.00 | - | 0.71 | 0.98 | - | 0.70 | 0.97 | - | 0.73 | 0.98 | - | 0.77 | 0.97 | - | 0.66 | 0.76 | - |
| (Ours) hellinger empty | **0.52** | *0.10* | 0.346 | 0.50 | 0.00 | 0.341 | 0.58 | 0.85 | 0.108 | 0.65 | **1.00** | 0.170 | 0.72 | *0.95* | 0.060 | *0.62* | **0.90** | 0.057 |
| (Ours) hellinger near | *0.56* | 0.70 | 0.320 | **0.74** | *0.97* | 0.302 | *0.70* | **0.97** | *0.047* | 0.78 | **1.00** | 0.072 | *0.73* | **1.00** | 0.060 | **0.66** | 0.60 | 0.029 |
| (Ours) hellinger superclass | 0.54 | 0.50 | 0.141 | 0.61 | 0.70 | *0.129* | 0.61 | 0.57 | 0.092 | 0.71 | 0.87 | 0.174 | 0.71 | 0.87 | 0.053 | **0.66** | *0.83* | 0.038 |
| (Ours) hellinger superclass [‡] | 0.61 | 1.00 | 0.097 | *0.70* | **1.00** | **0.034** | **0.75** | *0.90* | **0.032** | 0.76 | **1.00** | **0.010** | 0.74 | 0.90 | **0.011** | **0.66** | 0.80 | **0.005** |
| (Ours) mse empty | *0.56* | **0.08** | 0.312 | 0.43 | 0.12 | 0.303 | 0.51 | 0.87 | 0.126 | 0.56 | *0.96* | 0.208 | 0.66 | 0.82 | 0.072 | 0.59 | 0.67 | 0.038 |
| (Ours) mse near | 0.65 | 0.32 | 0.376 | 0.56 | 0.48 | 0.415 | 0.59 | 0.82 | 0.151 | 0.71 | 0.90 | 0.104 | 0.65 | 0.81 | 0.057 | 0.59 | 0.38 | 0.076 |
| Concept Ablation | 0.64 | 0.43 | 0.222 | 0.51 | 0.55 | 0.185 | 0.66 | 0.79 | 0.053 | 0.71 | 0.95 | *0.048* | 0.68 | 0.89 | *0.019* | 0.61 | 0.68 | 0.007 |
| **Erasing Grumpy Cat** | | | | | | | | | | | | | | | | | | |
| Method | CS↑ | CA↑ | KID↓ | CS↓ | CA↓ | KID | CS↑ | CA↑ | KID↓ | CS↑ | CA↑ | KID↓ | CS↑ | CA↑ | KID↓ | CS↑ | CA↑ | KID↓ |
| Original Model | 0.70 | 0.97 | - | 0.77 | 1.00 | - | 0.70 | 0.98 | - | 0.73 | 0.97 | - | 0.77 | 0.97 | - | 0.65 | 0.78 | - |
| (Ours) hellinger empty | 0.58 | 0.70 | 0.105 | **0.50** | 0.10 | 0.409 | *0.69* | *0.95* | 0.139 | 0.75 | **1.00** | 0.096 | 0.69 | *0.95* | 0.058 | **0.69** | **1.00** | *0.014* |
| (Ours) hellinger near | *0.72* | **1.00** | 0.244 | *0.53* | 0.00 | 0.360 | **0.72** | **0.97** | *0.102* | 0.77 | **1.00** | 0.075 | *0.72* | **1.00** | 0.032 | 0.66 | 0.90 | 0.053 |
| (Ours) hellinger superclass | 0.64 | *0.90* | 0.095 | 0.54 | 0.40 | 0.280 | 0.66 | 0.83 | 0.113 | 0.72 | 0.90 | 0.157 | 0.68 | 0.83 | 0.069 | 0.64 | 0.70 | 0.065 |
| (Ours) hellinger superclass [‡] | **0.73** | **1.00** | **0.010** | 0.61 | 1.00 | 0.088 | **0.72** | 0.90 | **0.027** | 0.76 | **1.00** | 0.032 | **0.77** | **1.00** | 0.063 | 0.67 | 0.80 | **0.010** |
| (Ours) mse empty | 0.50 | 0.48 | 0.138 | 0.54 | 0.16 | 0.489 | 0.59 | 0.90 | 0.143 | 0.69 | *0.94* | 0.092 | 0.65 | 0.88 | 0.059 | 0.61 | 0.83 | 0.017 |
| (Ours) mse near | 0.58 | 0.50 | 0.328 | 0.59 | *0.04* | 0.435 | 0.68 | 0.79 | 0.112 | 0.73 | 0.93 | 0.117 | 0.68 | 0.91 | *0.037* | 0.61 | 0.63 | 0.053 |
| Concept Ablation | 0.52 | 0.33 | 0.201 | 0.58 | 0.05 | 0.321 | 0.63 | 0.78 | 0.155 | 0.66 | 0.75 | 0.180 | 0.65 | 0.84 | 0.082 | 0.55 | 0.43 | 0.091 |
| **Erasing Wall-E** | | | | | | | | | | | | | | | | | | |
| Method | CS↑ | CA↑ | KID↓ | CS↑ | CA↑ | KID↓ | CS↓ | CA↓ | KID | CS↑ | CA↑ | KID↓ | CS↑ | CA↑ | KID↓ | CS↑ | CA↑ | KID↓ |
| Original Model | 0.71 | 0.97 | - | 0.71 | 0.97 | - | 0.76 | 1.00 | - | 0.72 | 0.98 | - | 0.77 | 0.97 | - | 0.66 | 0.77 | - |
| (Ours) hellinger empty | 0.58 | 0.95 | 0.064 | 0.66 | 0.85 | *0.017* | **0.55** | 0.75 | 0.197 | 0.53 | 0.55 | 0.243 | 0.72 | 0.90 | 0.036 | *0.67* | *0.70* | *0.010* |
| (Ours) hellinger near | *0.64* | *0.97* | 0.050 | 0.68 | *0.97* | 0.118 | *0.56* | *0.67* | 0.212 | *0.71* | 0.90 | 0.249 | 0.70 | *0.97* | *0.033* | 0.61 | **0.73** | 0.074 |
| (Ours) hellinger superclass | 0.63 | 0.73 | 0.207 | **0.71** | 0.90 | 0.224 | 0.57 | 0.30 | 0.274 | 0.65 | 0.73 | 0.301 | 0.70 | 0.80 | 0.036 | 0.62 | 0.60 | 0.057 |
| (Ours) hellinger superclass [‡] | **0.72** | **1.00** | **-0.007** | 0.71 | **1.00** | **-0.003** | 0.70 | 1.00 | 0.075 | **0.76** | **1.00** | 0.062 | **0.76** | **1.00** | 0.031 | **0.68** | 0.60 | **-0.003** |
| (Ours) mse empty | 0.49 | 0.67 | *0.047* | 0.62 | 0.89 | 0.021 | 0.59 | 0.85 | 0.167 | 0.48 | 0.50 | 0.302 | 0.67 | 0.90 | 0.048 | 0.59 | 0.54 | 0.033 |
| (Ours) mse near | 0.54 | 0.57 | 0.163 | 0.62 | 0.81 | 0.176 | 0.66 | 0.96 | 0.154 | 0.69 | *0.91* | 0.199 | 0.63 | 0.75 | 0.037 | 0.58 | 0.62 | 0.047 |
| Concept Ablation | 0.54 | 0.42 | 0.228 | 0.56 | 0.56 | 0.282 | 0.64 | **0.30** | 0.241 | 0.56 | 0.30 | 0.328 | 0.59 | 0.55 | 0.083 | 0.53 | 0.43 | 0.122 |
| **Erasing R2D2** | | | | | | | | | | | | | | | | | | |
| Method | CS↑ | CA↑ | KID↓ | CS↑ | CA↑ | KID↓ | CS↑ | CA↓ | KID | CS↓ | CA↓ | KID | CS↑ | CA↑ | KID↓ | CS↑ | CA↑ | KID↓ |
| Original Model | 0.70 | 0.97 | - | 0.71 | 0.97 | - | 0.70 | 0.97 | - | 0.79 | 1.00 | - | 0.77 | 0.97 | - | 0.65 | 0.78 | - |
| (Ours) hellinger empty | 0.58 | 0.65 | *0.030* | **0.73** | 0.95 | *0.020* | 0.53 | 0.70 | *0.128* | **0.54** | 0.65 | 0.319 | *0.71* | 0.90 | 0.054 | *0.66* | 0.70 | 0.023 |
| (Ours) hellinger near | 0.61 | *0.87* | 0.110 | *0.63* | 0.73 | 0.168 | 0.61 | 0.67 | 0.271 | 0.53 | 0.10 | 0.403 | 0.66 | **1.00** | 0.150 | 0.61 | **0.83** | 0.145 |
| (Ours) hellinger superclass | *0.62* | 0.57 | 0.192 | 0.60 | 0.47 | 0.176 | *0.63* | 0.57 | 0.212 | 0.55 | 0.20 | 0.382 | 0.64 | 0.73 | 0.110 | 0.61 | 0.67 | 0.063 |
| (Ours) hellinger superclass [‡] | **0.73** | **1.00** | 0.001 | 0.73 | **1.00** | 0.019 | 0.72 | **1.00** | 0.043 | 0.63 | **1.00** | 0.222 | 0.76 | **1.00** | *0.052* | 0.68 | 0.80 | 0.002 |
| (Ours) mse empty | 0.50 | 0.83 | 0.049 | 0.59 | 0.76 | 0.037 | 0.49 | 0.64 | 0.142 | 0.59 | 0.77 | 0.385 | 0.67 | *0.93* | 0.062 | 0.60 | 0.54 | *0.017* |
| (Ours) mse near | 0.57 | 0.52 | 0.108 | 0.57 | 0.67 | 0.146 | 0.51 | 0.56 | 0.184 | 0.59 | 0.46 | 0.321 | 0.61 | *0.93* | 0.085 | 0.56 | *0.81* | 0.055 |
| Concept Ablation | 0.58 | 0.67 | 0.199 | 0.60 | 0.60 | 0.180 | 0.57 | *0.73* | 0.182 | 0.65 | 0.36 | 0.341 | 0.66 | 0.82 | **0.050** | 0.57 | 0.71 | 0.062 |
| **Erasing Van Gogh** | | | | | | | | | | | | | | | | | | |
| Method | CS↑ | CA↑ | KID↓ | CS↑ | CA↑ | KID↓ | CS↑ | CA↑ | KID↓ | CS↑ | CA↑ | KID↓ | CS↓ | CA↓ | KID | CS↑ | CA↑ | KID↓ |
| Original Model | 0.70 | 0.97 | - | 0.71 | 0.97 | - | 0.70 | 0.97 | - | 0.72 | 0.97 | - | 0.83 | 1.00 | - | 0.65 | 0.77 | - |
| (Ours) hellinger empty | 0.69 | 0.95 | 0.060 | 0.67 | *0.95* | 0.037 | 0.68 | **1.00** | 0.118 | 0.71 | **1.00** | 0.087 | **0.52** | 0.05 | 0.499 | 0.59 | 0.70 | 0.135 |
| (Ours) hellinger near | *0.71* | **1.00** | 0.031 | *0.68* | 0.90 | 0.075 | *0.75* | *0.97* | 0.059 | **0.78** | **1.00** | 0.088 | 0.64 | 0.53 | 0.275 | **0.71** | **1.00** | 0.188 |
| (Ours) hellinger superclass | 0.69 | *0.97* | 0.059 | **0.74** | **1.00** | 0.090 | 0.71 | *0.97* | 0.062 | 0.73 | **1.00** | 0.139 | *0.58* | *0.10* | 0.145 | 0.61 | 0.40 | 0.082 |
| (Ours) hellinger superclass [‡] | **0.75** | **1.00** | 0.016 | 0.74 | **1.00** | 0.023 | **0.76** | 0.90 | **0.022** | 0.75 | **1.00** | **0.036** | 0.66 | 0.60 | 0.122 | *0.70* | 0.90 | **-0.003** |
| (Ours) mse empty | 0.59 | 0.90 | *0.022* | 0.60 | 0.79 | *0.031* | 0.58 | 0.93 | 0.111 | 0.55 | 0.94 | 0.144 | 0.59 | 0.19 | 0.447 | 0.52 | 0.33 | 0.167 |
| (Ours) mse near | 0.67 | 0.95 | 0.056 | 0.63 | 0.94 | 0.099 | 0.70 | 0.93 | *0.055* | 0.73 | *0.95* | *0.080* | 0.65 | 0.37 | 0.292 | 0.61 | *0.93* | 0.209 |
| Concept Ablation | 0.65 | 0.93 | 0.068 | *0.68* | 0.91 | 0.040 | 0.68 | 0.86 | 0.057 | 0.73 | 0.94 | *0.080* | 0.65 | 0.24 | 0.129 | 0.58 | 0.60 | *0.069* |
| **Erasing Salvador Dali** | | | | | | | | | | | | | | | | | | |
| Method | CS↑ | CA↑ | KID↓ | CS↑ | CA↑ | KID↓ | CS↑ | CA↑ | KID↓ | CS↑ | CA↑ | KID↓ | CS↑ | CA↑ | KID↓ | CS↓ | CA↓ | KID |
| Original Model | 0.70 | 0.97 | - | 0.71 | 0.97 | - | 0.70 | 0.97 | - | 0.73 | 0.96 | - | 0.77 | 0.97 | - | 0.71 | 0.84 | - |
| (Ours) hellinger empty | 0.62 | 0.90 | 0.042 | 0.65 | *0.95* | 0.049 | 0.63 | 0.80 | 0.062 | 0.66 | 0.95 | 0.136 | 0.55 | 0.60 | 0.138 | *0.60* | 0.35 | 0.296 |
| (Ours) hellinger near | *0.73* | **1.00** | 0.015 | 0.72 | **0.97** | **0.005** | 0.69 | **0.97** | *0.046* | **0.78** | **1.00** | 0.101 | *0.70* | **1.00** | *0.081* | **0.57** | 0.07 | 0.179 |
| (Ours) hellinger superclass | 0.66 | 0.87 | 0.073 | 0.63 | 0.73 | 0.147 | 0.67 | **0.97** | 0.104 | 0.74 | *0.97* | 0.187 | 0.61 | 0.47 | 0.265 | **0.57** | 0.27 | 0.172 |
| (Ours) hellinger superclass [‡] | **0.76** | **1.00** | **0.004** | *0.70* | 0.90 | 0.015 | **0.78** | 0.90 | **0.015** | 0.76 | **1.00** | 0.085 | **0.75** | **1.00** | 0.064 | 0.63 | 0.50 | 0.022 |
| (Ours) mse empty | 0.58 | 0.89 | 0.032 | 0.58 | 0.94 | 0.052 | 0.55 | *0.93* | 0.077 | 0.60 | 0.95 | 0.114 | 0.47 | 0.39 | 0.132 | 0.63 | 0.45 | 0.280 |
| (Ours) mse near | 0.62 | *0.95* | *0.008* | 0.66 | 0.90 | 0.017 | 0.62 | 0.87 | 0.065 | 0.73 | 0.96 | *0.086* | 0.65 | *0.94* | 0.086 | 0.64 | *0.08* | 0.175 |
| Concept Ablation | 0.58 | 0.64 | 0.131 | 0.67 | 0.79 | 0.104 | 0.60 | 0.60 | 0.243 | 0.70 | 0.94 | 0.174 | 0.56 | 0.43 | 0.349 | 0.62 | 0.17 | 0.238 |

Table 8: Quantitative results with 150 iterations for variational methods, reporting CS, CA, and KID. For methods other than the Original Model, the best values are in bold and the second-best in italics. † denotes Importance Sampling and ‡ denotes Prior Preservation and Gradient Surgery. Base model: SD 1.4.

**Erasing Snoopy**

| Method | Snoopy CS↓ | CA↓ | KID | Grumpy Cat CS↑ | CA↑ | KID↓ | Wall-E CS↑ | CA↑ | KID↓ | R2D2 CS↑ | CA↑ | KID↓ | Van Gogh CS↑ | CA↑ | KID↓ | Salvador Dali CS↑ | CA↑ | KID↓ |
|---|---|---|---|---|---|---|---|---|---|---|---|---|---|---|---|---|---|---|
| Original Model | 0.73 | 1.00 | - | 0.74 | 1.00 | - | 0.73 | 1.00 | - | 0.75 | 1.00 | - | 0.80 | 1.00 | - | 0.68 | 0.80 | - |
| (Ours) hellinger empty | 0.52 | 0.00 | 0.269 | 0.67 | 0.67 | 0.177 | 0.71 | 1.00 | 0.098 | 0.71 | 1.00 | 0.128 | 0.76 | 0.97 | 0.083 | 0.66 | 0.70 | 0.023 |
| (Ours) hellinger empty ‡ | 0.55 | 0.10 | 0.355 | 0.74 | 1.00 | 0.186 | 0.74 | 1.00 | 0.091 | 0.76 | 1.00 | 0.154 | 0.76 | 1.00 | 0.027 | 0.64 | 0.73 | 0.027 |
| (Ours) hellinger empty ‡ † | 0.57 | 0.00 | 0.368 | 0.71 | 1.00 | 0.137 | 0.72 | 0.93 | 0.069 | 0.78 | 1.00 | 0.144 | 0.72 | 1.00 | 0.045 | 0.65 | 0.83 | 0.023 |
| (Ours) hellinger near | 0.54 | 0.17 | 0.276 | 0.79 | 1.00 | 0.127 | 0.78 | 1.00 | 0.079 | 0.78 | 1.00 | 0.078 | 0.80 | 1.00 | 0.037 | 0.68 | 0.87 | 0.001 |
| (Ours) hellinger near ‡ | 0.54 | 0.00 | 0.331 | 0.75 | 1.00 | 0.144 | 0.72 | 1.00 | 0.084 | 0.75 | 1.00 | 0.119 | 0.75 | 0.97 | 0.051 | 0.66 | 0.77 | 0.029 |
| (Ours) hellinger near ‡ † | 0.54 | 0.03 | 0.414 | 0.76 | 1.00 | 0.173 | 0.70 | 0.97 | 0.116 | 0.76 | 1.00 | 0.194 | 0.73 | 0.93 | 0.050 | 0.63 | 0.77 | 0.058 |
| (Ours) hellinger superclass | 0.55 | 0.00 | 0.295 | 0.76 | 1.00 | 0.144 | 0.77 | 0.93 | 0.076 | 0.78 | 1.00 | 0.084 | 0.79 | 1.00 | 0.055 | 0.68 | 0.90 | 0.012 |
| (Ours) hellinger superclass ‡ | 0.56 | 0.03 | 0.293 | 0.74 | 1.00 | 0.183 | 0.71 | 0.97 | 0.100 | 0.79 | 1.00 | 0.124 | 0.74 | 1.00 | 0.043 | 0.65 | 0.77 | 0.027 |
| (Ours) hellinger superclass ‡ † | 0.57 | 0.03 | 0.271 | 0.76 | 1.00 | 0.159 | 0.69 | 1.00 | 0.095 | 0.78 | 1.00 | 0.097 | 0.74 | 1.00 | 0.023 | 0.64 | 0.80 | 0.042 |

**Erasing Grumpy Cat**

| Method | Snoopy CS↑ | CA↑ | KID↓ | Grumpy Cat CS↓ | CA↓ | KID | Wall-E CS↑ | CA↑ | KID↓ | R2D2 CS↑ | CA↑ | KID↓ | Van Gogh CS↑ | CA↑ | KID↓ | Salvador Dali CS↑ | CA↑ | KID↓ |
|---|---|---|---|---|---|---|---|---|---|---|---|---|---|---|---|---|---|---|
| Original Model | 0.73 | 1.00 | - | 0.74 | 1.00 | - | 0.73 | 1.00 | - | 0.75 | 1.00 | - | 0.80 | 1.00 | - | 0.68 | 0.80 | - |
| (Ours) hellinger empty | 0.70 | 0.90 | 0.096 | 0.52 | 0.07 | 0.412 | 0.75 | 0.97 | 0.106 | 0.77 | 1.00 | 0.080 | 0.76 | 0.97 | 0.060 | 0.68 | 0.97 | 0.029 |
| (Ours) hellinger empty ‡ | 0.67 | 0.97 | 0.113 | 0.53 | 0.03 | 0.488 | 0.69 | 0.93 | 0.084 | 0.78 | 1.00 | 0.102 | 0.76 | 1.00 | 0.049 | 0.67 | 0.80 | 0.021 |
| (Ours) hellinger empty ‡ † | 0.69 | 1.00 | 0.091 | 0.52 | 0.00 | 0.482 | 0.69 | 0.93 | 0.094 | 0.78 | 1.00 | 0.103 | 0.73 | 1.00 | 0.064 | 0.66 | 0.93 | 0.023 |
| (Ours) hellinger near | 0.76 | 1.00 | 0.068 | 0.55 | 0.00 | 0.335 | 0.76 | 0.93 | 0.099 | 0.79 | 1.00 | 0.093 | 0.80 | 1.00 | 0.055 | 0.68 | 0.90 | 0.015 |
| (Ours) hellinger near ‡ | 0.70 | 1.00 | 0.121 | 0.51 | 0.00 | 0.489 | 0.74 | 1.00 | 0.077 | 0.78 | 1.00 | 0.128 | 0.75 | 1.00 | 0.038 | 0.67 | 0.80 | 0.014 |
| (Ours) hellinger near ‡ † | 0.69 | 1.00 | 0.093 | 0.51 | 0.00 | 0.553 | 0.74 | 0.97 | 0.082 | 0.79 | 1.00 | 0.080 | 0.77 | 1.00 | 0.042 | 0.69 | 0.90 | 0.032 |
| (Ours) hellinger superclass | 0.77 | 1.00 | 0.142 | 0.55 | 0.00 | 0.399 | 0.76 | 0.97 | 0.084 | 0.79 | 1.00 | 0.082 | 0.77 | 1.00 | 0.069 | 0.67 | 0.97 | 0.022 |
| (Ours) hellinger superclass ‡ | 0.71 | 1.00 | 0.061 | 0.54 | 0.00 | 0.508 | 0.72 | 0.97 | 0.070 | 0.78 | 1.00 | 0.064 | 0.74 | 1.00 | 0.059 | 0.66 | 0.87 | 0.023 |
| (Ours) hellinger superclass ‡ † | 0.64 | 1.00 | 0.150 | 0.51 | 0.00 | 0.541 | 0.71 | 0.90 | 0.100 | 0.78 | 1.00 | 0.088 | 0.74 | 0.97 | 0.043 | 0.67 | 0.77 | 0.026 |

**Erasing Wall-E**

| Method | Snoopy CS↑ | CA↑ | KID↓ | Grumpy Cat CS↑ | CA↑ | KID↓ | Wall-E CS↓ | CA↓ | KID | R2D2 CS↑ | CA↑ | KID↓ | Van Gogh CS↑ | CA↑ | KID↓ | Salvador Dali CS↑ | CA↑ | KID↓ |
|---|---|---|---|---|---|---|---|---|---|---|---|---|---|---|---|---|---|---|
| Original Model | 0.73 | 1.00 | - | 0.74 | 1.00 | - | 0.73 | 1.00 | - | 0.75 | 1.00 | - | 0.80 | 1.00 | - | 0.68 | 0.80 | - |
| (Ours) hellinger empty | 0.70 | 0.90 | 0.057 | 0.73 | 1.00 | 0.033 | 0.55 | 0.23 | 0.256 | 0.69 | 0.83 | 0.172 | 0.78 | 1.00 | 0.038 | 0.69 | 0.90 | −0.007 |
| (Ours) hellinger empty ‡ | 0.70 | 1.00 | 0.097 | 0.75 | 1.00 | 0.087 | 0.60 | 0.87 | 0.256 | 0.61 | 0.93 | 0.292 | 0.76 | 1.00 | 0.044 | 0.66 | 0.87 | 0.010 |
| (Ours) hellinger empty ‡ † | 0.69 | 1.00 | 0.112 | 0.75 | 1.00 | 0.046 | 0.59 | 0.37 | 0.349 | 0.64 | 1.00 | 0.394 | 0.78 | 1.00 | 0.040 | 0.67 | 0.87 | 0.027 |
| (Ours) hellinger near | 0.76 | 1.00 | 0.066 | 0.75 | 1.00 | 0.047 | 0.60 | 0.53 | 0.251 | 0.78 | 1.00 | 0.112 | 0.80 | 1.00 | 0.050 | 0.69 | 0.90 | -0.013 |
| (Ours) hellinger near ‡ | 0.71 | 1.00 | 0.129 | 0.75 | 1.00 | 0.076 | 0.56 | 0.27 | 0.294 | 0.68 | 1.00 | 0.283 | 0.79 | 1.00 | 0.039 | 0.66 | 0.83 | 0.006 |
| (Ours) hellinger near ‡ † | 0.68 | 1.00 | 0.110 | 0.74 | 1.00 | 0.064 | 0.60 | 0.53 | 0.305 | 0.65 | 0.90 | 0.309 | 0.77 | 1.00 | 0.049 | 0.68 | 0.93 | 0.012 |
| (Ours) hellinger superclass | 0.78 | 1.00 | 0.064 | 0.75 | 1.00 | 0.014 | 0.59 | 0.50 | 0.250 | 0.77 | 1.00 | 0.091 | 0.80 | 1.00 | 0.026 | 0.70 | 0.93 | -0.012 |
| (Ours) hellinger superclass ‡ | 0.72 | 1.00 | 0.094 | 0.76 | 1.00 | 0.060 | 0.61 | 0.67 | 0.316 | 0.69 | 1.00 | 0.259 | 0.78 | 1.00 | 0.048 | 0.66 | 0.83 | 0.003 |
| (Ours) hellinger superclass ‡ † | 0.72 | 1.00 | 0.112 | 0.75 | 1.00 | 0.040 | 0.58 | 0.33 | 0.341 | 0.69 | 0.97 | 0.277 | 0.78 | 1.00 | 0.053 | 0.68 | 0.93 | 0.004 |

**Erasing R2D2**

| Method | Snoopy CS↑ | CA↑ | KID↓ | Grumpy Cat CS↑ | CA↑ | KID↓ | Wall-E CS↑ | CA↑ | KID↓ | R2D2 CS↓ | CA↓ | KID | Van Gogh CS↑ | CA↑ | KID↓ | Salvador Dali CS↑ | CA↑ | KID↓ |
|---|---|---|---|---|---|---|---|---|---|---|---|---|---|---|---|---|---|---|
| Original Model | 0.73 | 1.00 | - | 0.74 | 1.00 | - | 0.73 | 1.00 | - | 0.75 | 1.00 | - | 0.80 | 1.00 | - | 0.68 | 0.80 | - |
| (Ours) hellinger empty | 0.70 | 0.90 | 0.082 | 0.74 | 0.97 | 0.017 | 0.63 | 0.87 | 0.134 | 0.57 | 0.40 | 0.394 | 0.77 | 1.00 | 0.052 | 0.68 | 0.80 | 0.032 |
| (Ours) hellinger empty ‡ | 0.65 | 0.97 | 0.148 | 0.74 | 1.00 | 0.094 | 0.60 | 0.83 | 0.215 | 0.56 | 0.03 | 0.549 | 0.77 | 1.00 | 0.030 | 0.65 | 0.90 | 0.042 |
| (Ours) hellinger empty ‡ † | 0.70 | 0.97 | 0.127 | 0.76 | 1.00 | 0.066 | 0.61 | 0.97 | 0.207 | 0.56 | 0.10 | 0.516 | 0.79 | 1.00 | 0.062 | 0.69 | 0.97 | 0.006 |
| (Ours) hellinger near | 0.78 | 1.00 | 0.138 | 0.74 | 1.00 | 0.070 | 0.67 | 1.00 | 0.135 | 0.55 | 0.23 | 0.431 | 0.79 | 1.00 | 0.049 | 0.70 | 0.90 | 0.004 |
| (Ours) hellinger near ‡ | 0.68 | 1.00 | 0.227 | 0.73 | 1.00 | 0.123 | 0.65 | 1.00 | 0.144 | 0.57 | 0.13 | 0.545 | 0.79 | 1.00 | 0.057 | 0.67 | 0.83 | 0.019 |
| (Ours) hellinger near ‡ † | 0.67 | 1.00 | 0.147 | 0.75 | 1.00 | 0.071 | 0.62 | 0.93 | 0.192 | 0.55 | 0.00 | 0.561 | 0.76 | 1.00 | 0.058 | 0.70 | 0.83 | 0.018 |
| (Ours) hellinger superclass | 0.75 | 1.00 | 0.118 | 0.73 | 1.00 | 0.057 | 0.67 | 0.97 | 0.105 | 0.57 | 0.37 | 0.407 | 0.79 | 1.00 | 0.039 | 0.69 | 0.87 | 0.007 |
| (Ours) hellinger superclass ‡ | 0.71 | 0.97 | 0.142 | 0.76 | 1.00 | 0.041 | 0.59 | 0.83 | 0.186 | 0.55 | 0.03 | 0.450 | 0.78 | 1.00 | 0.034 | 0.67 | 0.90 | -0.002 |
| (Ours) hellinger superclass ‡ † | 0.67 | 1.00 | 0.141 | 0.72 | 1.00 | 0.084 | 0.59 | 0.87 | 0.205 | 0.58 | 0.13 | 0.535 | 0.78 | 1.00 | 0.064 | 0.68 | 0.93 | 0.027 |

**Erasing Van Gogh**

| Method | Snoopy CS↑ | CA↑ | KID↓ | Grumpy Cat CS↑ | CA↑ | KID↓ | Wall-E CS↑ | CA↑ | KID↓ | R2D2 CS↑ | CA↑ | KID↓ | Van Gogh CS↓ | CA↓ | KID | Salvador Dali CS↑ | CA↑ | KID↓ |
|---|---|---|---|---|---|---|---|---|---|---|---|---|---|---|---|---|---|---|
| Original Model | 0.73 | 1.00 | - | 0.74 | 1.00 | - | 0.73 | 1.00 | - | 0.75 | 1.00 | - | 0.80 | 1.00 | - | 0.68 | 0.80 | - |
| (Ours) hellinger empty | 0.73 | 1.00 | 0.049 | 0.71 | 0.97 | 0.061 | 0.72 | 1.00 | 0.067 | 0.74 | 1.00 | 0.043 | 0.53 | 0.17 | 0.322 | 0.64 | 0.90 | 0.081 |
| (Ours) hellinger empty ‡ | 0.76 | 1.00 | 0.089 | 0.76 | 1.00 | 0.045 | 0.75 | 0.93 | 0.067 | 0.79 | 1.00 | 0.079 | 0.59 | 0.57 | 0.344 | 0.67 | 0.03 | 0.316 |
| (Ours) hellinger empty ‡ † | 0.77 | 1.00 | 0.141 | 0.72 | 1.00 | 0.038 | 0.75 | 1.00 | 0.074 | 0.79 | 1.00 | 0.066 | 0.61 | 0.40 | 0.341 | 0.56 | 0.23 | 0.283 |
| (Ours) hellinger near | 0.76 | 1.00 | 0.048 | 0.74 | 0.97 | 0.016 | 0.76 | 0.93 | 0.041 | 0.77 | 1.00 | 0.030 | 0.55 | 0.00 | 0.311 | 0.63 | 0.73 | 0.061 |
| (Ours) hellinger near ‡ | 0.78 | 1.00 | 0.129 | 0.74 | 1.00 | 0.053 | 0.72 | 0.97 | 0.088 | 0.78 | 1.00 | 0.086 | 0.58 | 0.07 | 0.296 | 0.59 | 0.37 | 0.143 |
| (Ours) hellinger near ‡ † | 0.75 | 0.97 | 0.150 | 0.71 | 1.00 | 0.050 | 0.73 | 0.87 | 0.068 | 0.80 | 1.00 | 0.106 | 0.61 | 0.53 | 0.305 | 0.56 | 0.10 | 0.313 |
| (Ours) hellinger superclass | 0.76 | 1.00 | 0.036 | 0.75 | 1.00 | 0.035 | 0.77 | 0.97 | 0.056 | 0.76 | 1.00 | 0.040 | 0.69 | 0.83 | 0.294 | 0.69 | 0.83 | 0.048 |
| (Ours) hellinger superclass ‡ | 0.76 | 1.00 | 0.120 | 0.75 | 1.00 | 0.056 | 0.75 | 0.93 | 0.082 | 0.78 | 1.00 | 0.057 | 0.58 | 0.20 | 0.365 | 0.58 | 0.23 | 0.272 |
| (Ours) hellinger superclass ‡ † | 0.74 | 1.00 | 0.114 | 0.74 | 1.00 | 0.050 | 0.73 | 0.97 | 0.069 | 0.79 | 1.00 | 0.073 | 0.61 | 0.73 | 0.360 | 0.55 | 0.13 | 0.298 |

**Erasing Salvador Dali**

| Method | Snoopy CS↑ | CA↑ | KID↓ | Grumpy Cat CS↑ | CA↑ | KID↓ | Wall-E CS↑ | CA↑ | KID↓ | R2D2 CS↑ | CA↑ | KID↓ | Van Gogh CS↑ | CA↑ | KID↓ | Salvador Dali CS↓ | CA↓ | KID |
|---|---|---|---|---|---|---|---|---|---|---|---|---|---|---|---|---|---|---|
| Original Model | 0.73 | 1.00 | - | 0.74 | 1.00 | - | 0.73 | 1.00 | - | 0.75 | 1.00 | - | 0.80 | 1.00 | - | 0.68 | 0.80 | - |
| (Ours) hellinger empty | 0.73 | 1.00 | 0.062 | 0.71 | 1.00 | 0.010 | 0.72 | 0.93 | 0.077 | 0.71 | 0.97 | 0.066 | 0.60 | 0.53 | 0.138 | 0.58 | 0.23 | 0.315 |
| (Ours) hellinger empty ‡ | 0.72 | 0.90 | 0.112 | 0.75 | 1.00 | 0.018 | 0.71 | 0.93 | 0.109 | 0.78 | 1.00 | 0.065 | 0.58 | 0.30 | 0.293 | 0.58 | 0.10 | 0.246 |
| (Ours) hellinger empty ‡ † | 0.73 | 0.93 | 0.118 | 0.71 | 1.00 | 0.042 | 0.75 | 1.00 | 0.084 | 0.80 | 1.00 | 0.056 | 0.56 | 0.10 | 0.295 | 0.59 | 0.03 | 0.281 |
| (Ours) hellinger near | 0.78 | 1.00 | 0.076 | 0.75 | 1.00 | -0.008 | 0.76 | 1.00 | 0.060 | 0.76 | 1.00 | 0.032 | 0.61 | 0.50 | 0.190 | 0.55 | 0.13 | 0.313 |
| (Ours) hellinger near ‡ | 0.72 | 0.90 | 0.092 | 0.74 | 1.00 | 0.018 | 0.76 | 0.93 | 0.093 | 0.78 | 1.00 | 0.071 | 0.60 | 0.30 | 0.284 | 0.55 | 0.00 | 0.410 |
| (Ours) hellinger near ‡ † | 0.73 | 0.97 | 0.116 | 0.73 | 1.00 | 0.019 | 0.76 | 0.97 | 0.048 | 0.79 | 1.00 | 0.058 | 0.56 | 0.20 | 0.251 | 0.57 | 0.17 | 0.281 |
| (Ours) hellinger superclass | 0.77 | 1.00 | 0.032 | 0.74 | 1.00 | -0.001 | 0.77 | 0.97 | 0.073 | 0.76 | 1.00 | 0.023 | 0.69 | 0.97 | 0.122 | 0.56 | 0.13 | 0.291 |
| (Ours) hellinger superclass ‡ | 0.75 | 0.90 | 0.120 | 0.72 | 0.97 | 0.011 | 0.75 | 0.90 | 0.072 | 0.78 | 1.00 | 0.050 | 0.57 | 0.37 | 0.260 | 0.58 | 0.17 | 0.273 |
| (Ours) hellinger superclass ‡ † | 0.74 | 0.93 | 0.122 | 0.73 | 1.00 | 0.016 | 0.76 | 0.93 | 0.070 | 0.79 | 1.00 | 0.055 | 0.58 | 0.23 | 0.247 | 0.57 | 0.03 | 0.357 |

Table 9: Unlearning comparison for *Van Gogh*. Lower CS/CA is better for the erased concept (↓), higher is better for preserved concepts (↑). 500 iterations for non variational and 150 iterations for variational. Base model: SD 1.4.

| | Erased (Van Gogh) | | Preserved Concepts | | | | | | | | | |
|---|---|---|---|---|---|---|---|---|---|---|---|---|
| | | | J. Mann | | J. Vermeer | | S. Dali | | G. Rutkowski | | Monet | |
| Method | CS (↓) | CA (↓) | CS (↑) | CA (↑) | CS (↑) | CA (↑) | CS (↑) | CA (↑) | CS (↑) | CA (↑) | CS (↑) | CA (↑) |
| Original Model | 0.80 | 1.00 | 0.78 | 1.00 | 0.83 | 1.00 | 0.69 | 0.78 | 0.54 | 0.34 | 0.74 | 1.00 |
| MSE (Closed-Form) | 0.70 | 0.71 | 0.76 | **1.00** | 0.79 | **1.00** | 0.67 | 0.72 | 0.52 | 0.48 | 0.70 | 0.98 |
| Hellinger (Closed-Form) | **0.67** | **0.69** | 0.78 | **1.00** | 0.80 | 0.98 | 0.68 | 0.70 | 0.54 | **0.50** | 0.71 | 0.98 |
| $\chi^2$ (Closed-Form) | **0.67** | 0.71 | 0.78 | **1.00** | 0.81 | 0.96 | 0.68 | 0.78 | 0.54 | 0.48 | 0.72 | 0.98 |
| KL (Variational) | 0.75 | 0.92 | 0.78 | **1.00** | 0.82 | **1.00** | 0.68 | 0.72 | 0.54 | 0.48 | 0.73 | **1.00** |
| Hellinger (Variational) | 0.80 | 1.00 | 0.78 | **1.00** | 0.83 | **1.00** | 0.69 | 0.76 | 0.55 | 0.48 | 0.75 | **1.00** |
| Jensen-Shannon (Variational) | 0.71 | 0.88 | 0.77 | **1.00** | 0.82 | **1.00** | 0.68 | 0.74 | 0.55 | 0.48 | 0.71 | 0.96 |
| $\chi^2$ (Variational) | 0.82 | 1.00 | **0.79** | **1.00** | **0.84** | **1.00** | **0.70** | **0.82** | **0.55** | 0.46 | **0.76** | **1.00** |

Table 10: Unlearning comparison for *R2D2*. Lower CS/CA is better for the erased concept (↓), higher is better for preserved concepts (↑). 500 iterations for non-variational and 150 iterations for variational. Base model: SD 1.4.

| | Erased (R2D2) | | Preserved Concepts | | | | | | | |
|---|---|---|---|---|---|---|---|---|---|---|
| | | | Baymax | | Wall E | | C-3PO | | Bb8 | |
| Method | CS (↓) | CA (↓) | CS (↑) | CA (↑) | CS (↑) | CA (↑) | CS (↑) | CA (↑) | CS (↑) | CA (↑) |
| Original Model | 0.78 | 1.00 | 0.77 | 0.96 | 0.75 | 0.84 | 0.77 | 0.90 | 0.74 | 0.90 |
| MSE (Closed-Form) | 0.62 | 0.04 | 0.75 | 0.93 | 0.72 | 0.74 | 0.75 | 0.88 | 0.70 | 0.86 |
| Hellinger (Closed-Form) | **0.60** | **0.01** | **0.76** | **0.94** | 0.73 | 0.74 | 0.76 | 0.84 | 0.71 | 0.88 |
| $\chi^2$ (Closed-Form) | 0.62 | **0.01** | 0.74 | 0.92 | 0.73 | 0.80 | **0.77** | **0.90** | 0.71 | 0.90 |
| KL (Variational) | 0.63 | 0.55 | 0.75 | 0.92 | **0.74** | **0.86** | 0.76 | 0.86 | 0.69 | 0.90 |
| Hellinger (Variational) | 0.65 | 0.55 | 0.75 | **0.94** | 0.73 | 0.84 | 0.76 | 0.86 | 0.69 | 0.88 |
| Jensen-Shannon (Variational) | 0.65 | 0.40 | 0.76 | **0.94** | 0.73 | 0.80 | 0.76 | 0.86 | 0.71 | 0.90 |
| $\chi^2$ (Variational) | 0.71 | 0.74 | 0.75 | 0.92 | 0.73 | 0.80 | 0.76 | 0.84 | **0.72** | **0.92** |