# OpenReview forum: "A Unified Framework for Diffusion Model Unlearning with f-Divergence"
_ICLR.cc/2026/Conference — Submitted to ICLR 2026_

### Official Review · Reviewer_ygvk · 2025-10-31

**Soundness:** 2
**Presentation:** 2
**Contribution:** 2
**Rating:** 2
**Confidence:** 4

**Summary:**

The paper proposes f-DMU, a unified framework for diffusion model unlearning based on f-divergence. It generalizes existing MSE-based and KL-based unlearning approaches by allowing any f-divergence. The method provides two formulations—closed-form and variational—to balance simplicity and generality. The authors offer theoretical analyses of gradient behavior and convergence properties, showing that different divergences lead to distinct unlearning dynamics. Empirical results on Stable Diffusion v1.4 demonstrate that the Hellinger-based loss achieves better trade-offs between erasure effectiveness and image quality preservation compared to the standard MSE loss.

**Strengths:**

1. The authors rigorously generalize MSE-based unlearning to a broad family of f-divergences, showing that KL divergence and MSE are just special cases. It demonstrates a strong theoretical foundation linking unlearning dynamics to divergence curvature and gradient scaling.

2. The paper derives explicit gradient behaviors and proves local exponential stability for the variational form. The gradient comparison gives interpretable insight.

3. The experimental results clearly support the theoretical claims. The Concept Ablation explicitly shows that bounded gradients prevent artifacts such as distorted faces or textures. This close alignment between mathematical reasoning and empirical observation increases the paper’s credibility.

**Weaknesses:**

1. Although the authors discuss how curvature affects convergence, they don’t propose a quantitative criterion for choosing which f-divergence to use in practice. To my understanding, I think the choice is heuristic. For example, while Table 1 suggests H^2 outperforms MSE for concept erasure, the paper provides no procedure for selecting between closed-form vs. variational objectives or among multiple f-divergences based on task statistics.

2. All experiments are performed on Stable Diffusion v1.4 with a few synthetic concepts. Considering many DM unlearning works are performing unlearn multiple concepts at once, how does the proposed method performs when unlearn many concepts (say > 10 concepts)? It is important that the unlearning algorithm can handle such practical scenario, which demonstrate the robustness of the method.

3. While CAbl and MSE-based methods are compared, newer or adversarial unlearning methods (e.g., DoCo[1] or defensive adversarial unlearning [2]) are omitted. Besides these two, the author also needs to compare with other baselines such as SalUn, FMN, SPM, ESD, MACE, etc. I suggest the author compare with similar approach unlearning and different types of unlearning, which will make the proposed method more convincing.


[1] Wu, Yongliang, et al. "Unlearning concepts in diffusion model via concept domain correction and concept preserving gradient." Proceedings of the AAAI Conference on Artificial Intelligence. Vol. 39. No. 8. 2025.

[2] Zhang, Yimeng, et al. "Defensive unlearning with adversarial training for robust concept erasure in diffusion models." Advances in neural information processing systems 37 (2024): 36748-36776.

**Questions:**

Please refer to weaknesses.

---

> ### Author Response · Authors · 2025-11-19
>
> We thank the Reviewer for the detailed assessment and for acknowledging the **strong theoretical foundation** of f-DMU, particularly the generalization of existing MSE-based unlearning, the **rigorous gradient and convergence analyses**, and the clear **alignment between theory and empirical results** (e.g., Hellinger's bounded gradients preventing artifacts). We appreciate your critical feedback and address your concerns below
>
> **Responses to weakness**:
> 1. **Criterion for the choice of f-divergence**
> - We provide **clear guidelines** based on theoretical analysis and empirical results: **1) Closed-form vs. Variational**: Addressed in lines 429-431 of the revised paper. **2) Choosing f-Divergence**: for closed-form losses, we addressed it with the **gradient analysis** (Sec. 4.1, 5.2, 5.3) showing that different divergences lead to distinct dynamics (e.g., Hellinger's bounded gradient for better image quality preservation). For variational-based losses, we addressed it with our **theoretical analysis on convergence speed** (Sec. 4.2). Furthermore, we provide additional comments on f-divergences in lines 297-308 of the revised paper.
> - To improve the clarity of the paper, we additionally include in the revised version a **summary table** (Tab. 5 in the new paper), mapping the user’s specific unlearning goal to the optimal f-divergence and framework choice.
> 2. **All experiments on SD1.4 with few concepts**
> - We have expanded our empirical evaluation to **SD 2.1** (new Tab. 2) and **SDXL** (new Figure 5), to demonstrate the validity of f-DMU and to check the consistency of our theoretical analyses with different T2I models.
> - Regarding **multi-concept unlearning**, we already showed examples for the unlearning of three concepts (Figure 13 in the revised manuscript), where Hellinger loss performed the best. Since our framework is unified and our findings are theoretically grounded, these observations scale to the case of **10 artists erasure** (new Tab. 3 and new Figure 8).
> 3. **Comparison with SOTA baselines**
> - We reiterate that our primary contribution is the **theoretically-grounded unified framework**, which extends existing methods, rather than proposing a new SOTA approach.
> - We have included in the revised version a comparison (new Tab. 2) between f-DMU (both closed-form and variational) against Concept Ablation, UCE, and DoCo. Furthermore, we added an attack robustness analysis (new Figure 7) using Ring-A-Bell.
>
> We are confident that these major revisions, the new practical guide for divergence selection, experiments on SD2.1 and SDXL, a 10-concept erasure demonstration, and comparisons against multiple SOTA baselines fully address the issues raised by the Reviewer. The paper now offers a strong, well-supported, and broadly applicable framework for diffusion model unlearning. We hope the Reviewer will find these improvements satisfying and reconsider their evaluation.

---

### Official Review · Reviewer_thXe · 2025-11-02

**Soundness:** 3
**Presentation:** 3
**Contribution:** 2
**Rating:** 6
**Confidence:** 4

**Summary:**

This paper proposes a unified framework for diffusion model (DM) unlearning based on f-divergences. The work introduces a theoretical generalization of concept erasure via any f-divergence in DM. The contributions are: closed-form derivations for special cases (Hellinger, χ², Jeffreys), a variational adversarial formulation for general divergences and gradient behavior analysis & local convergence guarantees. Authors show empirical evaluation on Stable Diffusion 1.4 with multiple concepts and anchors. The paper argues that Hellinger divergence yields better stability and prior preservation while χ² gives more aggressive unlearning.

**Strengths:**

1) The paper elegantly reformulates prior MSE-based objectives as specific instances of the f-divergence family, providing a principled foundation for a range of unlearning losses.
2) The gradient and convergence analyses are rigorous, connecting divergence curvature to optimization dynamics in a meaningful way.
3) The experiments demonstrate that the squared Hellinger divergence achieves bounded gradients, leading to stable unlearning with fewer artifacts, a convincing empirical validation of the theory.
4) The inclusion of post-, fine-, and variational-based unlearning frameworks in the appendix positions this as a unifying theoretical effort.

**Weaknesses:**

1) Limited baselines: comparison with several recent baselines (e.g., MACE, SAFREE) are missing in the experiments section.
2) Empirical focus is narrow. All experiments use the same model (Stable Diffusion 1.4). Although reasonable, it would help to clarify whether results depend on architecture details or on properties of the diffusion loss. It is unclear how the proposed f-divergence solution are scalable to newer diffusion architectures (e.g., SDXL, Flux).

**Questions:**

Q1. Did you try benchmarking on LDM-based SDXL and Flow-matching models to validate generality?
Q2. Does the insight on having a bigger family of f-divergence lead to better unlearning performance? This requires comparison with latest DM unlearning methods which is missing in the paper. Particularly, the authors do not discuss the unlearned model's robustness to attacks like membership inference attacks.
Q3. Could you report compute cost vs standard MSE KL-minimization?
Q4. The authors highlight Jeffreys, Hellinger, and χ² but omit other f-divergences (e.g., JS, Rényi). A short discussion on why these three were prioritized would improve completeness.

---

> ### Author Response · Authors · 2025-11-19
>
> We thank the Reviewer for their positive feedback, particularly acknowledging the **principled foundation** provided by reformulating unlearning via the f-divergence family, the **rigorous gradient and convergence analyses**, and the strong **empirical validation** linking Hellinger divergence's bounded gradients to stable unlearning with fewer artifacts.
> We have **significantly expanded the experimental section** to provide a more comprehensive validation of our framework.
>
> **Responses to Weaknesses and Questions**:
> 1. **Limited baselines (W1 & Q2)**
> - While our primary contribution is the **theoretically-grounded unified framework**, we have added new comparisons in Tab. 2 on SD2.1 against recent methods like DoCo (2025) and UCE (2024). These results demonstrate the competitive performance of our Hellinger-based approach. [**Note**: We were unable to include MACE and SAFREE in this rebuttal cycle as the public SD2.1 checkpoint was removed from Hugging Face, preventing us from running the code. We will include them in the final version if a checkpoint becomes available.]
> - Regarding robustness (part of Q2), we have also included a **new adversarial analysis using Ring-A-Bell** (new Figure 7), which reveals interesting performance differences between divergences under attack, further highlighting the value of our unified view.
> 2. **Narrow Empirical Focus (W2 & Q1)**
> Thank you for encouraging us to validate the generality of our framework.
> - Our framework's formulation is **theoretically independent of the diffusion model architecture**, as the loss is computed on the noise prediction or generated images. This is a significant advantage over methods tied to specific architectures.
> - To empirically prove this, we have performed new experiments on newer architectures. We now include results on **SD2.1** (new Tab. 2) and on **SDXL** (new Figure 5). The SDXL experiment confirms that the gradient dynamics we analyzed persist in state-of-the-art models, underscoring the broad relevance of our findings. f-DMU is also directly applicable to flow-matching models like Flux, updating different parameter subsets (e.g.,  'context_embedder' or 'text_embedder.linear_1').
> 3. **Compute Cost (Q3)**
>  The compute cost for our closed-form losses (Hellinger, $\chi^2$) is **identical** to the standard MSE loss used in Concept Ablation (approx. 2.5 minutes for 500 steps on an NVIDIA GeForce RTX 5090). The variational framework is more expensive due to the need to train a discriminator (approx. 13.5 minutes for 500 steps). We have clarified this in the paper.
> 4. **Choice of f-divergence for closed-form losses (Q4)**
> We prioritized these three divergences because they admit closed-form expressions with peculiar behaviors. In the revised version of the paper (see Sections B.1.1, B.1.2, and new Figure 12), we provide an extended class of closed-form loss functions **for any $\alpha$-divergence**, which is a subclass of f-divergences including squared Hellinger distance ($\alpha=0.5$) and Pearson $\chi^2$ ($\alpha=2$). **We extended the gradient analysis** that we previously carried out for MSE, $H^2$, and $\chi^2$ for any $\alpha$-divergence (see new Figure 12), showing the different behavior for varying values of $\alpha$. This generalized analysis, which highlights the unique gradient behavior of the Hellinger distance, further justifies the importance of the chosen divergence.
>
> We thank the Reviewer for the feedback and for helping us improve the quality of our paper. We believe that the explanations provided and additional results performed make the contributions of the paper clearer and stronger. We hope the Reviewer agrees and will consider our work more favorably.

---

### Official Review · Reviewer_PcLV · 2025-11-03

**Soundness:** 3
**Presentation:** 2
**Contribution:** 3
**Rating:** 6
**Confidence:** 3

**Summary:**

The paper presents a generalization of the KL divergence-based unlearning algorithm to a broader class of f-divergence and performs a detailed analysis of the behavior of different divergence measures. Specifically, the paper compares the closed-form solution of Squared Hellinger ($H^2$) distance, the Pearson $\chi^2$ divergence, and the KL-divergence measures. The gradient analysis shows that $H^2$ distance has a much lower gradient norm, which leads to a more stable unlearning process with less disruption on the nearby unrelated concepts. The paper also proposes a variational framework for a general f-divergence-based training, but this is shown to be much more disruptive, although fast.

**Strengths:**

1. The generalization of the unlearning objective from MSE/KL divergence to a unified f-divergence framework is really insightful. It offers a flexible paradigm that allows for balancing the trade-offs between unlearning effectiveness and the preservation of other concepts. The theoretical analysis of the gradients for the closed-form loss functions provides a reasonable rationale for their differing empirical behaviors.

2. The experimental results align well with the gradient analysis. The $H^2$ divergence is shown to better preserve generative quality on unrelated concepts (as measured by metrics like KID) when compared to MSE, which appears to cause more degradation.

**Weaknesses:**

1. The empirical comparison is primarily against KL-based methods like Concept Ablation. The evaluation would be more comprehensive if benchmarked against other recent state-of-the-art baselines, such as EraseAnything [1] or MACE [2].

2. In addition, an analysis of adversarial robustness (e.g., using a method like Ring-A-Bell [3]) can provide a stronger and more rigorous measure of how completely a concept has been erased. The robustness behaviour of different divergence measures can be interesting to know as well.

3. The paper's evaluation of collateral damage from the unlearning method focuses on "unrelated concepts." A detailed quantitative evaluation of the impact on semantically nearby concepts would be valuable. For instance, in the style removal experiments, does unlearning "Van Gogh" (an Impressionist) also degrade the model's ability to generate "Monet" or "Impressionist paintings"? The current analysis (e.g., Fig. 4) is helpful, but a quantitative study on this would strengthen the work (as done in Concept Ablation as well)

4. The advantages of the variational framework versus the closed-form solutions are not fully clear. The results show it is faster while also more disruptive, but a deeper analysis of why the min-max optimization leads to faster convergence and a discussion of its stability trade-offs would be beneficial for developing a more general-purpose unlearning algorithm based on this paradigm.

[1] EraseAnything: Enabling Concept Erasure in Rectified Flow Transformers
[2] Mace: Mass concept erasure in diffusion models
[3] Ring-a-bell! how reliable are concept removal methods for diffusion models?

**Questions:**

Please look at the weakness section.

---

> ### Author Response · Authors · 2025-11-19
>
> We thank the Reviewer for their positive assessment, recognizing the insightfulness of the f-divergence generalization and the value of our theoretical gradient analysis.
> We have significantly expanded the experimental section to provide a more comprehensive validation of our framework.
>
> **Responses to weaknesses**:
> 1. **Empirical Comparison Against State-of-the-Art Baselines**
> - We have added new quantitative results in new Tab.2 on SD2.1, explicitly comparing various f-divergence losses against established recent baselines like DoCo (2025) and UCE (2024), alongside Concept Ablation.
> [**Note**: We were unable to include MACE and EraseAnything in this rebuttal cycle as the public SD2.1 checkpoint was removed from Hugging Face, preventing us from running the code. We will include them in the final version if a checkpoint becomes available.]
> - **Crucially, we emphasize that the core contribution of this paper is the theoretically-grounded unified f-divergence framework**, not proposing a new state-of-the-art method. The new comparisons, however, show the competitive performance of the H$^2$ divergence and the flexibility of f-DMU.
> 2. **Analysis of Adversarial Robustness using Ring-A-Bell**
> Thank you for the excellent suggestion. We have added the Ring-A-Bell analysis and presented the results in the new Figure 7.
> - We observe that the **variational framework is more susceptible to prompt attacks** than the closed-form losses.
> - **Detailed explanation**: This difference is theoretically linked to the stochasticity of the variational loss estimation. Unlike the closed-form losses (which minimize the true divergence given the current samples), **the variational approach requires minimizing an inner optimization to estimate the divergence**. When running the code with the small batch sizes typical for unlearning methods, this inner estimate is a high-variance approximation of the true divergence. This imprecision leads to a "rougher" concept boundary shift, leaving behind concept remnants that are more easily exposed by targeted adversarial prompts like Ring-A-Bell.
> 3. **Quantitative Evaluation of Impact on Semantically Near Concepts**
> - We have added Table 2 to the revised manuscript which explicitly measures the generative quality for semantically related concepts, for SD 2.1.
> - This new analysis quantitatively confirms the observation that H2, due to its bounded gradient, exhibits significantly lower collateral damage on nearby concepts compared to MSE, strengthening our claims on prior preservation. We also point out that Tab. 6 in the original appendix (Tab. 9 in the revised paper) already provided preliminary evidence for this (for SD 1.4), but the new table consolidates and expands this finding.
> 4. **Variational Framework's Trade-offs**
> - The trade-off is directly related to the point raised in Weakness 2. The small batch size causes the divergence estimate to be a rough approximation. When the diffusion model minimizes this noisy estimate, the resulting **distribution shift is imprecise** and does not necessarily lead to a perfect replacement with the anchor concept, thus leading to a quick concept degeneration. This **large, noisy shift rapidly erodes the concept** (leading to "faster convergence") but also leads to disruptive, unstable changes in the surrounding distribution and higher degradation. On the other hand, the **closed-form losses guarantee that we are minimizing a specific divergence for any batch size**, and having a small batch size only causes the model to see less examples of the concept to unlearn.
> - We have significantly revised Section 5 to include this deeper analysis, framing the variational framework as a high-speed, high-risk approach fundamentally limited by the stochastic estimation of the divergence.
>
> We believe these additions, new SOTA comparisons, a rigorous adversarial robustness analysis, quantitative evaluation of collateral damage on semantically near concepts, and a clearer explanation of the variational framework, substantially strengthen the paper. We hope the Reviewer finds these updates compelling and will consider them favorably.

---

> > ### Comment · Reviewer_PcLV · 2025-11-26
> > **Thanks for your comments**
> >
> > Hi, thanks a lot for addressing the concerns raised in the initial review. Some of the questions/concerns that I now have are:
> >
> > 1. I agree that the core contribution of this paper is the grounding the baselines into a unified f-divergence framework. However, it would be great to highlight its advantage, for example, H^2 and variational divergence are derived through this grounding but do not seem to show significant advantages in terms of the final performance.
> >
> > 2. Continuing on the previous point, and regarding the lower performance of variational divergence, its mentioned that its because of a lower batch-size resulting in high variance estimate. Adding an experiment that shows that with larger batch-size performance improves, would be really useful to validate this point and help towards better understanding of the framework in general.
> >
> > 3. I agree with other reviewer's concern about limited quantitative evaluation, such as Table 1 and 2 only show results when erasing 2 and 1 concept respectively. Expanding the set of earsed concepts to 10-15 and reporting average metrics will help in understanding the general trend across different f-divergence metrics and baselines.

---

> ### Author Response · Authors · 2025-11-27
>
> We thank the Reviewer for the additional questions. We invite the Reviewer to check the revised version of the paper (uploaded on Openreview). Please find below the point-to-point rebuttal:
> 1. **Advantages of H$^2$ and variational framework**
> In the Results section of the revised paper there are many qualitative and quantitative results highlighting that the H$^2$ closed-form loss function outperforms the MSE:
> - **Qualitative results: comparison between H$^2$ and MSE**: a) Figure 2 reports the better robustness to training instability. b) Figure 4 shows a better prior preservation and visual coherence. c) Figure 5 shows a higher image detail for SDXL (the MSE generated images are significantly blurred). d) Figure 6 shows better performance on different choices of anchor concept. e) Figure 7 shows better robustness (lower CA) to Ring-A-Bell framework. f) Figure 8 shows a better performance on the task of **10 artistic styles erasure**.
> - **Quantitative results: comparison between H$^2$ and MSE**: a) Tab. 1 shows the better performance for both erasure and prior preservation on SD 1.4. b) Tab. 2 (compare CAbl and Hellinger (closed-form)) shows better erasure and prior preservation on SD 2.1. c) Tab. 3 shows better prior preservation and comparable erasure performance when **erasing 10 artistic styles**.
> **All these numerical results are supported by our theoretical analysis**.
> For the variational framework, we refer to lines 429-432 where we explain its utility. The flexibility in the framework choice demonstrates the breadth of f-DMU.
> 2. The small batch size argument is a formal and well-documented argument for divergence estimation. Indeed, many works demonstrate that divergence estimation methods perform better when increasing the number of samples used for the estimate (e.g., [1,2]). In particular, for the variational formulation, two expected values must be computed, which are **approximated using Monte Carlo estimation**. This **Monte Carlo approximation yields a high-variance estimate when using small batch sizes**. To obtain accurate divergence estimates, very large batches (hundreds of images) are required, given the high-dimensionality of the image distributions. This would significantly slow down the training process, thus making this technique impractical. Finally, significantly increasing the batch size is limited by the memory constraints of our NVIDIA GeForce RTX 4090.
> 3. We have expanded the set of erased concepts to 10 and analyzed the performance in Tab. 3 (we already did this to answer question 2 of Reviewer ygvk). The metrics reported in Tab. 3 are averaged over multiple random seeds (as reported in its caption).
>
> [1] Moon, Kevin R., and Alfred O. Hero. "Multivariate f-divergence estimation with confidence." Advances in neural information processing systems 27 (2014).
> [2] Rubenstein, Paul, et al. "Practical and consistent estimation of f-divergences." Advances in Neural Information Processing Systems 32 (2019).

---

### Official Review · Reviewer_zCjU · 2025-11-03

**Soundness:** 2
**Presentation:** 1
**Contribution:** 2
**Rating:** 2
**Confidence:** 3

**Summary:**

The authors generalize existing concept unlearning methods in diffusion models by replacing KL divergence with more general f-divergence. The authors then theoretically analyze the impact of different choices of f on convergence behaviors near equilibria and gradient size. Empirically, it was found that the squared Hellinger distance has more stable gradient updates and better prior preservation.

**Strengths:**

- A detailed convergence analysis is provided and a theoretically principled way of choosing f to have better convergence properties is provided

**Weaknesses:**

- The results are poorly presented and some claims are not well justified
   - For example, in Figure 1, it is unclear how to interpret the white boxes that zoom in on portions of the image. Additionally, Figure 3 has a lot of noise in the plot, so some smoothing and a better caption explaining the main takeaway would help a great deal.
   - While a great deal of theoretical emphasis in the paper is regarding the variational approach, it appears details surrounding the training procedure on the minimax objective are not provided and the empirical results are minimal (lines 402-412) and not favorable.
   - There appears to be somewhat of an overreliance on qualitiative results for supporting important claims such as "This behavior characterizes the losses properties and explains why, for instance, H2 less probably leads to strange artifacts during fine-tuning." While Figure 2 is a great to include for us to visualize the nature of the benefit of H2, it would be great to quantitatively measure and keep track of these strange artifacts across multiple examples and present it in a table. Otherwise, the reader is left to believe that the Van Gogh example was ad hoc.
- Small note: It appears many citations that need to be using parantheticals are not using them (e.g., related works section doesn't have any paranthetical citations).

While I believe there is merit to the exploration of different choices of f and the authors' theoretical analysis, I believe the paper and experimental results will need a couple more iterations to be fully polished and conference ready.

**Questions:**

Please address the comments and concerns in the weaknesses section.

---

> ### Author Response · Authors · 2025-11-19
>
> We thank the Reviewer for their careful reading, helpful suggestions, and acknowledgment of the paper's theoretical contributions regarding the use of f-divergence and convergence analysis. We have significantly revised the manuscript based on the comments, particularly regarding presentation and experimental depth.
>
> **Responses to weaknesses**:
> 1. **Clarity of Figures 1 and 3**
> - To enhance clarity in Figure 1, we have updated the caption to explicitly state that the white boxes are magnified views highlighting the local removal of the Van Gogh brushstroke style, which may not be apparent at a glance.
> - We have updated Figure 3 by applying a moving average to smoothen the gradient amplitude plots. The caption has also been rewritten to improve clarity.
> 2. **Variational loss details and additional experiments**
> - We have expanded Appendix C.1 with comprehensive implementation details for the minimax objective, ensuring full reproducibility.
> - We have added **new quantitative results** (new Tab. 2) comparing both closed-form and variational f-DMU on SD2.1 across various f-divergences. While our original appendix contained preliminary results (Tabs. 9 and 10 in the revised manuscript), this new table provides an additional and clearer comparison.
> - In the revised manuscript, we have added a new figure (Figure 7), showing the performance difference between closed-form and variational f-DMU on the Ring-A-Bell benchmark, with a detailed discussion of the results.
> 3. **Lack of quantitative results**
> - The instability artifact exemplified in Figure 2 is a visual outcome of the large-magnitude gradient events due to the f-divergence choice. The gradient amplitude study is shown in Figure 3, which is a **quantitative** plot demonstrating the validity of the theoretical analysis in Section 4.1. We have other examples of instability during fine-tuning, but there is not a clear definition of an “artifact” in our case, so there is not a precise way of systematically counting these different artifact types. The fine-tuning instability is less evident (but still present) under other forms. For instance, the instability can manifest as a transient decay: the generated image $x$ temporarily deviates into an intermediate (often degraded) state $x^1$, and subsequently returns back to $x$. However, we believe that the **combination of Figure 2 and Figure 3** properly demonstrates the higher possibility of **instability of MSE compared to Hellinger-based loss**.
>  - Related to the importance of our gradient analysis, we have reported an example in Figure 5 in the revised manuscript about its effect on **SDXL**.
> 4. Thank you. We have adjusted the usage of parenthetical citations in the updated manuscript.
>
> We believe that the significant improvements in presentation, the expanded quantitative results for the closed-form and variational framework, and the new evidence supporting the gradient analysis importance fully address the Reviewer's concerns regarding soundness and contribution. The paper now offers a polished theoretical framework fully supported by more robust experimental evidence.

---

### Meta-Review · Area_Chair_7CmM · 2025-12-25

**Summary:**

This paper proposes an extension to the diffusion model unlearning by generalizing existing KL/MSE-based objectives to the broader family of f-divergences. The authors studied closed-form losses for certain divergences (e.g., squared Hellinger), and propose the variational min–max formulation for general f-divergences. The paper also provides a theoretical analysis of gradient behavior and local convergence.

All reviewers acknowledged the value of the theoretical framing and analysis; however, they consistently raised concerns regarding the insufficient empirical breadth of the current work. While the rebuttal added additional experiments and clarifications, these additions did not, in the AC’s assessment, fundamentally address the core concerns. Therefore, the AC, sadly, recommends **rejection**.

**Reviewer Concerns:**

A summary of reviewers' concerns is provided below

- Reviewer **zCjU** found the experimental validation limited.

- Reviewer **PcLV** mentioned that empirical comparisons are mostly against KL/MSE-style baselines, and requested broader benchmarking against recent SOTA concept erasure methods. They also suggested adding an adversarial robustness evaluation (e.g., Ring-A-Bell) to more rigorously test erasure performance. Another concern is related to collateral-damage analysis on semantically nearby concepts. The reviewer also found the analysis and discussion on the benefits/trade-offs of the variational framework vs closed-form losses insufficient.

- Reviewer **thXe** raised concerns about the limited empirical scope of the evaluation, particularly the lack of comparisons with recent diffusion unlearning methods. They also requested robustness analysis (eg., using membership inference attacks), and computational cost.


- Reviewer **ygvk** questioned the comprehensiveness of the empirical evaluation (absence of comparisons with SOTA diffusion unlearning baselines and limited evidence for large-scale multi-concept removal). The reviewer also identified the lack of a quantitative, principled criterion for selecting among f-divergences or between closed-form and variational objectives as a weakness.


Overall, the reviews consistently highlight a lack of compelling empirical evidence, positioning the proposed framework as a meaningful advance in diffusion model unlearning. Although the authors expanded the experimental section in response to reviewer feedback, the AC remains unconvinced that the results establish a clear improvement in unlearning performance or conceptual understanding over prior work. In addition, key theoretical elements rely on known formulations, and their contribution to advancing unlearning methodology is not sufficiently demonstrated.

**Reviewer Scores:**

The two reviewers with negative scores (**ygvk** and **zCjU**) raised fundamental concerns about empirical completeness, multi-concept unlearning, and comparisons with SOTA baselines, and it is unlikely that their scores would have changed to positive based on the rebuttal. The reviewers who scored the paper at 6 (**thXe** and **PcLV**) were generally positive about the theoretical framing but identified gaps in empirical validation and practical justification. It is unlikely that these reviewers would have increased their scores beyond their original assessments and champion the paper in its current state.

---

### Decision · Program_Chairs · 2026-01-26

Reject